# Analysis of Linear Mode Connectivity via Permutation-Based Weight Matching: With Insights into Other Permutation Search Methods

**Akira Ito[1], Masanori Yamada[1] & Atsutoshi Kumagai[2]**
[1]NTT Social Informatics Laboratories   [2]NTT Computer and Data Science Laboratories
`{akira.itoh, masanori.yamada, atsutoshi.kumagai}@ntt.com`

## Abstract

Recently, Ainsworth et al. (2023) showed that using weight matching (WM) to minimize the $L^2$ distance in a permutation search of model parameters effectively identifies permutations that satisfy linear mode connectivity (LMC), where the loss along a linear path between two independently trained models with different seeds remains nearly constant. This paper analyzes LMC using WM, which is useful for understanding stochastic gradient descent's effectiveness and its application in areas like model merging. We first empirically show that permutations found by WM do not significantly reduce the $L^2$ distance between two models, and the occurrence of LMC is not merely due to distance reduction by WM itself. We then demonstrate that permutations can change the directions of the singular vectors, but not the singular values, of the weight matrices in each layer. This finding shows that permutations found by WM primarily align the directions of singular vectors associated with large singular values across models. This alignment brings the singular vectors with large singular values, which determine the model's functionality, closer between the original and merged models, allowing the merged model to retain functionality similar to the original models, thereby satisfying LMC. This paper also analyzes activation matching (AM) in terms of singular vectors and finds that the principle of AM is likely the same as that of WM. Finally, we analyze the difference between WM and the straight-through estimator (STE), a dataset-dependent permutation search method, and show that WM can be more advantageous than STE in achieving LMC among three or more models.

## 1 Introduction

Large-scale neural networks (NNs) are widely used in various fields (Vaswani et al., 2017; van den Oord et al., 2016; Zhao et al., 2023), and optimizing their parameters poses a massive non-convex optimization problem. Remarkably, stochastic gradient descent (SGD), which is widely used for training NNs, is known to find good solutions despite its simplicity. One hypothesis for this seemingly counterintuitive phenomenon is that the landscape of the loss function may be much simpler than previously thought. Several studies (Garipov et al., 2018; Draxler et al., 2018; Freeman & Bruna, 2017) have found that different NN solutions can be connected by simple nonlinear paths with almost no increase in loss. Recently, Entezari et al. (2022) conjectured that Conjecture 1.1 holds, considering all possible permutation symmetries of NNs:

**Conjecture 1.1** (Permutation invariance, informal)**.** *Let $\boldsymbol{\theta}_a$ and $\boldsymbol{\theta}_b$ be two SGD solutions (model parameters). Then, with high probability, there exists a permutation $\pi$ such that the barrier (defined in Definition 2.1) between $\boldsymbol{\theta}_a$ and $\pi(\boldsymbol{\theta}_b)$ is sufficiently small.*

Here, the barrier represents the increase in loss observed when linearly interpolating between the weights of the two models. If the barrier between two models is sufficiently small, we say that linear mode connectivity (LMC) is satisfied between them (Frankle et al., 2020). Conjecture 1.1 suggests that most SGD solutions can be transferred into the same loss basin using permutations. Indeed, some studies (Ainsworth et al., 2023; Singh & Jaggi, 2020) have experimentally demonstrated that this conjecture is valid for various datasets and models using weight matching (WM), which identifies permutations that minimize the $L^2$ distance between the weights of two models.

Understanding LMC principles based on permutation symmetries is important not only for comprehending how SGD works in deep learning but also for its application in model merging (Singh & Jaggi, 2020), where two independently trained models are combined. The method of finding permutations using only the $L^2$ distance is particularly versatile, dataset-independent, and computationally efficient. In fact, several studies (Singh & Jaggi, 2020; Wang et al., 2020; Guerrero-Peña et al., 2023) have proposed applications of permutation symmetries in model merging, federated learning, and continual learning.

The current theoretical analysis of LMC relies on the feasibility of closely matching NN weights through permutations. Recently, Zhou et al. (2023) proved that if the distance between the weights of two models can be sufficiently reduced via permutation, then LMC holds. Intuitively, for two SGD solutions $\theta_a$ and $\theta_b$, if $\theta_a \approx \pi(\theta_b)$ holds for a permutation $\pi$, the outputs of the interpolated model will closely approximate those of the original models $\theta_a$ and $\theta_b$.

However, our analysis reveals that even if LMC holds, the permutations found by WM do not significantly reduce the distance between the two models (at most about a 20% reduction). This suggests that LMC is satisfied even when WM does not bring the two models very close (i.e., $\theta_a \not\approx \pi(\theta_b)$). Accordingly, this paper seeks to uncover a more fundamental reason why LMC holds through the permutations found by WM. Specifically, we demonstrate that singular vectors with large singular values of each weight in the models play a crucial role in LMC. Our analysis not only reveals the principle behind WM but also shows that WM may be more advantageous in merging more than two models compared to other methods such as STE.

The contributions of this paper are threefold:

**1. Demonstrating that the $L^2$ distance reduced by WM is not the direct cause of LMC.** We empirically show that permutations found by WM do not significantly reduce the $L^2$ distance between the two models. Our results show that, even when LMC is satisfied, permutations reduce the model weight distance by no more than 20%. Supported by a Taylor approximation, our findings suggest that reducing the $L^2$ distance through permutations is not the direct reason for LMC satisfaction.

**2. Revealing the reason why WM and activation matching (AM) satisfy LMC.** We analyze WM from the perspective of the function of each layer of the model. Specifically, we provide evidence that WM satisfies LMC by aligning the directions of singular vectors with large singular values in each layer's weights. This alignment ensures that the singular vectors with large singular values, which determine the functionality of each layer, become similar between the merged and original models. Additionally, we show that, from the perspective of the input distribution at each hidden layer, aligning singular vectors with large singular values can efficiently approximate the functionality between two models, even if the $L^2$ distance cannot be significantly reduced. As a result, the merged model retains functionality similar to the original models, which facilitates LMC. We also conducted experiments with AM and found that the reason why LMC holds in AM is likely the same as in WM.

**3. Revealing STE is fundamentally different from WM in principle, which leads to a significant difference between them when merging multiple models.** To distinguish WM from other permutation search methods that are independent of $L^2$ distance, we examine the straight-through estimator (STE), which focuses on minimizing the barrier itself rather than the $L^2$ distance. Our experiments reveal that the permutations found by STE do not align the directions of singular vectors, which is a critical difference compared to WM in achieving LMC. Furthermore, we demonstrate experimentally that this difference significantly impacts the satisfaction of LMC among three or more models.

## 2  BACKGROUND AND PRELIMINARIES

### 2.1  NOTATION

For any natural number $k \in \mathbb{N}$, let $[k] = \{1, 2, \ldots, k\}$. Bold uppercase variables represent tensors, including matrices (e.g., $\boldsymbol{X}$), and bold lowercase variables (e.g., $\boldsymbol{x}$) represent vectors. For any tensor $\boldsymbol{X}$, its vectorization is denoted by $\text{vec}(\boldsymbol{X})$, and $\|\boldsymbol{X}\|$ denotes its Frobenius ($L^2$) norm.

## 2.2 PERMUTATION INVARIANCE

We consider multilayer perceptrons (MLPs) $f(\boldsymbol{x}; \boldsymbol{\theta})$ with $L$ layers for simplicity while our analyses in this paper can be applied to any model architectures. Here, $\boldsymbol{x} \in \mathbb{R}^{d_{\mathrm{in}}}$ is the input to the NN, and $\boldsymbol{\theta} \in \mathbb{R}^{d_{\mathrm{param}}}$ represents the model parameters, where $d_{\mathrm{in}} \in \mathbb{N}$ is the dimension of the input, and $d_{\mathrm{param}} \in \mathbb{N}$ is the dimension of the parameters. Let $\boldsymbol{z}_\ell$ be the output of the $\ell$-th layer (i.e., $\boldsymbol{z}_0 = \boldsymbol{x}$, and, for all $\ell \in [L]$, $\boldsymbol{z}_\ell = \sigma(\boldsymbol{W}_\ell \boldsymbol{z}_{\ell-1} + \boldsymbol{b}_\ell)$). Here, $\sigma$ denotes the activation function, and $\boldsymbol{W}_\ell$ and $\boldsymbol{b}_\ell$ represent the weight and bias of the $\ell$-th layer, respectively. Note that in this MLP, we have $\boldsymbol{\theta} = \big\|_{\ell=1}^{L} (\mathrm{vec}(\boldsymbol{W}_\ell) \,\|\, \boldsymbol{b}_\ell)$, where $\|$ represents the concatenation of vectors.

NNs have permutation symmetries of weight space. Considering an NN with model parameters $\boldsymbol{\theta}$, for its $\ell$-th layer, $\boldsymbol{z}_\ell = \boldsymbol{P}^\top \boldsymbol{P} \boldsymbol{z}_\ell = \boldsymbol{P}^\top \sigma(\boldsymbol{P} \boldsymbol{W}_\ell \boldsymbol{z}_{\ell-1} + \boldsymbol{P} \boldsymbol{b}_\ell)$ holds, where $\boldsymbol{P}$ is a permutation matrix. Note that permutation matrices are orthogonal, so we have $\boldsymbol{P}^\top = \boldsymbol{P}^{-1}$. Therefore, by permuting the input of the $(\ell+1)$-st layer with $\boldsymbol{P}^\top$, the model parameters can be changed without altering the input-output relationship of the NN. Specifically, the new weights and bias are given by $\boldsymbol{W}_\ell' = \boldsymbol{P} \boldsymbol{W}_\ell$, $\boldsymbol{b}_\ell' = \boldsymbol{P} \boldsymbol{b}_\ell$, $\boldsymbol{W}_{\ell+1}' = \boldsymbol{W}_{\ell+1} \boldsymbol{P}^\top$. Such permutations can be applied to all layers. We denote the tuple of permutations corresponding to each layer as $\pi = (\boldsymbol{P}_\ell)_{\ell \in [L]}$. Moreover, if a model $\boldsymbol{\theta}$ is given, the application of permutation $\pi$ to $\boldsymbol{\theta}$ is denoted by $\pi(\boldsymbol{\theta})$.

## 2.3 LINEAR MODE CONNECTIVITY (LMC)

Let $\boldsymbol{\theta} \in \mathbb{R}^{d_{\mathrm{param}}}$ be a model and $\mathcal{L}(\boldsymbol{\theta})$ denote the value of the loss function for the model $\boldsymbol{\theta}$. Here, we define the loss barrier between two given models $\boldsymbol{\theta}_a$ and $\boldsymbol{\theta}_b$ as follows:

**Definition 2.1.** For two given models $\boldsymbol{\theta}_a$ and $\boldsymbol{\theta}_b$, their loss barrier is defined as

$$B(\boldsymbol{\theta}_a, \boldsymbol{\theta}_b) := \max_{\lambda \in [0,1]} \big( \mathcal{L}(\lambda \boldsymbol{\theta}_a + (1-\lambda)\boldsymbol{\theta}_b) - (\lambda \mathcal{L}(\boldsymbol{\theta}_a) + (1-\lambda)\mathcal{L}(\boldsymbol{\theta}_b)) \big).$$

Intuitively, the barrier represents the increase in loss due to the linear interpolation of the two models. Two models $\boldsymbol{\theta}_a$ and $\boldsymbol{\theta}_b$ are said to be linearly mode connected if their loss barrier is approximately zero.

## 2.4 PERMUTATION SELECTION

Entezari et al. (2022) conjectured that for SGD solutions $\boldsymbol{\theta}_a$ and $\boldsymbol{\theta}_b$, there exists a permutation $\pi$ such that LMC holds between $\boldsymbol{\theta}_a$ and $\pi(\boldsymbol{\theta}_b)$ with high probability. Afterward, Ainsworth et al. (2023) proposed WM, straight-through estimator (STE), and activation matching (AM) as methods for finding such permutations. This subsection explains WM, which is the main focus of this paper. AM and STE are discussed in Sections 5 and 6.

In WM, we search for a permutation that minimizes the $L^2$ distance between two models[1]:

$$\arg\min_\pi \|\boldsymbol{\theta}_a - \pi(\boldsymbol{\theta}_b)\|^2 = \arg\min_\pi \sum_{\ell \in [L]} \|\boldsymbol{W}_\ell^{(a)} - \boldsymbol{P}_\ell \boldsymbol{W}_\ell^{(b)} \boldsymbol{P}_{\ell-1}^\top\|^2, \tag{1}$$

where, without loss of generality, let $\boldsymbol{P}_L = \boldsymbol{I}$ and $\boldsymbol{P}_0 = \boldsymbol{I}$, and $\boldsymbol{I}$ is an identity matrix. This minimization problem is known as the sum of the bilinear assignments problem, which is NP-hard (Koopmans & Beckmann, 1957; Sahni & Gonzalez, 1976; Ainsworth et al., 2023). Recently, Guerrero-Peña et al. (2023) proposed solving Equation (1) using Sinkhorn's algorithm (Adams & Zemel, 2011) by using an optimal transport problem. We adopt their method because it allows for the optimization of all layers simultaneously, unlike the method by Ainsworth et al. (2023), and potentially finds better solutions.

## 3 MOTIVATING OBSERVATIONS

Previous studies have suggested that the closeness of two parameters in terms of $L^2$ distance is important for satisfying LMC. For example, Zhou et al. (2023) showed that LMC holds if a

---

[1]Although only the weights are considered here, the biases can also be dealt with by concatenating the biases and the weights.

Table 1: Results of WM and the estimated barrier value using Taylor approximation when $\lambda = 1/2$. The table presents the mean and standard deviation from five trials of model merging (i.e., the linear combination of the models $(\theta_a + \pi(\theta_b))/2$). The columns labeled "Barrier", "Taylor approx.", and "Diff." show the barrier value, the estimated barrier value using Equation (2) for the merged model at $\lambda = 1/2$, and their difference, respectively. In the "Diff." column, if a statistical significant difference is determined using a t-test at a 5% significance level, they are highlighted in bold. The table also shows the $L^2$ distance between the models $\theta_a$ and $\theta_b$ before and after applying the permutation, as well as the reduction rate of the $L^2$ distance (i.e., $(\|\theta_a - \theta_b\| - \|\theta_a - \pi(\theta_b)\|)/\|\theta_a - \theta_b\|$).

| Dataset | Network | Barrier ($\lambda = 1/2$) | Taylor approx. | Diff. | $\|\theta_a - \theta_b\|$ | $\|\theta_a - \pi(\theta_b)\|$ | Reduction rate [%] |
|---|---|---|---|---|---|---|---|
| CIFAR10 | VGG11 | $0.035 \pm 0.1$ | $2.956 \pm 0.35$ | $\mathbf{2.921 \pm 0.323}$ | $799.503 \pm 16.396$ | $746.465 \pm 19.576$ | $6.64 \pm 0.808$ |
| | ResNet20 | $0.167 \pm 0.035$ | $7.517 \pm 0.573$ | $\mathbf{7.349 \pm 0.599}$ | $710.762 \pm 16.261$ | $661.055 \pm 12.539$ | $6.987 \pm 0.472$ |
| FMNIST | MLP | $-0.183 \pm 0.049$ | $0.928 \pm 0.175$ | $\mathbf{1.111 \pm 0.152}$ | $121.853 \pm 5.83$ | $100.041 \pm 4.71$ | $17.897 \pm 0.348$ |
| MNIST | MLP | $-0.033 \pm 0.006$ | $0.036 \pm 0.03$ | $\mathbf{0.069 \pm 0.028}$ | $81.231 \pm 5.58$ | $64.751 \pm 4.795$ | $20.305 \pm 1.225$ |

commutativity property is satisfied. This property holds if, for all layers $\ell$, $\boldsymbol{W}_\ell^{(a)} - \boldsymbol{P}_\ell \boldsymbol{W}_\ell^{(b)} \boldsymbol{P}_{\ell-1}^\top = \boldsymbol{0}$. Zhou et al. (2023) argued in Section 5.2 that since WM finds the permutation that minimizes Equation (1), WM can be seen as searching for permutations that satisfy the commutativity property. In particular, there is a huge number of permutations because the total number of possible permutations grows exponentially as the number of layers and the width increase, and thus, some of them may sufficiently reduce the distance between the two models. However, this section explains from the perspective of a Taylor approximation that this intuition is not always correct. Our results demonstrate that even when LMC is satisfied, the permutations found by WM do not necessarily bring the models as close as expected. The facts observed from the experiments in this section motivate us to explore other reasons for satisfying LMC in the following sections.

## 3.1 CLOSENESS OF TWO MODELS IN TERMS OF TAYLOR APPROXIMATION

This subsection describes the estimation of the barrier value using the Taylor approximation. Let $\theta_a$ and $\theta_b$ be two SGD solutions, and $\pi$ be a permutation found by WM to make $\pi(\theta_b)$ close to the model $\theta_a$. Let $\theta_c = \lambda\theta_a + (1 - \lambda)\pi(\theta_b)$ be the merged model at a ratio $\lambda \in (0, 1)$. If $\theta_a$ and $\pi(\theta_b)$ are sufficiently close, then their linear interpolation $\theta_c$ should be close to both models. Therefore, the loss of the parameter $\theta_c$ should be able to be approximated by the Taylor approximation. In fact, the following theorem holds if $\theta_a$ and $\pi(\theta_b)$ are sufficiently close:

**Theorem 3.1.** *The loss function $\mathcal{L} : U \ni \theta \mapsto \mathcal{L}(\theta) \in \mathbb{R}$ is assumed to be of class $C^3$ on an open set $U$ over $\mathbb{R}^{d_{param}}$. Let $\boldsymbol{H}_a$ and $\boldsymbol{H}_b$ be the Hessian matrices centered at the models $\theta_a$ and $\pi(\theta_b)$, respectively. If, for any $\lambda \in (0, 1)$, $\lambda\theta_a + (1 - \lambda)\theta_b \in U$ holds, then we have*

$$B(\theta_a, \pi(\theta_b)) = \max_\lambda \lambda(1-\lambda)\left[\beta\boldsymbol{\mu}^\top \nabla(\mathcal{L}(\theta_a) - \mathcal{L}(\pi(\theta_b))) + \frac{1}{2}\beta^2\boldsymbol{\mu}^\top\left((1 - \lambda)\boldsymbol{H}_a + \lambda\boldsymbol{H}_b\right)\boldsymbol{\mu}\right] + O(\beta^3),$$

(2)

*where $\nabla$ is the gradient with respect to the parameters, $O$ is the Landau symbol, $\beta$ is the $L^2$ distance between $\theta_a$ and $\pi(\theta_b)$, and $\boldsymbol{\mu}$ is the unit vector from $\theta_a$ to $\pi(\theta_b)$ (i.e., $\boldsymbol{\mu} = (\pi(\theta_b) - \theta_a)/\beta$).*

We prove this theorem in Appendix G.1. The theorem states that if $\theta_a$ and $\pi(\theta_b)$ are sufficiently close, the barrier value can be predicted from the gradients and Hessian matrices around each model.

## 3.2 EXPERIMENTAL RESULTS

We conducted experiments to verify whether the Taylor approximation (Equation (2)) accurately estimates the barrier between two SGD solutions. Table 1 presents the experimental results of model merging. Details about the datasets, network training procedures, and permutation search methods used in these experiments are described in Appendix D. In the table, we chose $\lambda = 1/2$ because it is empirically known that the midpoint between two models results in the highest loss (Ainsworth et al., 2023; Guerrero-Peña et al., 2023). It is worth noting that Adilova et al. (2024) demonstrated that the location of the highest barrier shifts when one model is more generalized than the other, although not directly applicable to this paper as the two models are generalized to the same extent. We used the

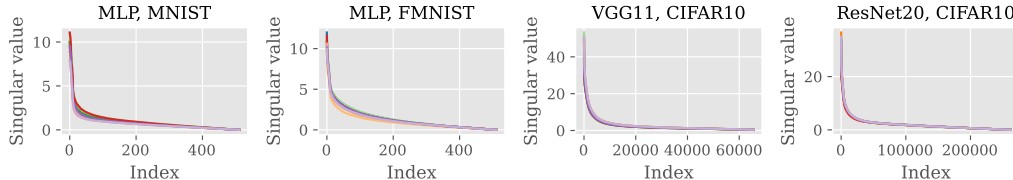

Figure 1: Distribution of the singular values in the second layer. The singular values of ten independently trained models (i.e., trained with different seeds) are plotted in different colors. The distribution of the singular values for all layers is shown in Appendix H.2.

`vhp` function provided in the PyTroch library[2] to efficiently compute $\boldsymbol{\mu}^\top \boldsymbol{H}$, which is required for the evaluation of Equation (2). A negative value in the Barrier column indicates that the loss of the merged model is lower than those of the original (pre-merged) models.

The table shows that for all datasets, there is a significant difference between the actual barrier values and those estimated by the Taylor approximation. These differences are particularly large for VGG11 and ResNet20. Additionally, the table indicates that the $L^2$ distance changes by only about 6% to 20% from the original distance. This suggests that WM does not bring the models sufficiently close, at least not close enough for a second-order Taylor approximation to hold.

## 4 ANALYSIS OF WM

The previous section demonstrates that the establishment of LMC by WM is not due to the reduction in $L^2$ distance itself, but rather because WM helps find permutations that result in a smaller barrier between the two models. To better understand why WM reduces the barrier, we first analyze WM by performing SVD on the weights of each layer of the model in Section 4.1. Then, in Section 4.2, we show that the singular value distribution of each layer is almost identical across independently trained models, and that the primary differences between the models are due to variations in their singular vectors. In Section 4.3, we demonstrate that WM preferentially aligns the directions of singular vectors with large singular values between the weights of the two models. Finally, in Section 4.4, we explain that aligning singular vectors with large singular values makes LMC more achievable because these singular vectors predominantly influence the outputs of the hidden layers of the models.

### 4.1 ANALYSIS BASED ON SVD

The basic idea in analyzing WM is to perform SVD on the weight in each layer. Although using SVD for analysis might seem overly simplistic given that WM reduces the $L^2$ distance, this approach provides important insights that are explored in subsequent sections. In WM, permutation matrices are searched to minimize Equation (1). We denote the SVDs of $\boldsymbol{W}_\ell^{(a)}$ and $\boldsymbol{W}_\ell^{(b)}$ by $\boldsymbol{W}_\ell^{(a)} = \boldsymbol{U}_\ell^{(a)} \boldsymbol{S}_\ell^{(a)} (\boldsymbol{V}_\ell^{(a)})^\top = \sum_i \boldsymbol{u}_{\ell,i}^{(a)} s_{\ell,i}^{(a)} (\boldsymbol{v}_{\ell,i}^{(a)})^\top$ and $\boldsymbol{W}_\ell^{(b)} = \boldsymbol{U}_\ell^{(b)} \boldsymbol{S}_\ell^{(b)} (\boldsymbol{V}_\ell^{(b)})^\top = \sum_j \boldsymbol{u}_{\ell,j}^{(b)} s_{\ell,j}^{(b)} (\boldsymbol{v}_{\ell,j}^{(b)})^\top$, respectively. Here, we assume that the singular values are ordered in descending order (i.e., for all $\ell \in [L]$, $s_{\ell,1}^{(a)} \geq s_{\ell,2}^{(a)} \geq \cdots \geq s_{\ell,n}^{(a)}$ and $s_{\ell,1}^{(b)} \geq s_{\ell,2}^{(b)} \geq \cdots \geq s_{\ell,n}^{(b)}$, where $n$ is the number of singular values). Then, we have

$$\arg\min_\pi \|\boldsymbol{\theta}_a - \pi(\boldsymbol{\theta}_b)\|^2 = \arg\min_\pi \sum_{\ell \in [L]} \left\| \sum_i \boldsymbol{u}_{\ell,i}^{(a)} s_{\ell,i}^{(a)} (\boldsymbol{v}_{\ell,i}^{(a)})^\top - \sum_j \boldsymbol{P}_\ell \boldsymbol{u}_{\ell,j}^{(b)} s_{\ell,j}^{(b)} (\boldsymbol{P}_{\ell-1} \boldsymbol{v}_{\ell,j}^{(b)})^\top \right\|^2. \quad (3)$$

Equation (3) shows that the permutation matrices $\boldsymbol{P}_\ell$ and $\boldsymbol{P}_{\ell-1}$ are multiplied by the left and right singular vectors of the model $\boldsymbol{\theta}_b$, respectively. The $L^2$ distance between the models is expressed by the difference in singular values and singular vectors between the two models, as indicated by Equation (3). Therefore, in the following, we will discuss the differences in (1) singular values and (2) singular vectors of independently trained models.

---

[2] https://pytorch.org/docs/stable/generated/torch.autograd.functional.vhp.html

Figure 2: Mean and standard deviation of $R(\boldsymbol{\theta}_a, \boldsymbol{\theta}_b)$ from five permutation searches using WM. The red and blue bars represent the results with and without applying a permutation to $\boldsymbol{\theta}_b$, respectively.

## 4.2 DIFFERENCES BETWEEN SINGULAR VALUES OF TWO MODELS

First, we investigate the differences between the singular values of two independently trained models. To this end, ten models are trained independently under identical conditions except for the seed, and their singular values are compared. Figure 1 plots the singular values in the second layer of independently trained models in descending order. The evaluation results for all the layers are shown in Figure 7. As can be seen in the figures, in the hidden layers, the singular values are very close across all models. Therefore, the differences in singular values between the models are not a significant obstacle to reducing the distance between the two models to zero.

## 4.3 SINGULAR-VECTOR ALIGNMENT

In the previous subsection, we confirmed that the distributions of the singular values of the weights of the two independently trained models were almost equal. In this subsection, we will show that the permutations found by WM preferentially align the dominant singular vectors of the two models, and cannot align all singular vectors between the two models. Therefore, WM cannot reduce the $L^2$ distance to zero.

First, we introduce the following theorem:

**Theorem 4.1.** *Given the trained $L$-layer MLPs $\boldsymbol{\theta}_a$ and $\boldsymbol{\theta}_b$, Equation (3) is equivalent to*

$$\underset{\pi}{\arg\min} \|\boldsymbol{\theta}_a - \pi(\boldsymbol{\theta}_b)\|^2 = \underset{\pi=(\boldsymbol{P}_\ell)_\ell}{\arg\max} \sum_{\ell,i,j} s_{\ell,i}^{(a)} s_{\ell,j}^{(b)} (\boldsymbol{u}_{\ell,i}^{(a)})^\top (\boldsymbol{P}_\ell \boldsymbol{u}_{\ell,j}^{(b)})(\boldsymbol{v}_{\ell,i}^{(a)})^\top (\boldsymbol{P}_{\ell-1} \boldsymbol{v}_{\ell,j}^{(b)}). \quad (4)$$

The proof of this theorem is shown in Appendix G.2. Focusing on the term for each layer $\sum_{i,j} s_{\ell,i}^{(a)} s_{\ell,j}^{(b)} (\boldsymbol{u}_{\ell,i}^{(a)})^\top (\boldsymbol{P}_\ell \boldsymbol{u}_{\ell,j}^{(b)})(\boldsymbol{v}_{\ell,i}^{(a)})^\top (\boldsymbol{P}_{\ell-1} \boldsymbol{v}_{\ell,j}^{(b)})$ in Equation (4), $(\boldsymbol{u}_{\ell,i}^{(a)})^\top (\boldsymbol{P}_\ell \boldsymbol{u}_{\ell,j}^{(b)})$ is the inner product between the left singular vector $\boldsymbol{u}_{\ell,i}^{(a)}$ of the model $\boldsymbol{\theta}_a$ and the left singular vector $\boldsymbol{u}_{\ell,j}^{(b)}$ of the model $\boldsymbol{\theta}_b$, applied with the permutation matrix $\boldsymbol{P}_\ell$. The permutation matrix is orthogonal, so it only permutes the elements without changing the norms of the left singular vectors. Therefore, this inner product is maximized when the directions of the two left singular vectors are aligned by the permutation matrix. The same applies to the right singular vectors. Thus, Equation (4) can be interpreted as finding permutation matrices that align the directions of the singular vectors for all layers between two models, especially those associated with large singular values. Like MLPs, Appendix F shows a similar analysis holds for convolutional layers.

Then, to empirically evaluate how well the singular vectors are aligned, we calculate

$$R(\boldsymbol{\theta}_a, \pi(\boldsymbol{\theta}_b)) = \frac{\sum_{\ell,i,j} (\boldsymbol{u}_{\ell,i}^{(a)})^\top \boldsymbol{P}_\ell \boldsymbol{u}_{\ell,j}^{(b)} (\boldsymbol{v}_{\ell,i}^{(a)})^\top \boldsymbol{P}_{\ell-1} \boldsymbol{v}_{\ell,j}^{(b)}}{\sum_\ell n_\ell},$$

where $n_\ell$ is the number of singular values in the $\ell$-th layer. Note that $|R(\boldsymbol{\theta}_a, \pi(\boldsymbol{\theta}_b))| \leq 1$ holds and, we have equality if for all $\ell$ and $i$, $\boldsymbol{u}_{\ell,i}^{(a)} = \boldsymbol{P}_\ell \boldsymbol{u}_{\ell,i}^{(b)}$ and $\boldsymbol{v}_{\ell,i}^{(a)} = \boldsymbol{P}_{\ell-1} \boldsymbol{v}_{\ell,i}^{(b)}$ (proof in Appendix G.5). Therefore, if $R(\boldsymbol{\theta}_a, \pi(\boldsymbol{\theta}_b))$ is close to one, the singular vectors of the models are well-aligned.

Figure 2 shows the experimental results of evaluating the $R$ value between $\boldsymbol{\theta}_a$ and $\pi(\boldsymbol{\theta}_b)$. A threshold $\gamma$ is introduced to examine whether the singular vectors with large singular values are preferentially aligned. For each model, we evaluate $R$ using only singular vectors whose ratio to the largest singular value is greater than $\gamma$. Thus, in the figure, $\gamma = 0$ corresponds to the results when all singular vectors

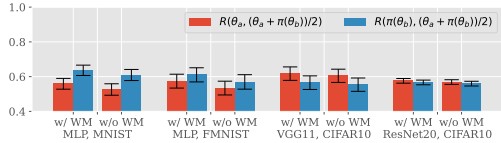 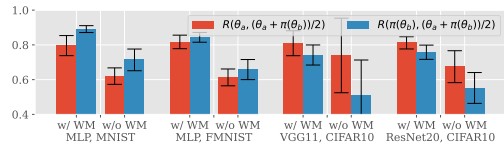

(a) Evaluation results with the threshold $\gamma = 0$.      (b) Evaluation results with the threshold $\gamma = 0.3$.

Figure 3: Evaluation results of $R$ between the pre- and post-merged models. The red and blue bars represent the evaluation results of $R(\boldsymbol{\theta}_a, (\boldsymbol{\theta}_a + \pi(\boldsymbol{\theta}_b))/2)$ and $R(\pi(\boldsymbol{\theta}_b), (\boldsymbol{\theta}_a + \pi(\boldsymbol{\theta}_b))/2)$, respectively.

are used, and $\gamma = 0.3$ corresponds to the results when only singular vectors with a ratio to the largest singular value exceeding 0.3 are used. When we calculated $R$, its denominator $\sum_\ell n_\ell$ was also adjusted according to the value of $\gamma$. The details of the calculation of $R$ are described in Appendix B.

The figure shows that the directions of the singular vectors are aligned with WM. Without WM, the value of $R$ is almost zero, indicating that the singular vectors are nearly orthogonal. Additionally, focusing on the difference in $\gamma$, when the singular vectors are aligned using WM, the value of $R$ is clearly larger when $\gamma$ is 0.3. This indicates that WM aligns singular vectors with larger singular values more closely. Although the value of $R$ is not necessarily very large, especially around 0.2 at most for VGG11 and ResNet20, this alignment of singular vectors still affects the merged models.

Figure 3 shows the evaluation results of $R$ between the merged model (i.e., $(\boldsymbol{\theta}_a + \pi(\boldsymbol{\theta}_b))/2$) and the pre-merged models (i.e., $\boldsymbol{\theta}_a$ and $\pi(\boldsymbol{\theta}_b)$). To investigate how well the directions of singular vectors with large values are aligned between the merged and pre-merged models, we also show the results for $\gamma = 0.3$ in Figure 3(b). The figures show that when $\gamma = 0$, the value of $R$ does not change regardless of the use of WM. However, when $\gamma = 0.3$, the value of $R$ changes significantly depending on whether WM is used. For example, the MLP results show that the value of $R$ exceeds 0.8 when using WM. This result indicates that the directions of singular vectors with particularly large singular values are better aligned between these models.

### 4.4 IMPORTANCE OF SINGULAR VECTORS IN LMC

In the previous section, we mentioned that WM aligns the directions of singular vectors with large singular values. This section clarifies why these singular vectors, rather than $L^2$ distances, play a crucial role in establishing LMC.

To explain this, we first focus on the difference between outputs at the $\ell$-th layers of two models, $\boldsymbol{W}_\ell^{(a)}$ and $\boldsymbol{W}_\ell^{(b)}$, given the same input $\boldsymbol{z}$ (e.g., $\boldsymbol{z} = \boldsymbol{z}_{\ell-1}^{(a)}$ or $\boldsymbol{z} = \boldsymbol{z}_{\ell-1}^{(b)}$). Suppose that the distributions of the singular values of the two weights are equal (this assumption holds for models trained with SGD, as shown in Figure 7). The difference between the outputs can be bounded from above:

$$\mathbb{E}\|\sigma(\boldsymbol{W}_\ell^{(a)}\boldsymbol{z}) - \sigma(\boldsymbol{W}_\ell^{(b)}\boldsymbol{z})\| \leq C\mathbb{E}\|\boldsymbol{W}_\ell^{(a)}\boldsymbol{z} - \boldsymbol{W}_\ell^{(b)}\boldsymbol{z}\|, \tag{5}$$

where $\sigma$ is a Lipschitz continuous activation function with a constant $C > 0$ (e.g., $C = 1$ for the ReLU function). From Equation (5), we can see that depending on the distribution of the input $\boldsymbol{z}$, the outputs of the two layers can be close even when the distance between the two weights is not.

Let $\boldsymbol{W}_\ell^{(a)} = \sum_i \boldsymbol{u}_{\ell,i}^{(a)} s_i^{(a)} (\boldsymbol{v}_{\ell,i}^{(a)})^\top$ and $\boldsymbol{W}_\ell^{(b)} = \sum_i \boldsymbol{u}_{\ell,i}^{(b)} s_i^{(b)} (\boldsymbol{v}_{\ell,i}^{(b)})^\top$ be the SVDs of their weights. Here, we assume that, for some index $k$, the direction of $\boldsymbol{z}$ is always in the direction of the $k$-th right singular vector $\boldsymbol{v}_{\ell,k}^{(a)}$ with $\boldsymbol{W}_\ell^{(a)}$. Then, the product between $\boldsymbol{W}_\ell^{(a)}$ and $\boldsymbol{z}$ is given by $\boldsymbol{W}_\ell^{(a)}\boldsymbol{z} = \sum_i \boldsymbol{u}_{\ell,i}^{(a)} s_i^{(a)} (\boldsymbol{v}_{\ell,i}^{(a)})^\top \boldsymbol{z} = \boldsymbol{u}_{\ell,k}^{(a)} s_k^{(a)} (\boldsymbol{v}_{\ell,k}^{(a)})^\top \boldsymbol{z}$ because the singular vectors are orthogonal. Therefore, Equation (5) can be rewritten as

$$\mathbb{E}\|\sigma(\boldsymbol{W}_\ell^{(a)}\boldsymbol{z}) - \sigma(\boldsymbol{W}_\ell^{(b)}\boldsymbol{z})\| \leq C\mathbb{E}\left\|\boldsymbol{u}_{\ell,k}^{(a)} s_k^{(a)} \boldsymbol{v}_{\ell,k}^{(a)} \boldsymbol{z} - \sum_i \boldsymbol{u}_{\ell,i}^{(b)} s_i^{(b)} \boldsymbol{v}_{\ell,i}^{(b)} \boldsymbol{z}\right\|.$$

Thus, as long as the directions of the $k$-th singular vectors of the two weights are aligned (i.e., $\boldsymbol{v}_{\ell,k}^{(a)} = \boldsymbol{v}_{\ell,k}^{(b)}$ and $\boldsymbol{u}_{\ell,k}^{(a)} = \boldsymbol{u}_{\ell,k}^{(b)}$ hold), the outputs of the two layers will coincide regardless of the other

Figure 4: Average absolute values of the inner products of the right singular vectors and the input of the second layer. The figure shows results for ten models trained with different seeds, each represented by a different color. The test dataset is used as input for the models. In each plot, the vertical axis denotes the value of $\mathbb{E}(\boldsymbol{v}_{\ell,i}^\top \boldsymbol{z}_{\ell-1})^2$, and the horizontal axis denotes the index $i$ of the right singular vector. The left side of each plot corresponds to singular vectors with large singular values. The results for all layers are shown in Figure 9.

singular vectors. Note that the $L^2$ distance between the two weights is not necessarily close to zero since the directions of the other singular vectors need not be aligned. In fact, in Appendix C, we provide an example where the $L^2$ distance is not close to zero even though the output is zero.

More generally, the following theorem holds for the difference between the outputs of the two layers.

**Theorem 4.2.** *For the difference, we have*

$$\mathbb{E}\|\sigma(\boldsymbol{W}_\ell^{(a)}\boldsymbol{z}) - \sigma(\boldsymbol{W}_\ell^{(b)}\boldsymbol{z})\| \le C\Bigg(\sum_i (s_{\ell,i}^{(a)})^2 \mathbb{E}((\boldsymbol{v}_{\ell,i}^{(a)})^\top \boldsymbol{z})^2 + \sum_i (s_{\ell,i}^{(b)})^2 \mathbb{E}((\boldsymbol{v}_{\ell,i}^{(b)})^\top \boldsymbol{z})^2$$

$$- 2\sum_{i,j} s_{\ell,i}^{(a)} s_{\ell,j}^{(b)} (\boldsymbol{u}_{\ell,i}^{(a)})^\top \boldsymbol{u}_{\ell,j}^{(b)} \mathbb{E}(\boldsymbol{v}_{\ell,i}^{(a)})^\top \boldsymbol{z}(\boldsymbol{v}_{\ell,j}^{(b)})^\top \boldsymbol{z}\Bigg)^{1/2}. \quad (6)$$

The proof of Theorem 4.2 is provided in Appendix G.6. If the right-hand side of Equation (6) in the theorem is small, the difference is also small. Note that each sum on the right-hand side includes the inner product between the right singular vector and the input (i.e., $(\boldsymbol{v}_{\ell,i}^{(a)})^\top \boldsymbol{z}$ and $(\boldsymbol{v}_{\ell,i}^{(b)})^\top \boldsymbol{z}$). Since singular vectors are orthogonal to each other, if $\boldsymbol{z}$ is aligned with one singular vector, its inner product with the other singular vectors will be small. In other words, right singular vectors with a large inner product with the input determine the difference between the outputs of the two layers.

In the context of WM, it is desirable that the input vector has the same direction as the right singular vectors with large singular values because WM preferentially aligns these singular vectors of the two weights. To verify this, we experimentally investigate the relationship between the directions of the right singular vectors and that of the input vector. Figure 4 shows the value of $\mathbb{E}(\boldsymbol{v}_{\ell,i}^\top \boldsymbol{z}_{\ell-1})^2$ for the $i$-th right singular vector $\boldsymbol{v}_{\ell,i}$ in the second layer (i.e., $\ell = 2$) and the corresponding hidden layer input $\boldsymbol{z}_{\ell-1}$ for each model. The results show that in the hidden layer, the singular vectors with large singular values have large inner products with the input vectors, which indicates that the permutations found by WM make LMC more feasible.[3] In particular, Figure 3(b) shows that by aligning the directions of the singular vectors between the two models $\boldsymbol{\theta_a}$ and $\boldsymbol{\theta_b}$ using WM, the directions of the singular vectors with large singular values of the models before and after merging (e.g., $\boldsymbol{\theta_a}$ and $(\boldsymbol{\theta_a} + \boldsymbol{\theta_b})/2$) are well aligned. This suggests that the hidden layer outputs of the models before and after merging are closer, which contributes to the establishment of LMC.

Some studies (Ainsworth et al., 2023; Entezari et al., 2022) have observed that increasing the model width makes it easier to satisfy LMC using WM. In Appendix H.4, we provide an empirical analysis to explain this observation in terms of singular-vector alignment. Furthermore, Qu & Horvath (2024) showed that strengthening weight decay and increasing the learning rate makes it easier for LMC to be established through WM. Appendix H.5 demonstrates empirically that increasing these values reduces the proportion of large singular values in the weights of each layer, facilitating the alignment of the corresponding singular vectors through WM, and thus making LMC easier to achieve.

---

[3]As shown in Figures 11 and 12, this tendency only occurs when the model is sufficiently wide. This suggests that LMC is unlikely to be established in WM unless the model is wide enough.

Table 2: Results of model merging with STE.

| Dataset | Network | Barrier ($\lambda = 1/2$) | $L^2$ dist. w/o STE | $L^2$ dist. w/ STE | $R(\theta_a, \pi(\theta_b))$ ($\gamma = 0.3$) |
|---------|---------|---------------------------|---------------------|--------------------|-----------------------------------------------|
| CIFAR10 | VGG11 | $0.06 \pm 0.042$ | $799.503 \pm 16.396$ | $799.779 \pm 16.177$ | $0.036 \pm 0.007$ |
| | ResNet20 | $0.119 \pm 0.119$ | $710.762 \pm 16.261$ | $711.142 \pm 16.048$ | $0.013 \pm 0.005$ |
| FMNIST | MLP | $-0.342 \pm 0.066$ | $121.853 \pm 5.83$ | $118.316 \pm 5.453$ | $0.081 \pm 0.008$ |
| MNIST | MLP | $-0.037 \pm 0.008$ | $81.231 \pm 5.58$ | $73.994 \pm 5.58$ | $0.211 \pm 0.013$ |

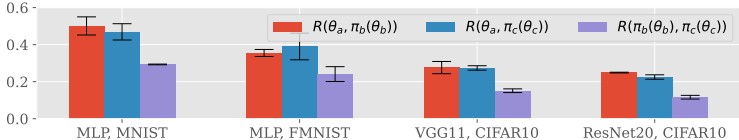

Figure 5: Evaluation results of $R$ between each pair of the models with $\gamma = 0.3$.

## 5 ACTIVATION MATCHING

Ainsworth et al. (2023) proposed activation matching (AM) as a permutation search method different from WM. This section compares AM and WM, and explains that their results are almost similar.

AM searches for a permutation $\pi^*$ based on the following equation:

$$\pi^* = \underset{\pi = (P_\ell)_\ell}{\arg\min} \sum_{\ell \in [L]} \mathbb{E} \| z_\ell^{(a)} - P_\ell z_\ell^{(b)} \|^2. \tag{7}$$

Unlike WM, AM can be solved as a simple linear sum assignment problem because it can be optimized independently for each layer, allowing for the optimal solution to be obtained.

The minimization of Equation (7) is related to Theorem 4.2. Specifically, in a permutation search for the $\ell$-th layer, if we assume that the outputs $z_{\ell-1}^{(a)}$ and $z_{\ell-1}^{(b)}$ from the previous layer are sufficiently close under the permutation $P_{\ell-1}$ (i.e., $z_{\ell-1}^{(a)} \approx P_{\ell-1} z_{\ell-1}^{(b)}$), then minimizing the right-hand side of Equation (6) becomes equivalent to reducing the objective function in Equation (7). In other words, similar to WM, AM may search for permutations that align the singular vectors with large singular values between two models. To verify this, the results of model merging using AM are presented in Table 4. The experimental settings are the same as those used for WM in Section 3.2. Additionally, to evaluate how well the singular vectors align through permutation, the $R$ calculation results are shown in Figures 15 and 16. These results closely resemble those for WM in Table 1 and Figures 2 and 3, suggesting that the reason AM achieves LMC is likely to be similar to that for WM.

## 6 COMPARISON WITH STRAIGHT-THROUGH ESTIMATOR

This section discusses the relationship between the straight-through estimator (STE), a more direct permutation search method, and WM in terms of singular vectors. STE uses a dataset to find permutations with a small barrier value. We also explain that STE and WM are based on fundamentally different principles and show how this difference impacts LMC among three or more models.

### 6.1 STRAIGHT-THROUGH ESTIMATOR (STE)

Ainsworth et al. (2023) proposed the STE, which finds a permutation $\pi$ such that

$$\underset{\pi}{\arg\min} \mathcal{L}\big( (\theta_a + \pi(\theta_b)) / 2 \big). \tag{8}$$

Since Equation (8) is difficult to solve directly, Ainsworth et al. (2023) proposed a method to approximate the solution. Later, Guerrero-Peña et al. (2023) proposed a method to solve Equation (8) directly using Sinkhorn's algorithm. We adopt the latter method, which we refer to as STE in this paper.

### 6.2 EXPERIMENTAL RESULTS OF MODEL MERGING BY STE

The experimental results of model merging using STE are shown in Table 2. This table also shows the $L^2$ distance and the $R$ value between the two models before and after the permutation. Despite

Table 3: Loss and accuracy barriers between $\pi_b(\boldsymbol{\theta}_b)$ and $\pi_c(\boldsymbol{\theta}_c)$. The table shows the mean and standard deviation over three model merging trials.

| | | Loss barrier $((\pi_b(\boldsymbol{\theta}_b) + \pi_c(\boldsymbol{\theta}_c))/2)$ | | Accuracy barrier $((\pi_b(\boldsymbol{\theta}_b) + \pi_c(\boldsymbol{\theta}_c))/2)$ | |
| Dataset | Network | WM | STE | WM | STE |
| --- | --- | --- | --- | --- | --- |
| CIFAR10 | VGG11 | $0.141 \pm 0.141$ | $2.172 \pm 0.989$ | $10.12 \pm 5.117$ | $32.013 \pm 8.193$ |
| | ResNet20 | $0.294 \pm 0.098$ | $1.693 \pm 0.168$ | $7.23 \pm 0.99$ | $34.483 \pm 2.426$ |
| FMNIST | MLP | $-0.174 \pm 0.051$ | $0.023 \pm 0.118$ | $4.337 \pm 1.434$ | $15.97 \pm 1.724$ |
| MNIST | MLP | $-0.031 \pm 0.003$ | $0.017 \pm 0.014$ | $0.475 \pm 0.069$ | $2.312 \pm 0.457$ |

the relatively small barrier value, Table 2 shows that the $L^2$ distance between the two models before and after permutation hardly changes compared to the results with WM shown in Table 1. Since $R(\boldsymbol{\theta}_a, \pi(\boldsymbol{\theta}_b))$ is nearly zero, the singular vectors between the two models are likely not aligned at all. Therefore, the reason for satisfying LMC by STE is completely different from that of WM.

### 6.3 LMC among Three Models

The previous subsection shows that the permutation matrices found by STE do not align the directions of the singular vectors of the models. This suggests that STE finds a permutation that reduces the loss of the merged model based on the loss landscape rather than the linear algebraic properties of the weight matrices of each layer. The difference between the principles of STE and WM could result in a qualitative difference in LMC among three or more models.

Suppose we have three SGD solutions: $\boldsymbol{\theta}_a$, $\boldsymbol{\theta}_b$, and $\boldsymbol{\theta}_c$. Let $\pi_b$ and $\pi_c$ be permutations that satisfy LMC between $\boldsymbol{\theta}_a$ and $\pi_b(\boldsymbol{\theta}_b)$, and $\boldsymbol{\theta}_a$ and $\pi_c(\boldsymbol{\theta}_c)$, respectively. If permutations found by STE depend on the locality of the loss landscape rather than the linear algebraic properties of the model weights, there is no guarantee that $\pi_b(\boldsymbol{\theta}_b)$ and $\pi_c(\boldsymbol{\theta}_c)$ are linearly mode-connected. In contrast, permutations found by WM align the directions of the singular vectors of the two models. This means that the singular vectors of $\pi_b(\boldsymbol{\theta}_b)$ and $\pi_c(\boldsymbol{\theta}_c)$ are also expected to be aligned. Thus, the LMC between $\pi_b(\boldsymbol{\theta}_b)$ and $\pi_c(\boldsymbol{\theta}_c)$ may not be satisfied with STE, while it is likely to be satisfied with WM.

We performed model merging experiments among three models to confirm the validity of the above discussion. First, Figure 5 presents the results of examining how well the singular vectors are aligned in each model pair by WM. Since the models $\boldsymbol{\theta}_b$ and $\boldsymbol{\theta}_c$ are matched to $\boldsymbol{\theta}_a$ through WM, it is expected that $R(\boldsymbol{\theta}_a, \pi_b(\boldsymbol{\theta}_b))$ and $R(\boldsymbol{\theta}_a, \pi_c(\boldsymbol{\theta}_c))$ would be large. On the other hand, although $\boldsymbol{\theta}_b$ and $\boldsymbol{\theta}_c$ were not explicitly aligned, $R(\pi_b(\boldsymbol{\theta}_b), \pi_c(\boldsymbol{\theta}_c))$ is clearly greater than zero, indicating that the directions of these two singular vectors are indirectly aligned by WM. From this result, the barrier between the models $\pi_b(\boldsymbol{\theta}_b)$ and $\pi_c(\boldsymbol{\theta}_c)$ is expected to be small. To confirm this, Table 3 shows the barriers between $\pi_b(\boldsymbol{\theta}_b)$ and $\pi_c(\boldsymbol{\theta}_c)$. Table 5 shows the detailed results, and Figure 17 shows the test accuracy landscape around $\boldsymbol{\theta}_a$, $\pi_b(\boldsymbol{\theta}_b)$, and $\pi_c(\boldsymbol{\theta}_c)$. As can be seen from Table 3, the barrier between $\pi_b(\boldsymbol{\theta}_b)$ and $\pi_c(\boldsymbol{\theta}_c)$ is smaller with WM than with STE. This means that there is a significant difference between the principles of permutations obtained by WM and STE. Figure 17 also shows that the landscape of test accuracy is flatter around the three models with WM than with STE. Therefore, WM is likely to be more advantageous, especially for merging three or more models.

## 7 Conclusion

This paper analyzed why linear mode connectivity (LMC) is satisfied through permutation search with weight matching (WM). First, we demonstrated that WM does not reduce the distance between the weights of two models as significantly as previously thought. We then analyzed WM using singular value decomposition (SVD) and found that WM aligns the directions of singular vectors with large singular values, which plays a crucial role in achieving LMC. Additionally, we showed that the reason LMC is established in AM is likely the same as for WM. Finally, we discussed the difference between STE and WM from the perspective of singular vectors.

Although this paper primarily analyzed WM from the perspective of individual layers (e.g., Theorem 4.2), it remains unclear why our analysis can explain the phenomenon so effectively since the actual network is multi-layered. In the future, a more comprehensive analysis that accounts for the multi-layered structure of the network will be necessary.

## ETHICS STATEMENT

This paper presents work whose goal is to advance the field of Machine Learning. There are many potential ethical consequences of our work, none which we feel must be specifically highlighted here.

## REPRODUCIBILITY

The settings for reproducing the experiments are described in Appendix D. All the proofs of the theorems are given in Appendix G.

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

APPENDIX

## A  EXTENDED RELATED WORK

**(Linear) mode connectivity.**    Several studies (Garipov et al., 2018; Draxler et al., 2018; Freeman & Bruna, 2017) have found that different neural network solutions can be connected by nonlinear paths with almost no increase in loss. Nagarajan & Kolter (2019) first discovered that solutions can be connected by linear paths with an almost constant loss value when training models on MNIST with the same random initial values. Later, Frankle et al. (2020) demonstrated experimentally that LMC is not always satisfied between two SGD solutions, even with the same initial parameters, depending on the datasets and model architectures. However, they also showed that if a single model is trained for a certain period and then two models are trained independently from this pre-trained model as a starting point, they are linearly mode-connected. Furthermore, Frankle et al. (2020) explored the relationship between LMC and the lottery-ticket hypothesis (Frankle & Carbin, 2019). Entezari et al. (2022) conjectured that LMC is satisfied with a high probability between two SGD solutions by accounting for permutation symmetries in the hidden layers. Subsequently, Ainsworth et al. (2023) proposed a WM method by formulating neuron alignment as a bipartite graph matching problem and solving it approximately. Later, Guerrero-Peña et al. (2023) suggested using Sinkhorn's algorithm to solve the WM directly. Some previous studies (Ainsworth et al., 2023; Crisostomi et al., 2024) have also proposed permutation search methods for achieving LMC between multiple models. However, all of these methods reduce the $L^2$ distances between the models, and no methods have been proposed that use information on the loss function, such as STE. Therefore, in this paper, we created a pair of models and performed a permutation search for each pair to clarify the differences between WM and STE in a fair manner. The investigation of permutation search methods for multiple models is a future work.

While several papers (Venturi et al., 2019; Nguyen et al., 2019; Nguyen, 2019; Kuditipudi et al., 2019) have discussed nonlinear mode connectivity, there is little theoretical analysis on LMC. Ferbach et al. (2024) provided an upper bound on the minimal width of the hidden layer to satisfy LMC. However, to prove this, they assumed the independence of all neuron's weight vectors inside a given layer. It is unlikely that this assumption holds for models after training. Partially similar to our paper, Singh et al. (2024) demonstrated that the barrier value can be approximated using a second-order Taylor approximation for the case of spawning (Zhou et al., 2023). However, they have not validated this approach for permutations, and we revealed that a second-order Taylor approximation fails to accurately estimate the barrier value in the case of permutations. Zhou et al. (2023) introduced the concept of layerwise linear feature connectivity (LLFC) and showed that LLFC implies LMC. Additionally, Zhou et al. (2023) demonstrated that if weak additivity for ReLU activation and the commutativity property are satisfied, then both LLFC and LMC are satisfied. However, we show that the $L^2$ distance between the models after permutation is not close enough to satisfy the commutativity property. This motivated us to investigate the relationship between LMC and WM.

**Model merging.**    Relevant topics of LMC include model merging and federated learning. McMahan et al. (2017) and Konečný et al. (2016) introduced the concept of federated learning, where a model is trained on divided datasets. Wang et al. (2020) proposed a federated learning method by permuting each component unit and then averaging the weights of the models. Singh & Jaggi (2020) proposed a method for merging models by performing alignments of model weights using optimal transport, which is similar to the method proposed by Ainsworth et al. (2023). Although their method is designed for model fusion and its performance is inferior to that of Ainsworth et al.'s method, it can be considered an LMC-based method because it uses hard alignments for the same architecture. Wortsman et al. (2022) proposed a method to improve test accuracy without increasing inference cost, unlike ensemble methods, by averaging the weights of models fine-tuned with different hyperparameters.

**Low-rank bias.**    Empirically, some previous studies (Tukan et al., 2020; Arora et al., 2018; Alvarez & Salzmann, 2017; Yu et al., 2017; Denton et al., 2014) on the compression of trained DNN models have pointed out that even when the weights are replaced with low-rank matrices, the accuracy does not decrease significantly. This suggests that SGD has an implicit bias toward reducing the model weights to low-rank. Yunis et al. (2024) explored the reduction of rank during SGD-based training and the orientation of top singular vectors via SVD on the weights. While this is relevant to our

paper, Yunis et al. (2024) do not examine the effects of permutations in detail. Moreover, it does not discuss the interplay between hidden layer inputs and top singular vectors under permutations. Galanti et al. (2022) and Timor et al. (2023) further discuss the low-rank effect on the weights of trained models introduced by weight decay and a small initialization scale. In particular, Galanti et al. (2022) state that the weights become more low-rank by strengthening the weight decay and increasing the learning rate, and this is expected to help establish LMC via WM (which we experimentally confirm in Appendix H.5).

## B  CALCULATION OF $R$ WITH THRESHOLD $\gamma$

This section describes how to calculate the $R$ value with a threshold $\gamma > 0$. Given two models, $\boldsymbol{\theta}_a$ and $\boldsymbol{\theta}_b$, let $\boldsymbol{W}_\ell^{(a)}$ and $\boldsymbol{W}_\ell^{(b)}$ be the weights of the $\ell$-th layers of the models $\boldsymbol{\theta}_a$ and $\boldsymbol{\theta}_b$, respectively. Let $\sum_i \boldsymbol{u}_{\ell,i}^{(a)} s_{\ell,i}^{(a)} (\boldsymbol{v}_{\ell,i}^{(a)})^\top$ and $\sum_i \boldsymbol{u}_{\ell,i}^{(b)} s_{\ell,i}^{(b)} (\boldsymbol{v}_{\ell,i}^{(b)})^\top$ be the SVDs of these weights. Also, let $s^{(a)}$ and $s^{(b)}$ be the maximum singular values in all the layers of the models $\boldsymbol{\theta}_a$ and $\boldsymbol{\theta}_b$, respectively. The $R$ value with the threshold $\gamma$ is calculated as follows:

$$R_\gamma(\boldsymbol{\theta}_a, \boldsymbol{\theta}_b) = \frac{\sum_{\ell,i,j} I[(s_{\ell,i}^{(a)} \geq \gamma s^{(a)}) \wedge (s_{\ell,i}^{(b)} \geq \gamma s^{(b)})](\boldsymbol{u}_{\ell,i}^{(a)})^\top (\boldsymbol{u}_{\ell,j}^{(b)})(\boldsymbol{v}_{\ell,i}^{(a)})^\top (\boldsymbol{v}_{\ell,j}^{(b)})}{\sum_\ell \min\{n_\ell^{(a)}, n_\ell^{(b)}\}},$$

where $n_\ell^{(a)}$ and $n_\ell^{(b)}$ are the numbers of singular values greater than $\gamma s^{(a)}$ and $\gamma s^{(b)}$, respectively, and $I$ is an indicator function that returns one if the given logical expression is true and zero if it is false.

Finally, we will briefly explain that $|R_\gamma(\boldsymbol{\theta}_a, \boldsymbol{\theta}_b)| \leq 1$ holds. We first define the new weight matrices by $\boldsymbol{W}_\ell^{\prime(a)} = \sum_i \boldsymbol{u}_{\ell,i}^{(a)} I[(s_{\ell,i}^{(a)} \geq \gamma s^{(a)})](\boldsymbol{v}_{\ell,i}^{(a)})^\top$ and $\boldsymbol{W}_\ell^{\prime(b)} = \sum_i \boldsymbol{u}_{\ell,i}^{(b)} I[(s_{\ell,j}^{(b)} \geq \gamma s^{(b)})](\boldsymbol{v}_{\ell,i}^{(b)})^\top$. From the definition, we have:

$$\mathrm{tr}\left((\boldsymbol{W}_\ell^{\prime(a)})^\top \boldsymbol{W}_\ell^{\prime(b)}\right) = \sum_{i,j} I[(s_{\ell,i}^{(a)} \geq \gamma s^{(a)}) \wedge (s_{\ell,j}^{(b)} \geq \gamma s^{(b)})](\boldsymbol{u}_{\ell,i}^{(a)})^\top (\boldsymbol{u}_{\ell,j}^{(b)})(\boldsymbol{v}_{\ell,i}^{(a)})^\top (\boldsymbol{v}_{\ell,j}^{(b)}). \quad (9)$$

Note that the $i$-th singular values of these weights $\boldsymbol{W}_\ell^{\prime(a)}$ and $\boldsymbol{W}_\ell^{\prime(b)}$ can be regarded as $I[s_{\ell,i}^{(a)} \geq \gamma s^{(a)}]$ and $I[s_{\ell,i}^{(b)} \geq \gamma s^{(b)}]$, respectively. Therefore, von Neumann's trace inequality (von Neumann, 1962) yields that:

$$\mathrm{tr}\left((\boldsymbol{W}_\ell^{\prime(a)})^\top \boldsymbol{W}_\ell^{\prime(b)}\right) \leq \sum_i I[s_{\ell,i}^{(a)} \geq \gamma s^{(a)}] I[s_{\ell,i}^{(b)} \geq \gamma s^{(b)}] = \min\{n_\ell^{(a)}, n_\ell^{(b)}\}. \quad (10)$$

From Equations (9) and (10), we have:

$$\sum_{i,j} I[(s_{\ell,i}^{(a)} \geq \gamma s^{(a)}) \wedge (s_{\ell,j}^{(b)} \geq \gamma s^{(b)})](\boldsymbol{u}_{\ell,i}^{(a)})^\top (\boldsymbol{u}_{\ell,j}^{(b)})(\boldsymbol{v}_{\ell,i}^{(a)})^\top (\boldsymbol{v}_{\ell,j}^{(b)}) \leq \min\{n_\ell^{(a)}, n_\ell^{(b)}\}.$$

By summing both sides for $\ell$, we get $|R_\gamma(\boldsymbol{\theta}_a, \boldsymbol{\theta}_b)| \leq 1$.

## C  SIMPLE EXAMPLE OF THEOREM 4.2

In Section 4.4, we explained that even when the $L^2$ distance between the weights of two models is large, their outputs can be close depending on the input distribution. Here, we use a simple example for a more detailed analysis.

Consider the weights of two models, $\boldsymbol{W}^{(a)}$ and $\boldsymbol{W}^{(b)}$, given by:

$$\boldsymbol{W}^{(a)} = \begin{pmatrix} -0.398 & -0.003 & 0.210 \\ 1.059 & 0.303 & 0.521 \\ 0.609 & -0.785 & -0.235 \end{pmatrix}, \boldsymbol{W}^{(b)} = \begin{pmatrix} -0.255 & -0.319 & -0.559 \\ 1.031 & -0.155 & 0.484 \\ 0.742 & -0.114 & -0.604 \end{pmatrix}.$$

The SVDs of these matrices are represented by:

$$\boldsymbol{W}^{(a)} = \boldsymbol{U}^{(a)} \boldsymbol{S}^{(a)} (\boldsymbol{V}^{(a)})^\top$$

$$\approx \begin{pmatrix} -0.260 & -0.127 & 0.957 \\ 0.850 & -0.501 & 0.165 \\ 0.458 & 0.856 & 0.238 \end{pmatrix} \begin{pmatrix} 1.317 & 0 & 0 \\ 0 & 0.959 & 0 \\ 0 & 0 & 0.277 \end{pmatrix} \begin{pmatrix} 0.974 & -0.077 & 0.213 \\ 0.043 & -0.859 & -0.510 \\ -0.222 & -0.506 & 0.833 \end{pmatrix}^\top,$$

and

$$\boldsymbol{W}^{(b)} = \boldsymbol{U}^{(b)} \boldsymbol{S}^{(b)} (\boldsymbol{V}^{(b)})^\top$$

$$\approx \begin{pmatrix} -0.261 & 0.587 & 0.767 \\ 0.850 & -0.237 & 0.470 \\ 0.458 & 0.774 & -0.437 \end{pmatrix} \begin{pmatrix} 1.317 & 0 & 0 \\ 0 & 0.958 & 0 \\ 0 & 0 & 0.277 \end{pmatrix} \begin{pmatrix} 0.974 & -0.077 & 0.213 \\ 0.188 & -0.249 & -0.950 \\ -0.126 & -0.965 & 0.228 \end{pmatrix}^\top ,$$

respectively.

In this case, the distance between these weights is $\|\boldsymbol{W}^{(a)} - \boldsymbol{W}^{(b)}\| \approx 1.236$. On the other hand, if the input vector $\boldsymbol{z}$ is given by $\boldsymbol{z} = k \begin{pmatrix} 0.974 & -0.077 & 0.213 \end{pmatrix}$, where $k$ is an arbitrary (but not too large) real scalar value, then $\|\sigma(\boldsymbol{W}^{(a)}\boldsymbol{z}) - \sigma(\boldsymbol{W}^{(b)}\boldsymbol{z})\| \approx 0$ holds.

## D EXPERIMENTAL SETUP

This section describes the experimental setup for training neural networks to obtain SGD solutions. We apply Sinkhorn's algorithm for permutation based on WM and STE. Thus, we also provide detailed information on the experimental setup for Sinkhorn's algorithm. Four datasets were used in this study: MNIST (Lecun et al., 1998), Fashion-MNIST (FMNIST) (Xiao et al., 2017), CIFAR10 (Krizhevsky & Hinton, 2009), and ImageNet (Deng et al., 2009).

All experiments were conducted on a Linux workstation with two AMD EPYC 7543 32-Core processors, eight NVIDIA A30 GPUs, and 512 GB of memory. The PyTorch 2.1.0[4], PyTorch Lightning 2.1.0[5], and torchvision 0.16.0[6] libraries were used for model training and evaluation.

### D.1 MODEL TRAINING

**MLP on MNIST and FMNIST.** Following the settings in (Ainsworth et al., 2023), we trained a Multi-Layer Perceptron (MLP) with three hidden layers, each comprising 512 units. The hidden layers use the ReLU function as their activation function. For the MNIST and FMNIST datasets, we optimized using the Adam algorithm with a learning rate of $1 \times 10^{-3}$. The batch size and maximum number of epochs were set to 512 and 100, respectively.

**VGG11 and ResNet20 on CIFAR10.** We utilized the VGG16 and ResNet20 architectures of (Ainsworth et al., 2023). To accomplish Linear Mode Connectivity (LMC), we increased the widths of VGG11 and ResNet20 by factors of 4 and 16, respectively. As described in (Jordan et al., 2023), we used the training dataset to repair the BatchNorm layers in these models during model merging. Optimization was conducted using Adam with a learning rate of $1 \times 10^{-3}$. The batch size and maximum number of epochs were set to 512 and 100, respectively. The following data augmentations were performed during training: random $32 \times 32$ pixel crops, and random horizontal flips.

**ResNet50 on ImageNet.** ResNet50 models were trained using a training script published on GitHub[7] by the FFCV library (Leclerc et al., 2023). The "rn50_40_epochs.yaml" file in the repository was used for the training setup. In the file, we changed `use_blurpool` to "0". As described in (Jordan et al., 2023), we repaired the BatchNorm layers in these models during model merging by using the training dataset. Since ImageNet is a large dataset, we used 50,000 randomly selected images from the training set to repair the batch normalization layers.

### D.2 PERMUTATION SEARCH

For permutation search in WM and STE, we employed the method based on Sinkhorn's algorithm as proposed by Guerrero-Peña et al. (2023). We utilized the implementation provided by the authors in their GitHub repository[8]. DistL2Loss and MidLoss were used as loss functions for permutation

---

[4]https://pytorch.org/
[5]https://lightning.ai/docs/pytorch/stable/
[6]https://pytorch.org/vision/stable/index.html
[7]https://github.com/libffcv/ffcv-imagenet/tree/main
[8]https://github.com/fagp/sinkhorn-rebasin

searches corresponding to WM and STE, respectively. Optimization was performed using Adam with a learning rate of 1 for MLP, VGG11, and ResNet20 and 10 for ResNet50, setting the maximum number of epochs to 10 for DistL2Loss and five for MidLoss. For MidLoss, the batch size was set to 512; for DistL2Loss, there was no batch size because the dataset was not used. 100 iterations of parameter updates were performed per epoch for DistL2Loss.

For activation matching (AM), the permutation search is divided for each layer, so the optimal solution can be obtained efficiently. We implemented AM-based permutation search following the GitHub repository released by Ainsworth et al. (2023)[9]. In this paper, the `linear_sum_assignment` function of Scipy (Virtanen et al., 2020) was used for the permutation search in AM.

## E    DISCUSSION ON COMMUTATIVITY PROPERTY

Zhou et al. (2023) show that LMC is satisfied if weak additivity for ReLU activations and commutativity hold. Given two models, $\boldsymbol{\theta}_a$ and $\boldsymbol{\theta}_b$, commutativity between them is satisfied if for all layers $\ell \in [L]$, $\boldsymbol{W}_\ell^{(a)} \boldsymbol{z}_{\ell-1}^{(a)} + \boldsymbol{W}_\ell^{(b)} \boldsymbol{z}_{\ell-1}^{(b)} = \boldsymbol{W}_\ell^{(a)} \boldsymbol{z}_{\ell-1}^{(b)} + \boldsymbol{W}_\ell^{(b)} \boldsymbol{z}_{\ell-1}^{(a)}$ holds, where $L$ is the number of layers of the models[10]. The commutativity property can be rewritten as $\forall \ell \in [L]; (\boldsymbol{W}_\ell^{(a)} - \boldsymbol{W}_\ell^{(b)})(\boldsymbol{z}_{\ell-1}^{(a)} - \boldsymbol{z}_{\ell-1}^{(b)}) = \boldsymbol{0}$. Therefore, Zhou et al. (2023) in Section 5.2 justify the WM-based permutation search method because WM aims to minimize Equation (1), which corresponds to the first factor in the equation. However, as shown in our paper, WM only slightly reduces the distance between the two models, contradicting their claim.

Appendix B.5 of (Zhou et al., 2023) explains in a different way why the commutativity property is satisfied in WM. Specifically, they consider a stronger form of the commutativity property:

$$\forall \ell \in [L]; \boldsymbol{W}_\ell^{(a)} \boldsymbol{z}_{\ell-1}^{(a)} = \boldsymbol{W}_\ell^{(b)} \boldsymbol{z}_{\ell-1}^{(a)} \wedge \boldsymbol{W}_\ell^{(b)} \boldsymbol{z}_{\ell-1}^{(b)} = \boldsymbol{W}_\ell^{(a)} \boldsymbol{z}_{\ell-1}^{(b)}. \tag{11}$$

To ensure this stronger form, we need to find $\boldsymbol{P}_\ell$ and $\boldsymbol{P}_{\ell-1}$ such that:

$$(\boldsymbol{W}_\ell^{(a)} - \boldsymbol{P}_\ell \boldsymbol{W}_\ell^{(b)} \boldsymbol{P}_{\ell-1}^\top) \boldsymbol{z}_{\ell-1}^{(a)} = \boldsymbol{0} \wedge (\boldsymbol{P}_\ell \boldsymbol{W}_\ell^{(b)} \boldsymbol{P}_{\ell-1}^\top - \boldsymbol{W}_\ell^{(a)}) \boldsymbol{P}_{\ell-1} \boldsymbol{z}_{\ell-1}^{(b)} = \boldsymbol{0}. \tag{12}$$

They argue that the commutativity property easily holds because it is easy to find permutation matrices $\boldsymbol{P}_\ell$ and $\boldsymbol{P}_{\ell-1}$ that satisfy Equation (12) due to the small actual dimension of the hidden layer inputs $\boldsymbol{z}_{\ell-1}^{(a)}$ and $\boldsymbol{z}_{\ell-1}^{(b)}$ (i.e., these vectors are biased in a particular direction).

However, several points need to be addressed regarding this explanation. First, if Equation (11) holds, then LMC is satisfied without any assumptions, such as the commutativity property or weak additivity of ReLU activations. Since this equation must hold for all layers, it must also hold for the input layer where $\boldsymbol{z}_{\ell-1}^{(a)} = \boldsymbol{z}_{\ell-1}^{(b)} = \boldsymbol{x}$. In that case, the outputs of the input layers are equivalent between the two models. The same holds for subsequent layers, so the outputs of the two models are the same in all hidden layers. Therefore, the outputs of the hidden layers of the merged model must also be identical to those of the pre-merged models in all layers. Thus, LMC obviously holds. In other words, if the reason of the establishment of LMC is that Equation (11) holds, then the essential reason for the establishment of LMC is that WM makes the outputs of the hidden layers of the two models close, suggesting that our argument in this paper is more fundamental in establishing LMC.

Second, the small actual dimension of the hidden layer inputs $\boldsymbol{z}_{\ell-1}^{(a)}$ and $\boldsymbol{z}_{\ell-1}^{(b)}$ does not necessarily mean that Equation (12) is easier to satisfy by finding permutations that minimize Equation (1). As shown in Section 4.3, WM preferentially aligns the directions of singular vectors with large singular values, while other singular vectors are difficult to align. If the hidden layer inputs $\boldsymbol{z}_{\ell-1}^{(a)}$ and $\boldsymbol{z}_{\ell-1}^{(b)}$ are not oriented in the same directions as the right singular vectors with large singular values, then WM will not help satisfy Equation (12). Zhou et al. (2023) did not mention this second point. On the other hand, we analyzed this point in Section 4.4.

## F    CONVOLUTIONAL LAYERS

This section discusses a theorem similar to Theorem 4.1 for convolutional neural networks (CNNs).

---

[9] https://github.com/samuela/git-re-basin

[10] Strictly speaking, given a data distribution $\mathcal{D}$, the commutativity property is satisfied if the equation almost surely holds for $\mathcal{D}$. This definition is equivalent to Zhou et al. (2023), although it differs slightly.

### F.1 NOTATION

We introduce the notation used in the following sections. Each element of a tensor is specified by a simple italic variable with subscripts. For example, for a third-order tensor $\boldsymbol{X}$, its $i, j, k$-th component is denoted by $X_{i,j,k}$. We also use Python-like slice notation. For example, $\boldsymbol{X}_{1,:}$ denotes the first row of the matrix $\boldsymbol{X}$. For a complex matrix $\boldsymbol{X}$, let $\boldsymbol{X}^* = \overline{\boldsymbol{X}}^\top$ be its unitary transpose, where $\overline{\boldsymbol{X}}$ denotes the complex conjugate of $\boldsymbol{X}$.

### F.2 MATRIX REPRESENTATION OF CONVOLUTIONAL LAYER

This subsection introduces the matrix representation of a convolutional layer. Let $\boldsymbol{X} \in \mathbb{R}^{m \times n \times n}$ and $\boldsymbol{Y} \in \mathbb{R}^{m \times n \times n}$ be the input and the output of the $\ell$-th convolutional layer, respectively. Here, $m$ denotes the number of input and output channels and $n$ denotes the size of the height and width of the input. For simplicity, we assume that the numbers of output channels and input channels are identical, as well as the sizes of the height and width of the input, although our analysis is applicable even when they are not. Let $\boldsymbol{K} \in \mathbb{R}^{n \times n \times m \times m}$ be the kernel of the $\ell$-th layer. Then, for the $c, r, i$-th element of the output $\boldsymbol{Y}$ is given by

$$Y_{c,r,i} = \sum_{d \in [m], p \in [n], q \in [n]} X_{d,r+p,i+q} K_{p,q,c,d}.$$

There exists a matrix $\boldsymbol{M}$ such that $\text{vec}(\boldsymbol{Y}) = \boldsymbol{M}\text{vec}(\boldsymbol{X})$ holds (Sedghi et al., 2019; Jain, 1989; Salakhutdinov, 2014), where

$$\boldsymbol{M} = \begin{pmatrix} \boldsymbol{B}_{1,1} & \boldsymbol{B}_{1,2} & \dots & \boldsymbol{B}_{1,m} \\ \boldsymbol{B}_{2,1} & \boldsymbol{B}_{2,2} & \dots & \boldsymbol{B}_{2,m} \\ \vdots & \vdots & \ddots & \vdots \\ \boldsymbol{B}_{m,1} & \boldsymbol{B}_{m,2} & \dots & \boldsymbol{B}_{m,m} \end{pmatrix}. \tag{13}$$

Here, for all $c, d \in [m]$, $\boldsymbol{B}_{c,d}$ is a doubly circulant matrix defined by

$$\boldsymbol{B}_{c,d} = \begin{pmatrix} \text{circ}(K_{1,:,c,d}) & \text{circ}(K_{2,:,c,d}) & \dots & \text{circ}(K_{n,:,c,d}) \\ \text{circ}(K_{n,:,c,d}) & \text{circ}(K_{1,:,c,d}) & \dots & \text{circ}(K_{n-1,:,c,d}) \\ \vdots & \vdots & \ddots & \vdots \\ \text{circ}(K_{2,:,c,d}) & \text{circ}(K_{3,:,c,d}) & \dots & \text{circ}(K_{1,:,c,d}) \end{pmatrix},$$

where circ is a function to generate a circulant matrix from a given vector. For example, given a vector $\boldsymbol{a} = (a_1, a_2, \dots, a_3)$, the circulant matrix generated by $\boldsymbol{a}$ is given by $\text{circ}(\boldsymbol{a}) = \begin{pmatrix} a_1 & a_2 & a_3 \\ a_3 & a_1 & a_2 \\ a_2 & a_3 & a_1 \end{pmatrix}$.

### F.3 SINGULAR VALUE DECOMPOSITION AND WEIGHT MATCHING OF CONVOLUTIONAL LAYERS

Since Equation (13) represents the matrix form of the convolutional layer, we can reach a conclusion similar to Theorem 4.1 by performing a singular value decomposition (SVD) on it. However, this matrix is very large, with a size of $mn^2 \times mn^2$, making direct SVD impractical. Therefore, we decompose it into a more SVD-friendly form using a Fourier transform. Using the imaginary unit as $\eta = \sqrt{-1}$, and setting $\omega = e^{-2\pi\eta/n}$, a one-dimensional Fourier transform matrix $\boldsymbol{F}$ is defined by $F_{i,j} = (\omega^{(i-1)(j-1)})_{i,j}$.[11] A matrix for the two-dimensional Fourier transform can be defined as $\boldsymbol{Q} = (\boldsymbol{F} \otimes \boldsymbol{F})/n$. Here, $\otimes$ denotes the Kronecker product. By using this two-dimensional Fourier transform matrix $\boldsymbol{Q}$, the matrix $\boldsymbol{M}$ can be decomposed as follows:

$$\boldsymbol{M} = (\boldsymbol{I}_m \otimes \boldsymbol{Q})^* \boldsymbol{L} (\boldsymbol{I}_m \otimes \boldsymbol{Q}),$$

---

[11]Usually, the alphabet letters $i$ or $j$ are used for the imaginary unit, but since they are used as indices here, we use $\eta$.

where $\boldsymbol{I}_m$ denotes the identity matrix of size $m \times m$. We then have

$$\boldsymbol{L} = \begin{pmatrix} \boldsymbol{D}_{1,1} & \boldsymbol{D}_{1,2} & \dots & \boldsymbol{D}_{1,m} \\ \boldsymbol{D}_{2,1} & \boldsymbol{D}_{2,2} & \dots & \boldsymbol{D}_{2,m} \\ \vdots & \vdots & \ddots & \vdots \\ \boldsymbol{D}_{m,1} & \boldsymbol{D}_{m,2} & \dots & \boldsymbol{D}_{m,m} \end{pmatrix}.$$

Here, for all $c, d \in [m]$, $\boldsymbol{D}_{c,d} = \boldsymbol{Q}\boldsymbol{B}_{c,d}\boldsymbol{Q}^*$ is a complex diagonal matrix (Sedghi et al., 2019). Let $\boldsymbol{G}_{:,:,w}$ be a matrix formed by extracting the $w$-th diagonal element of each diagonal matrix $\boldsymbol{D}_{c,d}$ and arranging them (i.e., $G_{c,d,w} = (\boldsymbol{D}_{c,d})_{w,w}$). Then, the following theorem holds:

**Theorem F.1** (SVD of convolutional layer). *Let $s_{w,i}$, $\boldsymbol{u}_{w,i}$, and $\boldsymbol{v}_{w,i}$ be the $i$-th singular value, left singular vector, and right singular vector of $\boldsymbol{G}_{:,:,w}$, respectively. Then, the matrix $\boldsymbol{M}$ representing the convolutional layer can be decomposed as follows:*

$$\boldsymbol{M} = \sum_{w,i} (\boldsymbol{u}_{w,i} \otimes \boldsymbol{Q}^* \boldsymbol{e}_w) s_{w,i} (\boldsymbol{v}_{w,i} \otimes \boldsymbol{Q}^* \boldsymbol{e}_w)^*,$$

*where $\boldsymbol{e}_w$ represents the orthonormal basis in Euclidean space $\mathbb{R}^{n^2}$, and $s_{w,i}$, $\boldsymbol{u}_{w,i} \otimes \boldsymbol{Q}^* \boldsymbol{e}_w$, and $\boldsymbol{v}_{w,i} \otimes \boldsymbol{Q}^* \boldsymbol{e}_w$ are the singular value, left singular vector, and right singular vector of $\boldsymbol{M}$, respectively.*

The proof is shown in Appendix G.3. From this theorem, the following theorem can be proved:

**Theorem F.2.** *Let $\boldsymbol{M}^{(a)}$ and $\boldsymbol{M}^{(b)}$ be the matrix representations of convolutional layers of two CNNs. From Theorem F.1, their SVDs are given by $\boldsymbol{M}^{(a)} = \sum_{w,i}(\boldsymbol{u}_{w,i}^{(a)} \otimes \boldsymbol{Q}^*\boldsymbol{e}_w)s_{w,i}^{(a)}(\boldsymbol{v}_{w,i}^{(a)} \otimes \boldsymbol{Q}^*\boldsymbol{e}_w)^*$ and $\boldsymbol{M}^{(b)} = \sum_{w,i}(\boldsymbol{u}_{w,i}^{(b)} \otimes \boldsymbol{Q}^*\boldsymbol{e}_w)s_{w,i}^{(b)}(\boldsymbol{v}_{w,i}^{(b)} \otimes \boldsymbol{Q}^*\boldsymbol{e}_w)^*$, respectively. Then, the WM between $\boldsymbol{M}^{(a)}$ and $\boldsymbol{M}^{(b)}$ is equivalent to finding permutation matrices $\boldsymbol{P}_\ell$ and $\boldsymbol{P}_{\ell-1}$ such that*

$$\underset{\boldsymbol{P}_\ell, \boldsymbol{P}_{\ell-1}}{\arg\max} \, \Re \sum_{w,i,j} s_{w,i}^{(a)} s_{w,j}^{(b)} \overline{(\boldsymbol{u}_{w,i}^{(a)})^*(\boldsymbol{P}_\ell \boldsymbol{u}_{w,j}^{(b)})} (\boldsymbol{v}_{w,i}^{(a)})^*(\boldsymbol{P}_{\ell-1}\boldsymbol{v}_{w,j}^{(b)}),$$

*where $\Re z$ is the real part of $z$ for a complex number $z$.*

The proof is shown in Appendix G.4. Similar to the case of MLP (Theorem 4.1), Theorem F.2 indicates that WM has the effect of aligning the directions of the corresponding singular vectors in convolutional layers.

# G PROOFS

## G.1 PROOF OF THEOREM 3.1

*Proof.* From the assumption and Taylor theorem centered at $\boldsymbol{\theta}_a$, we have

$$\mathcal{L}(\boldsymbol{\theta}_c) = \mathcal{L}(\boldsymbol{\theta}_a) + (\boldsymbol{\theta}_c - \boldsymbol{\theta}_a)\nabla\mathcal{L}(\boldsymbol{\theta}_a) + \frac{1}{2}(\boldsymbol{\theta}_c - \boldsymbol{\theta}_a)^\top \boldsymbol{H}_a(\boldsymbol{\theta}_c - \boldsymbol{\theta}_a) + O(\|\boldsymbol{\theta}_c - \boldsymbol{\theta}_a\|^3)$$

$$= \mathcal{L}(\boldsymbol{\theta}_a) + (1 - \lambda)\beta\boldsymbol{\mu}\nabla\mathcal{L}(\boldsymbol{\theta}_a) + \frac{1}{2}(1 - \lambda)^2\beta^2\boldsymbol{\mu}^\top \boldsymbol{H}_a\boldsymbol{\mu} + O(\beta^3).$$

Similarly, using Taylor theorem centered at $\pi(\boldsymbol{\theta}_b)$, we get

$$\mathcal{L}(\boldsymbol{\theta}_c) = \mathcal{L}(\pi(\boldsymbol{\theta}_b)) + (\boldsymbol{\theta}_c - \pi(\boldsymbol{\theta}_b))\nabla\mathcal{L}(\pi(\boldsymbol{\theta}_b)) + \frac{1}{2}(\boldsymbol{\theta}_c - \pi(\boldsymbol{\theta}_b))^\top \boldsymbol{H}_a(\boldsymbol{\theta}_c - \pi(\boldsymbol{\theta}_b)) + O(\|\boldsymbol{\theta}_c - \pi(\boldsymbol{\theta}_b)\|^3)$$

$$= \mathcal{L}(\pi(\boldsymbol{\theta}_b)) - \lambda\beta\boldsymbol{\mu}\nabla\mathcal{L}(\pi(\boldsymbol{\theta}_b)) + \frac{1}{2}\lambda^2\beta^2\boldsymbol{\mu}^\top \boldsymbol{H}_a\boldsymbol{\mu} + O(\beta^3).$$

Combining these equations, the barrier can be obtained as

$$B(\boldsymbol{\theta}_a, \pi(\boldsymbol{\theta}_b)) = \max_\lambda \left(\mathcal{L}(\boldsymbol{\theta}_c) - \lambda\mathcal{L}(\boldsymbol{\theta}_a) - (1 - \lambda)\mathcal{L}(\pi(\boldsymbol{\theta}_b))\right)$$

$$= \max_\lambda \left(\lambda(\mathcal{L}(\boldsymbol{\theta}_c) - \mathcal{L}(\boldsymbol{\theta}_a)) + (1 - \lambda)(\mathcal{L}(\boldsymbol{\theta}_c) - \mathcal{L}(\pi(\boldsymbol{\theta}_b)))\right)$$

$$= \max_\lambda \left(\beta\lambda(1 - \lambda)\boldsymbol{\mu}^\top(\nabla\mathcal{L}(\boldsymbol{\theta}_a) - \nabla\mathcal{L}(\pi(\boldsymbol{\theta}_b)))\right.$$

$$\left. + \frac{1}{2}\beta^2\lambda(1 - \lambda)\boldsymbol{\mu}^\top((1 - \lambda)\boldsymbol{H}_a + \lambda\boldsymbol{H}_b)\boldsymbol{\mu}\right) + O(\beta^3).$$

$\square$

### G.2 PROOF OF THEOREM 4.1

*Proof.* Consider the $L^2$ norm of the $\ell$-th layer $\|\boldsymbol{W}_\ell^{(a)} - \boldsymbol{P}_\ell \boldsymbol{W}_\ell^{(b)} \boldsymbol{P}_{\ell-1}^\top\|^2$. Using the fact that the $L^2$ norm can be rewritten using trace, we have

$$
\begin{aligned}
\|\boldsymbol{W}_\ell^{(a)} - \boldsymbol{P}_\ell \boldsymbol{W}_\ell^{(b)} \boldsymbol{P}_{\ell-1}^\top\|^2 &= \operatorname{tr}\left((\boldsymbol{W}_\ell^{(a)} - \boldsymbol{P}_\ell \boldsymbol{W}_\ell^{(b)} \boldsymbol{P}_{\ell-1}^\top)(\boldsymbol{W}_\ell^{(a)} - \boldsymbol{P}_\ell \boldsymbol{W}_\ell^{(b)} \boldsymbol{P}_{\ell-1}^\top)^\top\right) \\
&= \operatorname{tr}\left(\boldsymbol{W}_\ell^{(a)}(\boldsymbol{W}_\ell^{(a)})^\top\right) + \operatorname{tr}\left(\boldsymbol{W}_\ell^{(b)}(\boldsymbol{W}_\ell^{(b)})^\top\right) \\
&\qquad\qquad - 2\operatorname{tr}\left(\boldsymbol{P}_\ell \boldsymbol{W}_\ell^{(b)} \boldsymbol{P}_{\ell-1}^\top (\boldsymbol{W}_\ell^{(a)})^\top\right).
\end{aligned}
\tag{14}
$$

We focus on the last term because only it depends on the permutation matrices. The SVDs of the weights $\boldsymbol{W}_\ell^{(a)}$ and $\boldsymbol{W}_\ell^{(b)}$ are denoted by $\boldsymbol{W}_\ell^{(a)} = \boldsymbol{U}_\ell^{(a)} \boldsymbol{S}_\ell^{(a)} (\boldsymbol{V}_\ell^{(a)})^\top = \sum_i \boldsymbol{u}_{\ell,i}^{(a)} s_{\ell,i}^{(a)} (\boldsymbol{v}_{\ell,i}^{(a)})^\top$ and $\boldsymbol{W}_\ell^{(b)} = \boldsymbol{U}_\ell^{(b)} \boldsymbol{S}_\ell^{(b)} (\boldsymbol{V}_\ell^{(b)})^\top = \sum_j \boldsymbol{u}_{\ell,j}^{(b)} s_{\ell,j}^{(b)} (\boldsymbol{v}_{\ell,j}^{(b)})^\top$, respectively. Thus, the last term of [Equation (14)] can be rewritten as

$$
-2\operatorname{tr}\left(\boldsymbol{P}_\ell \boldsymbol{W}_\ell^{(b)} \boldsymbol{P}_{\ell-1}^\top (\boldsymbol{W}_\ell^{(a)})^\top\right) = -2\sum_{i,j} s_{\ell,i}^{(a)} s_{\ell,j}^{(b)} (\boldsymbol{u}_{\ell,i}^{(a)})^\top (\boldsymbol{P}_\ell \boldsymbol{u}_{\ell,j}^{(b)})(\boldsymbol{v}_{\ell,i}^{(a)})^\top (\boldsymbol{P}_{\ell-1} \boldsymbol{v}_{\ell,j}^{(b)}).
$$

Therefore, [Equation (1)] equals

$$
\arg\min_\pi \|\boldsymbol{\theta}_a - \pi(\boldsymbol{\theta}_b)\|^2 = \arg\max_{\pi=(\boldsymbol{P}_\ell)_\ell} \sum_{\ell,i,j} s_{\ell,i}^{(a)} s_{\ell,j}^{(b)} (\boldsymbol{u}_{\ell,i}^{(a)})^\top (\boldsymbol{P}_\ell \boldsymbol{u}_{\ell,j}^{(b)})(\boldsymbol{v}_{\ell,i}^{(a)})^\top (\boldsymbol{P}_{\ell-1} \boldsymbol{v}_{\ell,j}^{(b)}),
$$

which completes the proof. $\square$

### G.3 PROOF OF THEOREM F.1

*Proof.* Note that the matrix $\boldsymbol{L}$ can be decomposed to $\boldsymbol{L} = \sum_w \boldsymbol{G}_{:,:,w} \otimes (\boldsymbol{e}_w \boldsymbol{e}_w^\top)$ by using the tensor $\boldsymbol{G}$, where $\boldsymbol{e}_w$ is the orthonormal basis in Eucrlidean space $\boldsymbol{R}^{n^2}$. Thus, the SVD of $\boldsymbol{L}$ is given by

$$
\begin{aligned}
\boldsymbol{L} &= \sum_w \left(\sum_i \boldsymbol{u}_{w,i} s_{w,i} \boldsymbol{v}_{w,i}^*\right) \otimes (\boldsymbol{e}_w \boldsymbol{e}_w^\top) \\
&= \sum_w \sum_i s_{w,i} (\boldsymbol{u}_{w,i} \boldsymbol{v}_{w,i}^*) \otimes (\boldsymbol{e}_w \boldsymbol{e}_w^\top) \\
&= \sum_w \sum_i s_{w,i} (\boldsymbol{u}_{w,i} \otimes \boldsymbol{e}_w)(\boldsymbol{v}_{w,i} \otimes \boldsymbol{e}_w)^*.
\end{aligned}
$$

Thus, the SVD of $\boldsymbol{M}$ is also given by

$$
\begin{aligned}
\boldsymbol{M} &= \sum_w \sum_i s_{w,i} (\boldsymbol{I}_m \otimes \boldsymbol{Q})^* (\boldsymbol{u}_{w,i} \otimes \boldsymbol{e}_w)(\boldsymbol{v}_{w,i} \otimes \boldsymbol{e}_w)^* (\boldsymbol{I}_m \otimes \boldsymbol{Q}) \\
&= \sum_w \sum_i s_{w,i} (\boldsymbol{u}_{w,i} \otimes \boldsymbol{Q}^* \boldsymbol{e}_w)(\boldsymbol{v}_{w,i}^* \otimes \boldsymbol{e}_w^\top \boldsymbol{Q}),
\end{aligned}
$$

which completes the proof. $\square$

### G.4 PROOF OF THEOREM F.2

Before proving [Theorem F.2], we first prove the following lemma:

**Lemma G.1.** *Let $\boldsymbol{K}$ and $\boldsymbol{K}'$ be kernels of convolutional layers. We have $\|\boldsymbol{M} - \boldsymbol{M}'\|^2 = n^2\|\boldsymbol{K} - \boldsymbol{K}'\|^2$, where $\boldsymbol{M}$ and $\boldsymbol{M}'$ are the matrix representations of $\boldsymbol{K}$ and $\boldsymbol{K}'$, respectively.*

*Proof.* From the definition of $\boldsymbol{M}$ and $\boldsymbol{M}'$, we have

$$
\boldsymbol{M} = \begin{pmatrix} \boldsymbol{B}_{1,1} & \boldsymbol{B}_{1,2} & \cdots & \boldsymbol{B}_{1,m} \\ \boldsymbol{B}_{2,1} & \boldsymbol{B}_{2,2} & \cdots & \boldsymbol{B}_{2,m} \\ \vdots & \vdots & \ddots & \vdots \\ \boldsymbol{B}_{m,1} & \boldsymbol{B}_{m,2} & \cdots & \boldsymbol{B}_{m,m} \end{pmatrix}, \quad \boldsymbol{M}' = \begin{pmatrix} \boldsymbol{B}'_{1,1} & \boldsymbol{B}'_{1,2} & \cdots & \boldsymbol{B}'_{1,m} \\ \boldsymbol{B}'_{2,1} & \boldsymbol{B}'_{2,2} & \cdots & \boldsymbol{B}'_{2,m} \\ \vdots & \vdots & \ddots & \vdots \\ \boldsymbol{B}'_{m,1} & \boldsymbol{B}'_{m,2} & \cdots & \boldsymbol{B}'_{m,m} \end{pmatrix},
$$

where for any $c, d \in [m]$, $\boldsymbol{B}_{c,d}$ and $\boldsymbol{B}'_{c,d}$ denote the doubly circulant matrices obtained from the kernels $\boldsymbol{K}$ and $\boldsymbol{K}'$. Thus, $\|\boldsymbol{M} - \boldsymbol{M}'\|^2 = \sum_{c,d}\|\boldsymbol{B}_{c,d} - \boldsymbol{B}'_{c,d}\|^2 = n\sum_{c,d}\sum_i\|\mathrm{circ}(K_{i,:,c,d}) - \mathrm{circ}(K'_{i,:,c,d})\|^2 = n^2\sum_{c,d}\sum_{i,j}(K_{i,j,c,d} - K'_{i,j,c,d})^2 = n^2\|\boldsymbol{K} - \boldsymbol{K}'\|^2$ holds. $\qquad\square$

*Proof of Theorem F.2.* In convolutional layers, permutation matrices permute the input and output channels of the kernel. Therefore, the permutation matrices $\boldsymbol{P}_\ell$ and $\boldsymbol{P}_{\ell-1}$ corresponding to the input and output are $m \times m$ matrices. By using these matrices, the permutation of the matrix representation of the convolutional layer of the model $\boldsymbol{\theta}_b$ is denoted by $(\boldsymbol{P}_\ell \otimes \boldsymbol{I}_m)\boldsymbol{M}^{(b)}(\boldsymbol{P}_{\ell-1} \otimes \boldsymbol{I}_m)^\top$. Lemma G.1 indicates that finding the permutation matrices that minimize the $L^2$ distance between the two kernels is equivalent to minimizing $\|\boldsymbol{M}^{(a)} - (\boldsymbol{P}_\ell \otimes \boldsymbol{I}_m)\boldsymbol{M}^{(b)}(\boldsymbol{P}_{\ell-1} \otimes \boldsymbol{I}_m)^\top\|$. Therefore, we have

$$\|\boldsymbol{M}^{(a)} - (\boldsymbol{P}_\ell \otimes \boldsymbol{I}_m)\boldsymbol{M}^{(b)}(\boldsymbol{P}_{\ell-1} \otimes \boldsymbol{I}_m)^\top\|^2$$

$$= \mathrm{tr}\left(\left(\boldsymbol{M}^{(a)} - (\boldsymbol{P}_\ell \otimes \boldsymbol{I}_m)\boldsymbol{M}^{(b)}(\boldsymbol{P}_{\ell-1} \otimes \boldsymbol{I}_m)^\top\right)\left(\boldsymbol{M}^{(a)} - (\boldsymbol{P}_\ell \otimes \boldsymbol{I}_m)\boldsymbol{M}^{(b)}(\boldsymbol{P}_{\ell-1} \otimes \boldsymbol{I}_m)^\top\right)^*\right)$$

$$= \mathrm{tr}\left(\boldsymbol{M}^{(a)}(\boldsymbol{M}^{(a)})^\top\right) + \mathrm{tr}\left(\boldsymbol{M}^{(b)}(\boldsymbol{M}^{(b)})^\top\right) - \mathrm{tr}\left(\boldsymbol{M}^{(a)}\left((\boldsymbol{P}_\ell \otimes \boldsymbol{I}_m)\boldsymbol{M}^{(b)}(\boldsymbol{P}_{\ell-1} \otimes \boldsymbol{I}_m)^\top\right)^*\right)$$

$$- \mathrm{tr}\left((\boldsymbol{P}_\ell \otimes \boldsymbol{I}_m)\boldsymbol{M}^{(b)}(\boldsymbol{P}_{\ell-1} \otimes \boldsymbol{I}_m)^\top(\boldsymbol{M}^{(a)})^*\right). \tag{15}$$

In Equation (15), the permutation matrices $\boldsymbol{P}_\ell$ and $\boldsymbol{P}_{\ell-1}$ are only related to the last two terms. Therefore, we focus only on them. By using some properties of trace, we have

$$- \mathrm{tr}\left(\boldsymbol{M}^{(a)}\left((\boldsymbol{P}_\ell \otimes \boldsymbol{I}_m)\boldsymbol{M}^{(b)}(\boldsymbol{P}_{\ell-1} \otimes \boldsymbol{I}_m)^\top\right)^*\right) - \mathrm{tr}\left((\boldsymbol{P}_\ell \otimes \boldsymbol{I}_m)\boldsymbol{M}^{(b)}(\boldsymbol{P}_{\ell-1} \otimes \boldsymbol{I}_m)^\top(\boldsymbol{M}^{(a)})^*\right)$$

$$= -\mathrm{tr}\left(\boldsymbol{M}^{(a)}\left((\boldsymbol{P}_\ell \otimes \boldsymbol{I}_m)\boldsymbol{M}^{(b)}(\boldsymbol{P}_{\ell-1} \otimes \boldsymbol{I}_m)^\top\right)^*\right) - \mathrm{tr}\left(\left(\boldsymbol{M}^{(a)}\left((\boldsymbol{P}_\ell \otimes \boldsymbol{I}_m)\boldsymbol{M}^{(b)}(\boldsymbol{P}_{\ell-1} \otimes \boldsymbol{I}_m)^\top\right)^*\right)^*\right)$$

$$= -\mathrm{tr}\left(\boldsymbol{M}^{(a)}\left((\boldsymbol{P}_\ell \otimes \boldsymbol{I}_m)\boldsymbol{M}^{(b)}(\boldsymbol{P}_{\ell-1} \otimes \boldsymbol{I}_m)^\top\right)^*\right) - \overline{\mathrm{tr}\left(\boldsymbol{M}^{(a)}\left((\boldsymbol{P}_\ell \otimes \boldsymbol{I}_m)\boldsymbol{M}^{(b)}(\boldsymbol{P}_{\ell-1} \otimes \boldsymbol{I}_m)^\top\right)^*\right)}$$

$$= -2\Re\,\mathrm{tr}\left(\boldsymbol{M}^{(a)}\left((\boldsymbol{P}_\ell \otimes \boldsymbol{I}_m)\boldsymbol{M}^{(b)}(\boldsymbol{P}_{\ell-1} \otimes \boldsymbol{I}_m)^\top\right)^*\right).$$

Here, from Theorem F.1,

$$\boldsymbol{M}^{(a)} = \sum_{w,i}(\boldsymbol{u}_{w,i}^{(a)} \otimes \boldsymbol{Q}^*\boldsymbol{e}_w)s_{w,i}^{(a)}(\boldsymbol{v}_{w,i}^{(a)} \otimes \boldsymbol{Q}^*\boldsymbol{e}_w)^*$$

$$= \sum_{w,i}s_{w,i}^{(a)}(\boldsymbol{u}_{w,i}^{(a)}(\boldsymbol{v}_{w,i}^{(a)})^* \otimes \boldsymbol{Q}^*\boldsymbol{e}_w\boldsymbol{e}_w^\top\boldsymbol{Q})$$

$$= \sum_w(\boldsymbol{C}_w^{(a)} \otimes \boldsymbol{Q}^*\boldsymbol{e}_w\boldsymbol{e}_w^\top\boldsymbol{Q}), \tag{16}$$

and

$$(\boldsymbol{P}_\ell \otimes \boldsymbol{I}_m)\boldsymbol{M}^{(b)}(\boldsymbol{P}_{\ell-1} \otimes \boldsymbol{I}_m)^\top = \sum_{w,i}s_{w,i}^{(b)}(\boldsymbol{P}_\ell \otimes \boldsymbol{I}_m)(\boldsymbol{u}_{w,i}^{(b)} \otimes \boldsymbol{Q}^*\boldsymbol{e}_w)(\boldsymbol{v}_{w,i}^{(b)} \otimes \boldsymbol{Q}^*\boldsymbol{e}_w)^*(\boldsymbol{P}_{\ell-1} \otimes \boldsymbol{I}_m)^\top$$

$$= \sum_{w,i}s_{w,i}^{(b)}(\boldsymbol{P}_\ell\boldsymbol{u}_{w,i}^{(b)}(\boldsymbol{P}_{\ell-1}\boldsymbol{v}_{w,i}^{(b)})^* \otimes \boldsymbol{Q}^*\boldsymbol{e}_w\boldsymbol{e}_w^\top\boldsymbol{Q})$$

$$= \sum_w(\boldsymbol{C}_w^{(b)} \otimes \boldsymbol{Q}^*\boldsymbol{e}_w\boldsymbol{e}_w^\top\boldsymbol{Q}) \tag{17}$$

hold, where we let $\boldsymbol{C}_w^{(a)} = \sum_i s_{w,i}^{(a)}\boldsymbol{u}_{w,i}^{(a)}(\boldsymbol{v}_{w,i}^{(a)})^*$ and $\boldsymbol{C}_w^{(b)} = \sum_i s_{w,i}^{(b)}(\boldsymbol{P}_\ell\boldsymbol{u}_{w,i}^{(a)})(\boldsymbol{P}_{\ell-1}\boldsymbol{v}_{w,i}^{(a)})^*$. From Equation (16) and Equation (17), we have

$$-2\Re\,\mathrm{tr}\left(\boldsymbol{M}^{(a)}\left((\boldsymbol{P}_\ell \otimes \boldsymbol{I}_m)\boldsymbol{M}^{(b)}(\boldsymbol{P}_{\ell-1} \otimes \boldsymbol{I}_m)^\top\right)^*\right)$$

$$= -2\Re\,\mathrm{tr}\left(\sum_w(\boldsymbol{C}_w^{(a)} \otimes \boldsymbol{Q}^*\boldsymbol{e}_w\boldsymbol{e}_w^\top\boldsymbol{Q})\left(\sum_{w'}(\boldsymbol{C}_{w'}^{(b)} \otimes \boldsymbol{Q}^*\boldsymbol{e}_{w'}\boldsymbol{e}_{w'}^\top\boldsymbol{Q})\right)^*\right)$$

$$= -2\Re\,\mathrm{tr}\left(\sum_{w,w'}\left(\boldsymbol{C}_w^{(a)}(\boldsymbol{C}_{w'}^{(b)})^* \otimes \boldsymbol{Q}^*\boldsymbol{e}_w\boldsymbol{e}_w^\top\boldsymbol{e}_{w'}\boldsymbol{e}_{w'}^\top\boldsymbol{Q}\right)\right).$$

Using the fact that if $w \neq w'$, then $\boldsymbol{e}_w^\top \boldsymbol{e}_{w'} = 0$, and otherwise, $\boldsymbol{e}_w^\top \boldsymbol{e}_{w'} = 1$, we get

$$-2\Re \operatorname{tr}\left(\boldsymbol{M}^{(a)}\left(\left(\boldsymbol{P}_\ell \otimes \boldsymbol{I}_m\right)\boldsymbol{M}^{(b)}\left(\boldsymbol{P}_{\ell-1} \otimes \boldsymbol{I}_m\right)^\top\right)^\top\right)$$

$$= -2\Re \sum_w \operatorname{tr}\left(\boldsymbol{C}_w^{(a)}(\boldsymbol{C}_w^{(b)})^* \otimes \boldsymbol{Q}^* \boldsymbol{e}_w \boldsymbol{e}_w^\top \boldsymbol{Q}\right)$$

$$= -2\Re \sum_w \operatorname{tr}\left(\boldsymbol{C}_w^{(a)}(\boldsymbol{C}_w^{(b)})^*\right)\operatorname{tr}\left(\boldsymbol{Q}^* \boldsymbol{e}_w \boldsymbol{e}_w^\top \boldsymbol{Q}\right)$$

$$= -2\Re \sum_w \operatorname{tr}\left(\boldsymbol{C}_w^{(a)}(\boldsymbol{C}_w^{(b)})^*\right)$$

$$= -2\Re \sum_w \operatorname{tr}\left(\left(\sum_i s_{w,i}^{(a)} \boldsymbol{u}_{w,i}^{(a)}(\boldsymbol{v}_{w,i}^{(a)})^*\right)\left(\sum_j s_{w,j}^{(b)}(\boldsymbol{P}_\ell \boldsymbol{u}_{w,j}^{(b)})(\boldsymbol{P}_{\ell-1}\boldsymbol{v}_{w,j}^{(b)})^*\right)^*\right)$$

$$= -2\Re \sum_w \sum_{i,j} s_{w,i}^{(a)} s_{w,j}^{(b)}\left((\boldsymbol{u}_{w,i}^{(a)})^*(\boldsymbol{P}_\ell \boldsymbol{u}_{w,j}^{(b)})\right)^*\left((\boldsymbol{v}_{w,i}^{(a)})^*(\boldsymbol{P}_{\ell-1}\boldsymbol{v}_{w,j}^{(b)})\right).$$

From the above, the minimization of $\|\boldsymbol{M}^{(a)} - (\boldsymbol{P}_\ell \otimes \boldsymbol{I}_m)\boldsymbol{M}^{(b)}(\boldsymbol{P}_{\ell-1} \otimes \boldsymbol{I}_m)^\top\|$ is equivalent to the maximization of $\Re \sum_w \sum_{i,j} s_{w,i}^{(a)} s_{w,j}^{(b)}\left((\boldsymbol{u}_{w,i}^{(a)})^*(\boldsymbol{P}_\ell \boldsymbol{u}_{w,j}^{(b)})\right)^*((\boldsymbol{v}_{w,i}^{(a)})^*(\boldsymbol{P}_{\ell-1}\boldsymbol{v}_{w,j}^{(b)})).$ □

## G.5 PROOF OF $|R(\boldsymbol{\theta}_a, \pi(\boldsymbol{\theta}_b))| \leq 1$

This subsection proves the following theorem:

**Theorem G.2.** *Let $\boldsymbol{\theta}_a$ and $\boldsymbol{\theta}_b$ be the parameters of two MLPs with L-layers. For any permutation $\pi = (\boldsymbol{P}_\ell)_{\ell \in [L]}$, we have*

$$|R(\boldsymbol{\theta}_a, \pi(\boldsymbol{\theta}_b))| = \frac{\left|\sum_{\ell,i,j}(\boldsymbol{u}_{\ell,i}^{(a)})^\top(\boldsymbol{P}_\ell \boldsymbol{u}_{\ell,j}^{(b)})(\boldsymbol{v}_{\ell,i}^{(a)})^\top(\boldsymbol{P}_{\ell-1}\boldsymbol{v}_{\ell,j}^{(b)})\right|}{\sum_\ell n_\ell} \leq 1. \tag{18}$$

*The equality holds if, for all $\ell$, $i$, $\boldsymbol{u}_{\ell,i}^{(a)} = \boldsymbol{P}_\ell \boldsymbol{u}_{\ell,i}^{(b)}$ and $\boldsymbol{v}_{\ell,i}^{(a)} = \boldsymbol{P}_{\ell-1}\boldsymbol{v}_{\ell,i}^{(b)}$.*

*Proof.* If, for all $\ell$, $i$, $\boldsymbol{u}_{\ell,i}^{(a)} = \boldsymbol{P}_\ell \boldsymbol{u}_{\ell,i}^{(b)}$ and $\boldsymbol{v}_{\ell,i}^{(a)} = \boldsymbol{P}_{\ell-1}\boldsymbol{v}_{\ell,i}^{(b)}$, then the equality obviously holds. Thus, we prove that Equation (18) holds. Using the property of trace, we have

$$\sum_{\ell,i,j}(\boldsymbol{u}_{\ell,i}^{(a)})^\top(\boldsymbol{P}_\ell \boldsymbol{u}_{\ell,j}^{(b)})(\boldsymbol{v}_{\ell,i}^{(a)})^\top(\boldsymbol{P}_{\ell-1}\boldsymbol{v}_{\ell,j}^{(b)}) = \sum_{\ell,i,j}(\boldsymbol{P}_\ell \boldsymbol{u}_{\ell,j}^{(b)})^\top(\boldsymbol{u}_{\ell,i}^{(a)})(\boldsymbol{v}_{\ell,i}^{(a)})^\top(\boldsymbol{P}_{\ell-1}\boldsymbol{v}_{\ell,j}^{(b)})$$

$$= \sum_\ell \operatorname{tr}\left(\sum_i (\boldsymbol{u}_{\ell,i}^{(a)})(\boldsymbol{v}_{\ell,i}^{(a)})^\top \sum_j \left((\boldsymbol{P}_\ell \boldsymbol{u}_{\ell,j}^{(b)})(\boldsymbol{P}_{\ell-1}\boldsymbol{v}_{\ell,j}^{(b)})^\top\right)^\top\right).$$

Let $\boldsymbol{W}_\ell'^{(a)} = \sum_i(\boldsymbol{u}_{\ell,i}^{(a)})(\boldsymbol{v}_{\ell,i}^{(a)})^\top$ and $\boldsymbol{W}_\ell'^{(b)} = \sum_j(\boldsymbol{P}_\ell \boldsymbol{u}_{\ell,j}^{(b)})(\boldsymbol{P}_{\ell-1}\boldsymbol{v}_{\ell,j}^{(b)})^\top$. Obviously, they are matrices with all singular values of 1, and thus by using von Neuman's trace inequality (von Neumann, 1962), we have

$$\sum_\ell \left|\operatorname{tr}\left(\boldsymbol{W}_\ell'^{(b)}(\boldsymbol{W}_\ell'^{(a)})^\top\right)\right| \leq \sum_\ell n_\ell.$$

Therefore, the triangle inequality yields that

$$\left|\sum_{\ell,i,j}(\boldsymbol{u}_{\ell,i}^{(a)})^\top(\boldsymbol{P}_\ell \boldsymbol{u}_{\ell,j}^{(b)})(\boldsymbol{v}_{\ell,i}^{(a)})^\top(\boldsymbol{P}_{\ell-1}\boldsymbol{v}_{\ell,j}^{(b)})\right| = \left|\sum_\ell \operatorname{tr}\left(\boldsymbol{W}_\ell'^{(b)}(\boldsymbol{W}_\ell'^{(a)})^\top\right)\right|$$

$$\leq \sum_\ell \left|\operatorname{tr}\left(\boldsymbol{W}_\ell'^{(b)}(\boldsymbol{W}_\ell'^{(a)})^\top\right)\right|$$

$$\leq \sum_\ell n_\ell,$$

which completes the proof. □

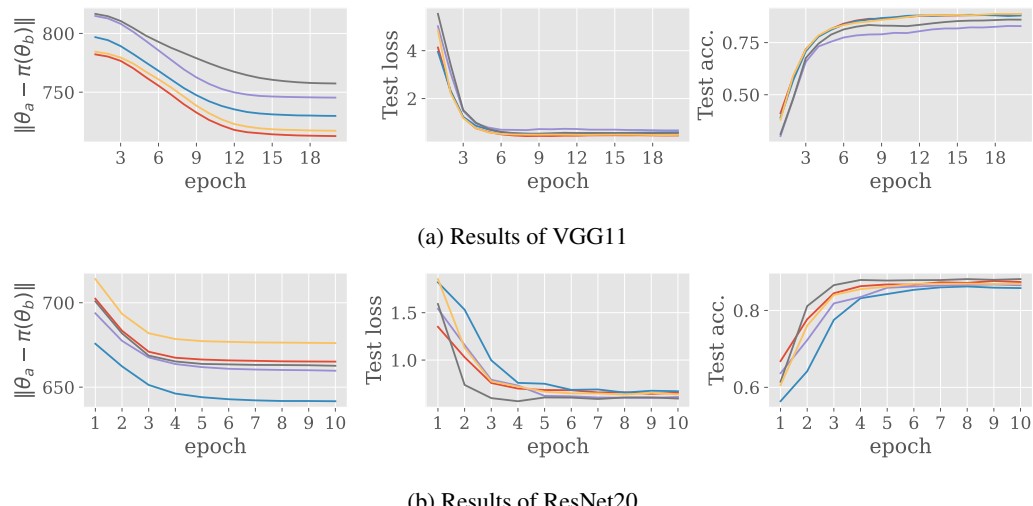

(a) Results of VGG11

(b) Results of ResNet20

Figure 6: $L^2$ distance between two models, test loss, and accuracy of merged models when optimizing permutations using Sinkhorn's algorithm for WM. Five permutation search trials were conducted with independently trained models (i.e., ten independently trained models were prepared to form five model pairs, and WM was performed for each pair). These results are plotted in different colors.

### G.6 PROOF OF THEOREM 4.2

*Proof.* Because $\sigma$ is Lipschitz continuous with the constant $C$, we have

$$\mathbb{E}\|\sigma(\boldsymbol{W}_\ell^{(a)}\boldsymbol{z}) - \sigma(\boldsymbol{W}_\ell^{(b)}\boldsymbol{z})\| \le C\mathbb{E}\|\boldsymbol{W}_\ell^{(a)}\boldsymbol{z} - \boldsymbol{W}_\ell^{(b)}\boldsymbol{z}\| \le C\sqrt{\mathbb{E}\|\boldsymbol{W}_\ell^{(a)}\boldsymbol{z} - \boldsymbol{W}_\ell^{(b)}\boldsymbol{z}\|^2}, \quad (19)$$

where we use Jensen's inequality since the squre root function is concave. Focusing on the difference between the outputs in the square root, we get

$$\|\boldsymbol{W}_\ell^{(a)}\boldsymbol{z} - \boldsymbol{W}_\ell^{(b)}\boldsymbol{z}\|^2 = \boldsymbol{z}^\top(\boldsymbol{W}_\ell^{(a)})^\top\boldsymbol{W}_\ell^{(a)}\boldsymbol{z} + \boldsymbol{z}^\top(\boldsymbol{W}_\ell^{(b)})^\top\boldsymbol{W}_\ell^{(b)}\boldsymbol{z} - 2\boldsymbol{z}^\top(\boldsymbol{W}_\ell^{(a)})^\top\boldsymbol{W}_\ell^{(b)}\boldsymbol{z}.$$

From the SVDs of weights $\boldsymbol{W}_\ell^{(a)} = \sum_i \boldsymbol{u}_{\ell,i}^{(a)} s_i^{(a)} \boldsymbol{v}_{\ell,i}^{(a)}$ and $\boldsymbol{W}_\ell^{(b)} = \sum_i \boldsymbol{u}_{\ell,i}^{(b)} s_i^{(b)} \boldsymbol{v}_{\ell,i}^{(b)}$, we have $\boldsymbol{z}^\top(\boldsymbol{W}_\ell^{(a)})^\top\boldsymbol{W}_\ell^{(a)}\boldsymbol{z} = \sum_i (s_i^{(a)})^2(\boldsymbol{v}_{\ell,i}^{(a)}\boldsymbol{z})^2$, $\boldsymbol{z}^\top(\boldsymbol{W}_\ell^{(b)})^\top\boldsymbol{W}_\ell^{(b)}\boldsymbol{z} = \sum_i (s_i^{(b)})^2(\boldsymbol{v}_{\ell,i}^{(b)}\boldsymbol{z})^2$, and $\boldsymbol{z}^\top(\boldsymbol{W}_\ell^{(a)})^\top\boldsymbol{W}_\ell^{(b)}\boldsymbol{z} = \sum_{i,j} s_i^{(a)} s_j^{(b)}(\boldsymbol{u}_{\ell,i}^{(a)})^\top\boldsymbol{u}_{\ell,j}^{(b)}(\boldsymbol{v}_{\ell,i}^{(a)})^\top\boldsymbol{z}(\boldsymbol{v}_{\ell,j}^{(b)})^\top\boldsymbol{z}$. Therefore, Equation (19) can be rewritten as

$$\mathbb{E}\|\sigma(\boldsymbol{W}_\ell^{(a)}\boldsymbol{z}) - \sigma(\boldsymbol{W}_\ell^{(b)}\boldsymbol{z})\|$$
$$\le C\sqrt{\sum_i (s_{\ell,i}^{(a)})^2\mathbb{E}((\boldsymbol{v}_{\ell,i}^{(a)})^\top\boldsymbol{z})^2 + \sum_i (s_{\ell,i}^{(b)})^2\mathbb{E}((\boldsymbol{v}_{\ell,i}^{(b)})^\top\boldsymbol{z})^2 - 2\sum_{i,j} s_{\ell,i}^{(a)} s_{\ell,j}^{(b)}(\boldsymbol{u}_{\ell,i}^{(a)})^\top\boldsymbol{u}_{\ell,j}^{(b)}\mathbb{E}(\boldsymbol{v}_{\ell,i}^{(a)})^\top\boldsymbol{z}(\boldsymbol{v}_{\ell,j}^{(b)})^\top\boldsymbol{z}}.$$

$\square$

## H ADDITIONAL EXPERIMENTAL RESULTS

### H.1 LEARNING CURVE OF WM

In this subsection, Figure 6 shows the learning curves for VGG11 and ResNet20 when WM is performed using Sinkhorn's algorithm. The figure shows that the distance between the two models decreases as the training progresses, and the performance of the merged model also improves. In this paper, for both VGG11 and ResNet20, we used permutations at the 10th epoch, when the loss of the merged model is stably small.

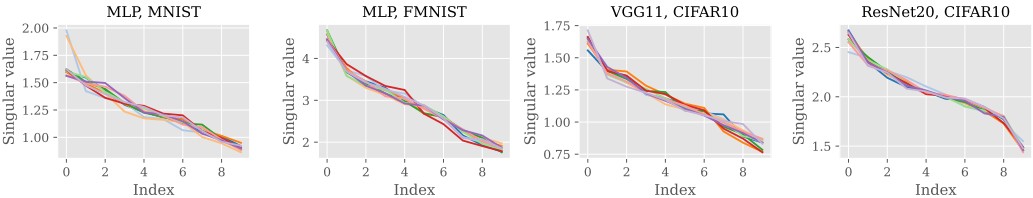

(a) MLP, MNIST.

(b) MLP, FMNIST.

(c) VGG11, CIFAR10.

(d) ResNet20, CIFAR10.

Figure 7: Distributions of the singular values of each layer.

Figure 8: Adjusted distributions of the singular values of the output layer.

## H.2 Distribution of Singular Values

Figure 7 shows the distributions of the singular values of all the layers. The figure demonstrates that, in all layers except for the output layer, the singular values are very similar across all models. Meanwhile, in the output layer, there is variability in the singular values. However, this variability does not affect the accuracy of the merged model. Let $\boldsymbol{W}_L^{(a)} = \sum_i s_{L,i}^{(a)} \boldsymbol{u}_{L,i}^{(a)} (\boldsymbol{v}^{(a)})_{L,i}^\top$ and $\boldsymbol{W}_L^{(b)} = \sum_i s_{L,i}^{(b)} \boldsymbol{u}_{L,i}^{(b)} (\boldsymbol{v}_{L,i}^{(b)})^\top$ represent the output layer weights of the two trained models. The figure shows that the difference between the singular values of the two models is approximately a constant multiple. In other words, there exists a constant $\alpha$ such that $s_{L,i}^{(a)} \approx \alpha s_{L,i}^{(b)}$ for all $i$. To confirm this, Figure 8 shows the distribution of singular values when the constant $\alpha$ is calculated and the weight of the output layer is adjusted, demonstrating that correcting the output layer by a constant factor can address the differences in the distribution. If the singular vectors of the two weights are equal (i.e., $\boldsymbol{v}_{L,i}^{(a)} = \boldsymbol{v}_{L,i}^{(b)}$ and $\boldsymbol{u}_{L,i}^{(a)} = \boldsymbol{u}_{L,i}^{(b)}$ for all $i$), then $\boldsymbol{W}_L^{(a)} \approx \alpha \boldsymbol{W}_L^{(b)}$ holds (indeed, as mentioned in Section 4.3, the permutation matrix aligns the directions of the singular vectors). Therefore, the weight of the output layer of the merged model at the ratio $\lambda \in [0,1]$ is given by $\lambda \boldsymbol{W}_L^{(a)} + (1-\lambda)\boldsymbol{W}_L^{(b)} \approx \lambda \boldsymbol{W}_L^{(a)} + (1-\lambda)\alpha \boldsymbol{W}_L^{(a)} = (\lambda + (1-\lambda)\alpha)\boldsymbol{W}_L^{(a)}$. Thus, we can consider that the weight and the activation function of the merged model are given by $\boldsymbol{W}_L^{(a)}$ and a softmax function with an inverse temperature of $1/(\lambda + (1-\lambda)\alpha)$, respectively. Since the inverse temperature does not affect the accuracy value, the difference in the singular values of the output layer would not matter in satisfying LMC, at least in terms of accuracy.

## H.3 Inner Products Between Right Singular Vectors of Hidden Layers And Their Input

Figure 9 shows the average absolute values of inner products between the right singular vectors and the input in each layer of models trained by SGD. The figure demonstrates that, except for the input and output layers, the singular vectors with larger singular values generally have larger inner products with the inputs. Note that the results of the input layers do not affect the permutation search based on WM because their right singular vectors are not changed by the permutations.

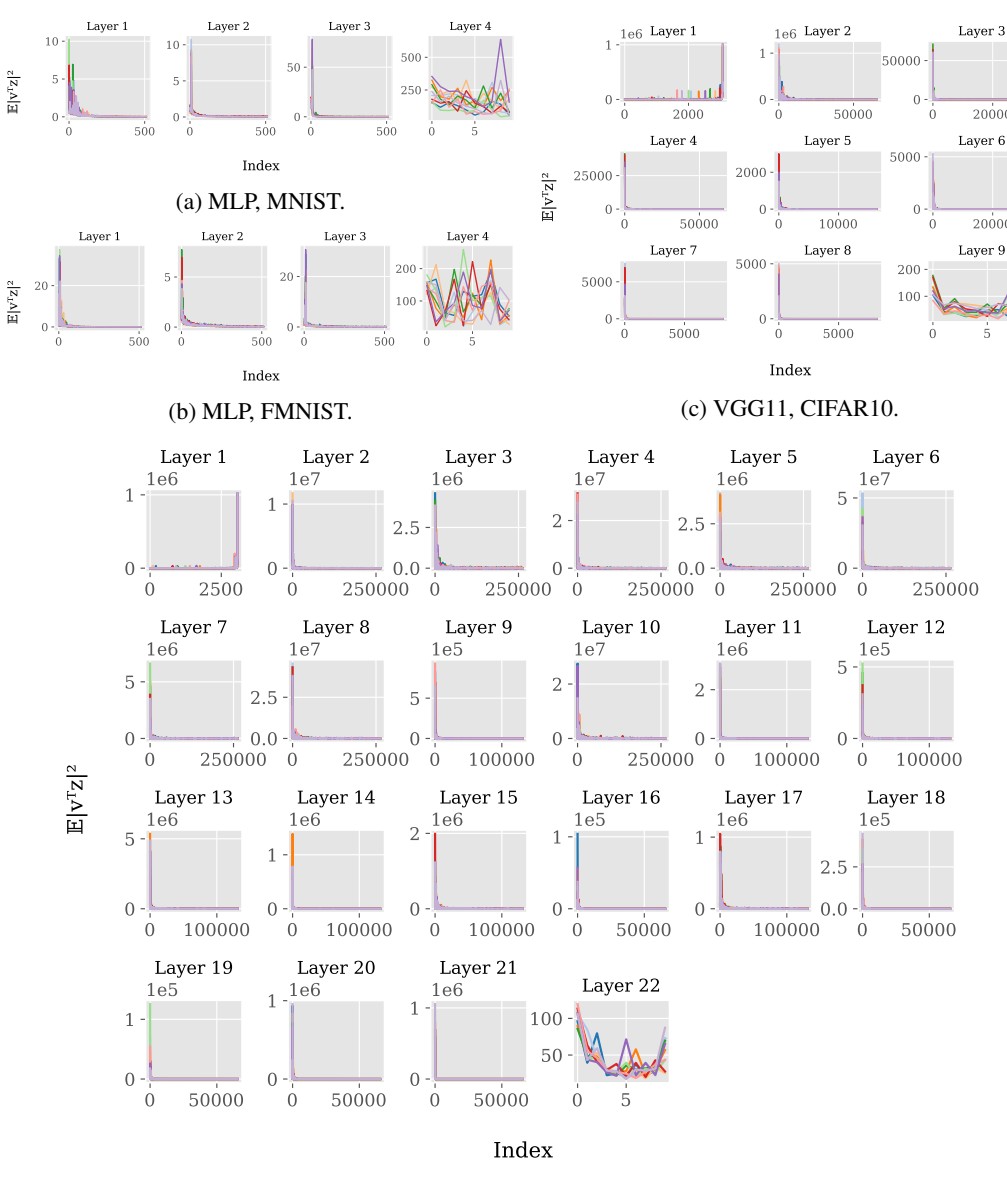

Figure 9: Average absolute values of inner products between the right singular vectors and the input of each layer. The horizontal axis represents the index of the left singular vector, while the vertical axis shows the mean square of the inner product. The left side of the horizontal axis corresponds to singular vectors with large singular values.

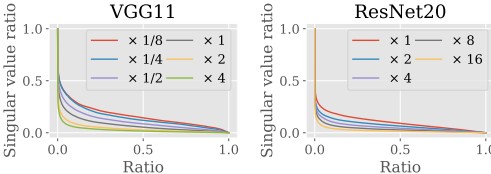

Figure 10: Distribution of all singular values normalized by the largest one in the model as the model width multiplier changes. The vertical axis represents the singular values divided by the maximum singular value of each model, and the horizontal axis represents the ratio among all singular values (e.g., the point at 0.5 on the horizontal axis represents the singular value in the middle of all values sorted in descending order).

### H.4 RELATIONSHIP WITH MODEL WIDTH

Previous studies have demonstrated that the width of the model architecture affects the ease of achieving LMC. In this subsection, we explain this phenomenon based on the following three facts: as the model width increases, (i) the proportion of dominant singular values decreases, (ii) the right singular vectors corresponding to these dominant singular values will have large inner product values with the inputs of the hidden layers, and (iii) the WM preferentially aligns the directions of singular vectors corresponding to these dominant singular values.

**(i) Dependency of model width on singular values.** As we mentioned, the proportion of relatively large singular values in all singular values decreases as the model width increases. To verify this, Figure 10 shows the distribution of the singular values of all layers of VGG11 and ResNet20 trained on CIFAR10. Figure 10 shows the results of different model widths (i.e., dimensionality). As can be seen, the proportion of relatively large singular values decreases as the model width increases. Thus, the proportion of singular vectors that need to be aligned in the model decreases as the width increases.

**(ii) Dependency of model width on inner products of right singular vectors.** We also investigated the effect of model width on the inner products between the hidden layer inputs and the right singular vectors. Figures 11 and 12 show the values of these inner products for each layer as model width changes. Figures 11 and 12 show the distributions of inner products for VGG11 and ResNet20 models trained on CIFAR10, respectively. These figures demonstrate that as model width increases, the inner products between the right singular vectors with large singular values and the inputs also increase.

**(iii) Singular-vector alignment.** We conducted an experiment to examine how well the directions of singular vectors are aligned as model width increases when applying permutations found by WM. The results are shown in Figure 13. The figures display the evaluation of $R(\boldsymbol{\theta}_a, \pi(\boldsymbol{\theta}_b))$ for the trained models $\boldsymbol{\theta}_a$ and $\boldsymbol{\theta}_b$ by searching for permutations $\pi$. For comparison, the case where no permutations are applied (i.e., $\pi$ is an identity map) is also shown. Additionally, a threshold $\gamma$ was introduced to assess the alignment of singular vectors with large singular values.

First, focusing on the results in Figure 13(a) with $\gamma = 0$, we observe that the value of $R$ decreases even when the width increases and WM is used. Conversely, Figure 13(b) shows that the directions of singular vectors with particularly large singular values are aligned by permutation as model width increases. This suggests that even with WM, it is difficult to perfectly align the directions of singular vectors between the two models. However, increasing the width decreases the fraction of singular vectors with large singular values, thus making it easier for WM to align the directions of these dominant singular vectors.

As shown in Figure 10, when the model is sufficiently wide, the proportion of large singular values is very small compared to the total number of singular values. Furthermore, Figures 11 and 12 demonstrate that the right singular vectors associated with these relatively large singular values have a large inner product with the hidden layer input. This means that the number of singular vectors that WM needs to align to achieve LMC is reduced when the model is wide enough, as discussed in

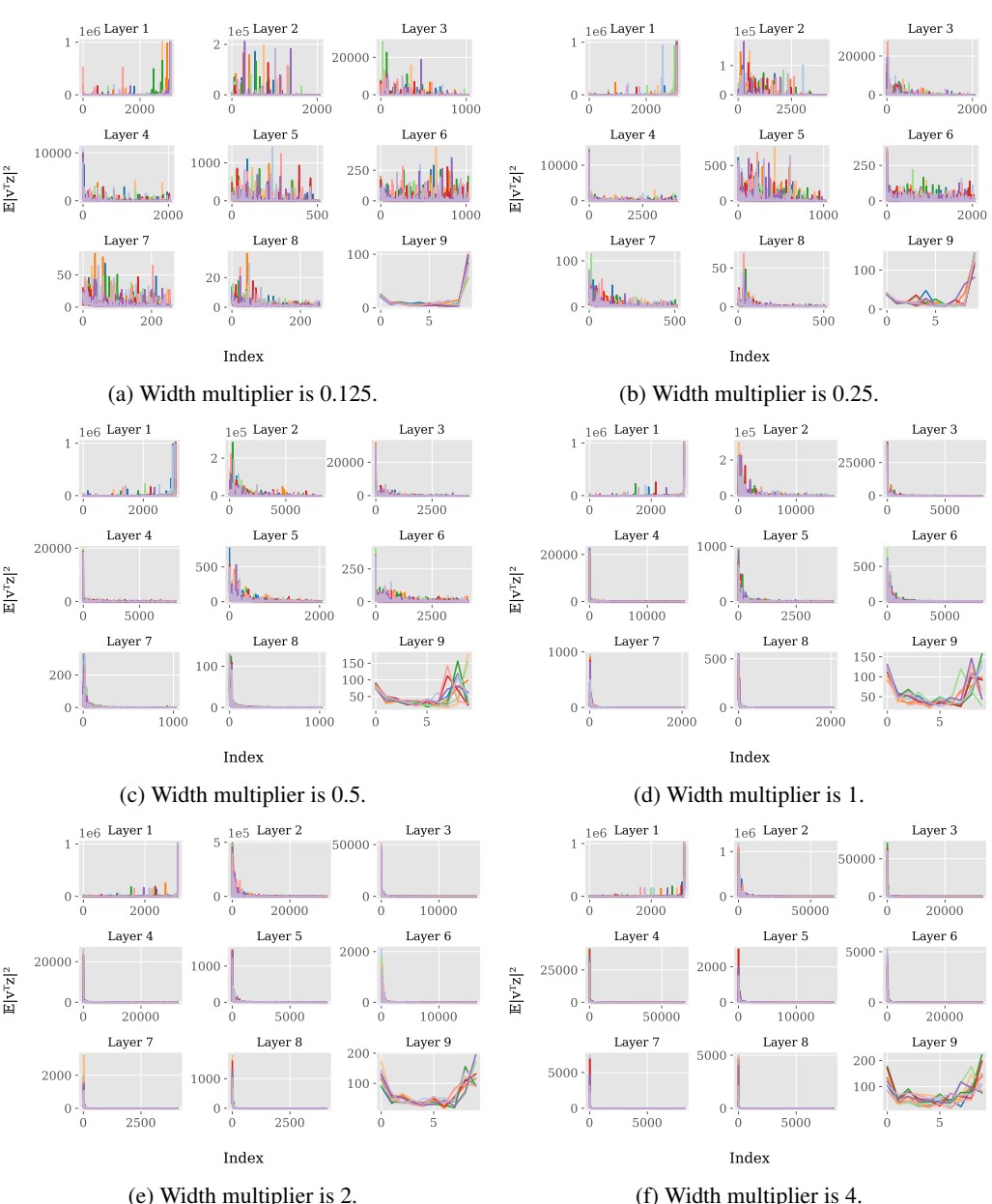

Figure 11: Average absolute values of inner products between the right singular vectors and the input of each layer of VGG11 trained on CIFAR10.

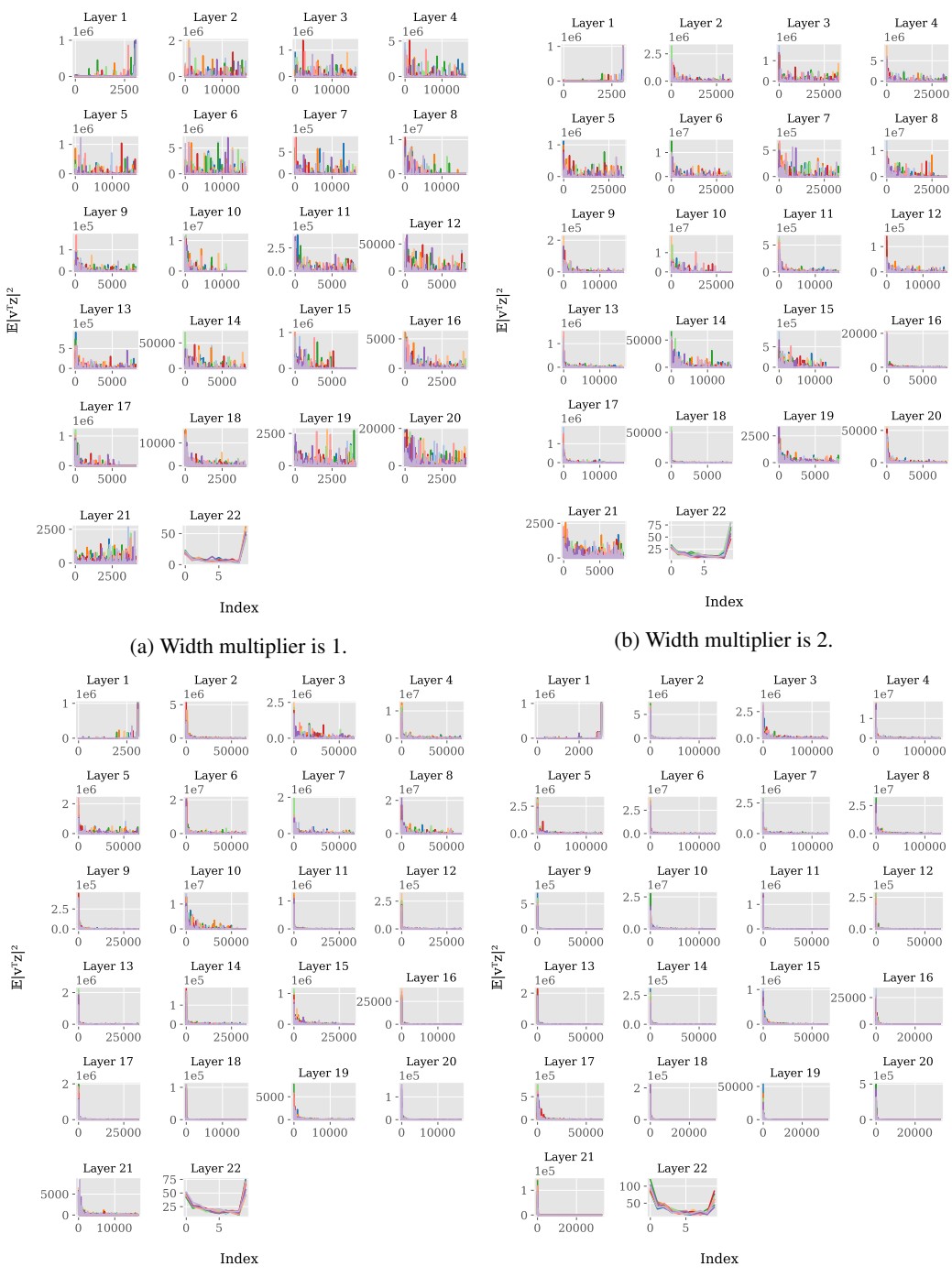

(a) Width multiplier is 1.

(b) Width multiplier is 2.

(c) Width multiplier is 4.

(d) Width multiplier is 8.

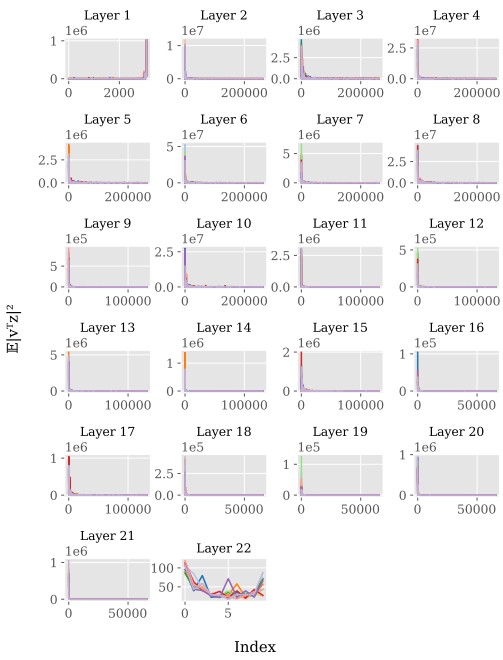

(e) Width multiplier is 16.

Figure 12: Average absolute values of inner products between the right singular vectors and the input of each layer of ResNet20 trained on CIFAR10.

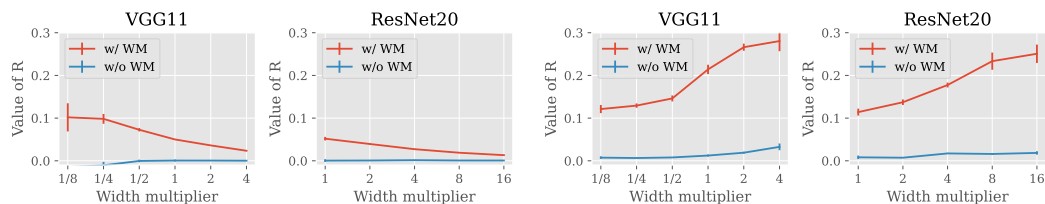

(a) Evaluation results of $R(\boldsymbol{\theta}_a, \pi(\boldsymbol{\theta}_b))$ for all the singular vectors. (b) Evaluation results of $R(\boldsymbol{\theta}_a, \pi(\boldsymbol{\theta}_b))$ for the singular vectors with $\gamma = 0.3$.

Figure 13: Relation between the model width and the difficulty in aligning the directions of singular vectors.

Section 4.4. Indeed, Figure 13(b) suggests that WM preferentially aligns these significant singular vectors. Therefore, increasing the width is expected to make LMC more feasible.

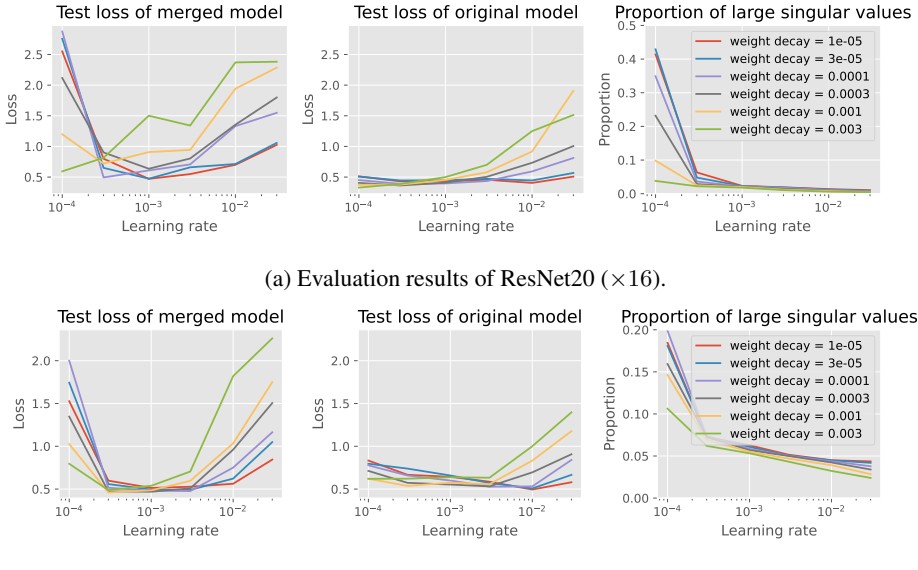

(a) Evaluation results of ResNet20 ($\times 16$).

(b) Evaluation results of VGG11 ($\times 2$).

Figure 14: Experimental results of model merging with WM under different learning rates and weight decay strength.

## H.5 DEPENDENCY OF WEIGHT DECAY AND LEARNING RATE

Qu & Horvath (2024) have observed that strengthening weight decay and increasing the learning rate make it easier for LMC to be established through WM. In this section, we will explain this observation from the perspective of singular values of weights.

In Section 4, we stated that the reason why WM can establish LMC is that the permutation found by WM aligns singular vectors with large singular values between two models. In other words, the smaller the proportion of large singular values in the weights of each layer of the models, the more likely LMC is to be established by WM. In fact, some previous studies (Galanti et al., 2022; Timor et al., 2023) have shown that increasing the weight decay and learning rate during model training can reduce the ranks of the weights in the trained model. Therefore, it is highly likely that the results observed by Qu & Horvath (2024) were caused by the reduction in the ranks of the weights. In the following, we will experimentally confirm this prediction.

Figure 14 shows the experimental results of ResNet20 and VGG11 models. During model training, the learning rate was varied from $0.0001, 0.0003, \ldots, 0.03$ and the weight decay strength was varied from $0.00001, 0.00003, \ldots, 0.003$. For each condition, six models were trained, and model merging was performed three times by creating three pairs from them. The conditions for model training were the same as in Appendix D, except for the learning rate and weight decay. For VGG11, when the model width is quadrupled, the ratio of large singular values becomes very small regardless of the learning rate or weight decay, making it difficult to understand the relationship between the loss of the merged model and large singular values. Thus, the model width was doubled for VGG11 models. In addition, the permutation search method used in WM was based on the method of Ainsworth et al. (2023).

Figure 14 shows the test losses of the merged model and the original model, as well as the ratio of the number of large singular values to the width of the original model. Figure 14 displays the averaged results over three runs of model merging. The ratio in the figure was calculated as follows. Let $s_{\ell,1}, s_{\ell,2}, \ldots, s_{\ell,n_\ell}$ be the singular values of the $\ell$-th layer, where $n_\ell$ represents the number of singular values, and $s_{\ell,1}$ is the largest singular value. Each singular value is divided by the largest singular value, and those whose ratio is 0.3 or more are counted (i.e., $s_{\ell,i}/s_{\ell,1} \geq 0.3$ for $i \in [n_\ell]$). Next, for all layers, we calculate the sum of these numbers and divide it by the sum of $n_\ell$ for all layers. This procedure can be written as $\sum_{\ell,i} I[s_{\ell,i}/s_{\ell,1} \geq 0.3] / \sum_\ell n_\ell$, where $I$ is an indicator function. In

Table 4: Model Merging results with **AM** and the estimated barrier value using Taylor approximation when $\lambda = 1/2$

| Dataset | Network | Test acc. | Barrier ($\lambda = 1/2$) | Taylor approx. | $\|\boldsymbol{\theta}_a - \boldsymbol{\theta}_b\|$ | $\|\boldsymbol{\theta}_a - \pi(\boldsymbol{\theta}_b)\|$ | Reduction rate [%] |
|---|---|---|---|---|---|---|---|
| CIFAR10 | VGG11 | $88.786 \pm 0.186$ | $0.077 \pm 0.044$ | $2.491 \pm 0.266$ | $799.503 \pm 16.396$ | $742.300 \pm 18.526$ | $7.161 \pm 0.572$ |
| | ResNet20 | $89.190 \pm 0.192$ | $0.189 \pm 0.031$ | $7.431 \pm 0.667$ | $710.762 \pm 16.261$ | $671.373 \pm 13.816$ | $5.538 \pm 0.226$ |
| FMNIST | MLP | $88.356 \pm 0.221$ | $-0.236 \pm 0.024$ | $0.948 \pm 0.173$ | $121.853 \pm 5.830$ | $108.968 \pm 5.365$ | $10.578 \pm 0.343$ |
| MNIST | MLP | $98.274 \pm 0.084$ | $-0.020 \pm 0.004$ | $0.064 \pm 0.035$ | $81.231 \pm 5.580$ | $68.722 \pm 5.197$ | $15.428 \pm 1.037$ |

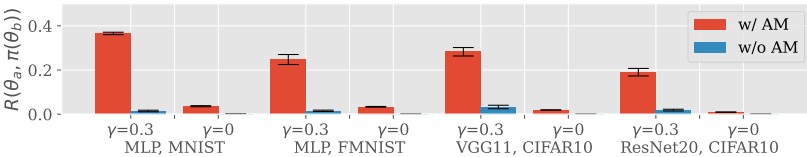

Figure 15: Evaluation results of $R(\boldsymbol{\theta}_a, \boldsymbol{\theta}_b)$ **with and without AM**.

other words, this ratio becomes smaller when the number of singular values that are relatively large compared to the maximum singular value is small compared to the width of the model.

From Figure 14, we can see that the ratio of large singular values decreases as both the weight decay and the learning rate increase. This has already been noted in previous research (Galanti et al., 2022; Timor et al., 2023). Additionally, Figure 14 shows that the test loss of the merged model decreases when both the test loss of the original model and the ratio of large singular values are small. This can be explained by our analysis in Section 4. As we described, WM facilitates LMC by aligning the singular vectors between the two models and making the functions of the middle layers of the merged model and the original model more similar. In other words, this suggests that it is challenging for the merged model to outperform the original model. Furthermore, the smaller the ratio of large singular values, the easier it is to align the singular vectors using WM, so the functions of the merged model and the original model become more closely aligned. From this, we conclude that the higher the performance of the original model and the smaller the ratio of large singular values, the better the performance of the merged model. This is consistent with the results shown in Figure 14.

### H.6 ACTIVATION MATCHING

Table 4 shows the results of model merging using AM under the same conditions as in Section 3. The table indicates that the loss barrier is sufficiently small when AM is applied. Interestingly, AM reduces the distance between the two models to the same extent as WM.

Figure 15 shows the value of $R$ between the two models $\boldsymbol{\theta}_a$ and $\boldsymbol{\theta}_b$. The figure clearly shows that the value of $R$ is larger using AM when $\gamma = 0.3$. This indicates that AM aligns the directions of singular vectors with large singular values for the two models, similar to the result of WM (Figure 2). To further examine the relationship of singular vectors between the models before and after merging, Figure 16 shows the values of $R$ between these models. This result also demonstrates a similar trend to the WM results shown in Figure 3, suggesting that the reasons for the establishment of the LMC are almost the same for both WM and AM.

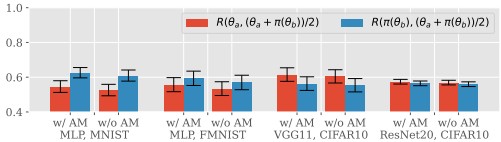 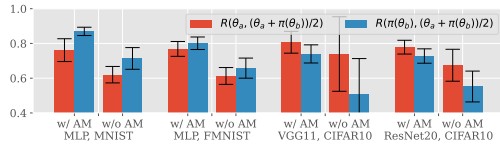

(a) Evaluation results with the threshold $\gamma = 0$.      (b) Evaluation results with the threshold $\gamma = 0.3$.

Figure 16: Evaluation results of $R$ value between the merged model and the pre-merged models (i.e., $R(\boldsymbol{\theta}_a, (\boldsymbol{\theta}_a + \pi(\boldsymbol{\theta}_b))/2)$ and $R(\boldsymbol{\theta}_a, (\boldsymbol{\theta}_a + \pi(\boldsymbol{\theta}_b))/2)$) when **AM is used**. The blue and red bars represent the evaluation results of $R(\boldsymbol{\theta}_a, (\boldsymbol{\theta}_a + \pi(\boldsymbol{\theta}_b))/2)$ and $R(\boldsymbol{\theta}_b, (\boldsymbol{\theta}_a + \pi(\boldsymbol{\theta}_b))/2)$, respectively.

Table 5: Evaluation results of barrier between each pair of models

| | Dataset | Network | Loss barrier $(\boldsymbol{\theta}_a + \pi_b(\boldsymbol{\theta}_b))/2$ | $(\boldsymbol{\theta}_a + \pi_c(\boldsymbol{\theta}_c))/2$ | $(\pi_b(\boldsymbol{\theta}_b) + \pi_c(\boldsymbol{\theta}_c))/2$ | Accuracy barrier $(\boldsymbol{\theta}_a + \pi_b(\boldsymbol{\theta}_b))/2$ | $(\boldsymbol{\theta}_a + \pi_c(\boldsymbol{\theta}_c))/2$ | $(\pi_b(\boldsymbol{\theta}_b) + \pi_c(\boldsymbol{\theta}_c))/2$ |
|---|---|---|---|---|---|---|---|---|
| WM | CIFAR10 | VGG11 | $0.094 \pm 0.158$ | $0.037 \pm 0.156$ | $0.141 \pm 0.141$ | $8.362 \pm 5.677$ | $7.555 \pm 4.978$ | $10.12 \pm 5.117$ |
| | | ResNet20 | $0.135 \pm 0.026$ | $0.098 \pm 0.011$ | $0.294 \pm 0.098$ | $3.312 \pm 0.61$ | $2.995 \pm 0.064$ | $7.23 \pm 0.99$ |
| | FMNIST | MLP | $-0.211 \pm 0.029$ | $-0.174 \pm 0.044$ | $-0.174 \pm 0.051$ | $1.947 \pm 0.501$ | $1.703 \pm 0.289$ | $4.337 \pm 1.434$ |
| | MNIST | MLP | $-0.027 \pm 0.005$ | $-0.034 \pm 0.003$ | $-0.031 \pm 0.003$ | $0.173 \pm 0.04$ | $0.198 \pm 0.032$ | $0.475 \pm 0.069$ |
| STE | CIFAR10 | VGG11 | $0.081 \pm 0.031$ | $0.099 \pm 0.042$ | $2.172 \pm 0.989$ | $4.86 \pm 0.815$ | $5.76 \pm 0.537$ | $32.013 \pm 8.193$ |
| | | ResNet20 | $0.466 \pm 0.154$ | $0.446 \pm 0.138$ | $1.693 \pm 0.168$ | $15.005 \pm 3.765$ | $13.942 \pm 4.008$ | $34.483 \pm 2.426$ |
| | FMNIST | MLP | $-0.372 \pm 0.016$ | $-0.343 \pm 0.055$ | $0.023 \pm 0.118$ | $2.667 \pm 0.248$ | $2.483 \pm 0.621$ | $15.97 \pm 1.724$ |
| | MNIST | MLP | $-0.037 \pm 0.011$ | $-0.039 \pm 0.006$ | $0.017 \pm 0.014$ | $0.253 \pm 0.176$ | $0.358 \pm 0.198$ | $2.312 \pm 0.457$ |

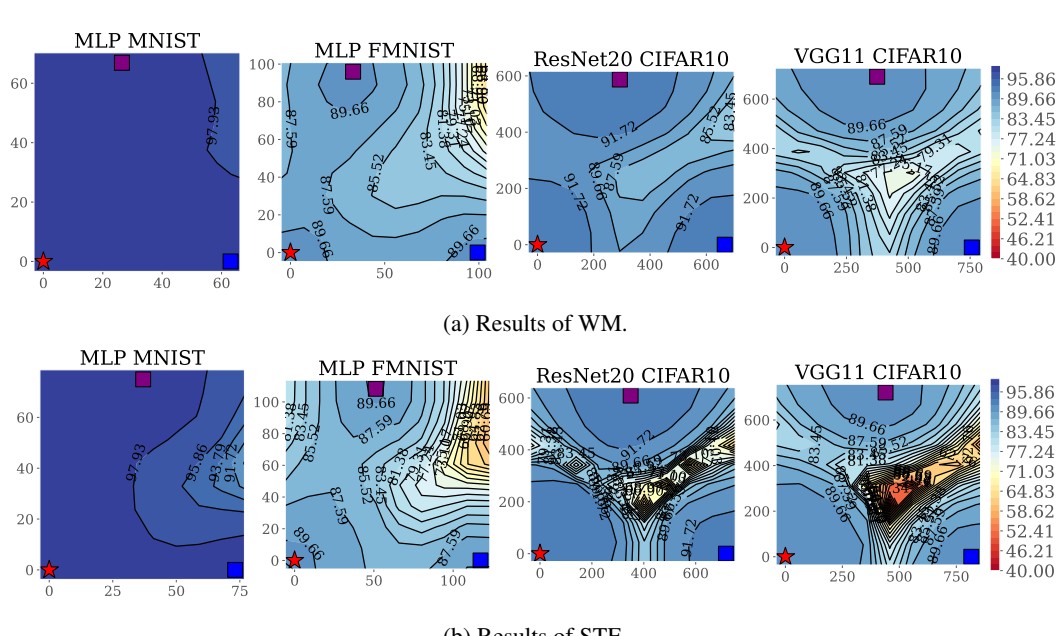

(a) Results of WM.

(b) Results of STE.

Figure 17: Accuracy landscape around $\boldsymbol{\theta}_a$, $\pi_b(\boldsymbol{\theta}_b)$ and $\pi_c(\boldsymbol{\theta}_c)$. The star in the lower left represents $\boldsymbol{\theta}_a$, and the squares in the lower right and upper represent $\pi_b(\boldsymbol{\theta}_b)$, and $\pi_c(\boldsymbol{\theta}_c)$, respectively.

## H.7 STE AND WM

In this subsection, additional experimental results for Section 6.3 are shown in Table 5 for the barrier values between each pair of models. The table shows the model-merging results with $\lambda = 1/2$, and the mean and standard deviation of three model merges. In the table, a negative value for the barrier indicates an improvement in performance due to the merging. In addition, Figure 17 shows the accuracy landscape around $\boldsymbol{\theta}_a$, $\pi_b(\boldsymbol{\theta}_b)$, and $\pi_c(\boldsymbol{\theta}_c)$. From Table 5 and Figure 17, we can see that the barrier between $\pi_b(\boldsymbol{\theta}_b)$ and $\pi_c(\boldsymbol{\theta}_c)$ is also smaller for WM than for STE.

## H.8 DEPENDENCY OF $R$ ON THRESHOLD $\gamma$

Figure 18 shows the value of $R$ when the threshold $\gamma$ is varied. Figure 18(a) displays the $R$ value between model $\boldsymbol{\theta}_a$ and the permuted model $\pi(\boldsymbol{\theta}_b)$, along with the $R$ value before permutation for comparison. Figure 18(b) illustrates the $R$ values between the merged model $(\boldsymbol{\theta}_a + \pi(\boldsymbol{\theta}_b))/2$ and the original models $\boldsymbol{\theta}_a$ and $\boldsymbol{\theta}_b$, also including the $R$ value without permutation for comparison.

In Figure 18(a), the $R$ value between the original models is nearly zero regardless of the $\gamma$ value. However, when permutation is applied, the $R$ value increases as $\gamma$ increases. This indicates that WM preferentially aligns the directions of the larger singular vectors between models $\boldsymbol{\theta}_a$ and $\boldsymbol{\theta}_b$. As shown in Figure 18(b), this effect helps align the singular vectors with larger singular values between the merged and original models. Aligning these singular vectors more closely makes LMC more feasible because the outputs between the two models are closer, as discussed in Section 4.4.

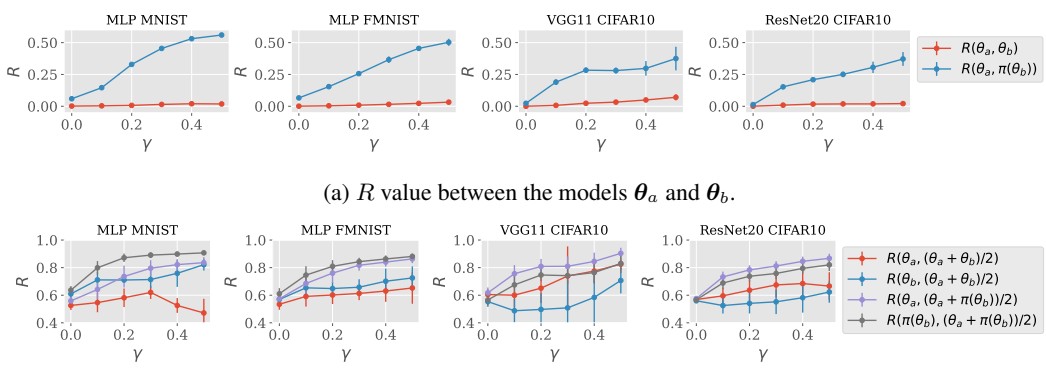

(a) $R$ value between the models $\boldsymbol{\theta}_a$ and $\boldsymbol{\theta}_b$.

(b) $R$ value between the merged model and the original model.

Figure 18: $R$ values when the threshold $\gamma$ is changed.

Table 6: Results of model merging of ResNet50 models trained on ImageNet dataset.

(a) Test loss and top-1 accuracy of each model.

| | $\boldsymbol{\theta}_c$ w/ WM | $\boldsymbol{\theta}_c$ w/o WM | $\boldsymbol{\theta}_a$ | $\boldsymbol{\theta}_b$ |
|---|---|---|---|---|
| Test loss | $5.207 \pm 0.073$ | $6.897 \pm 0.001$ | $1.491 \pm 0.011$ | $1.493 \pm 0.007$ |
| Test acc. | $40.239 \pm 2.088$ | $0.179 \pm 0.020$ | $75.741 \pm 2.088$ | $75.856 \pm 0.107$ |

(b) $L^2$ distance between $\boldsymbol{\theta}_a$ and $\boldsymbol{\theta}_b$.

| $L^2$ dist. w/ WM | $L^2$ dist. w/o WM |
|---|---|
| $126.823 \pm 0.533$ | $174.247 \pm 0.577$ |

## H.9 LMC ON RESNET50 TRAINED ON IMAGENET

In the paper, most of the analysis was performed on relatively small datasets such as MNIST and CIFAR10. In this subsection, we train ResNet50 models on a larger dataset, ImageNet, and analyze the results of model merging based on WM.

**Experimental results of model merging.** Table 6 presents the results of merging ResNet50 models trained on the ImageNet dataset. Table 6(a) shows the test loss and top-1 accuracy of the models before and after merging (i.e., the pre-merged models $\boldsymbol{\theta}_a$ and $\boldsymbol{\theta}_b$, and the merged model $\boldsymbol{\theta}_c$). Table 6(b) shows the $L^2$ distance between models $\boldsymbol{\theta}_a$ and $\boldsymbol{\theta}_b$ before and after applying permutations. These tables report the mean and standard deviation for five independent model merges. According to Table 6(a), the test loss and accuracy of the merged model are clearly improved by using WM. However, they are still worse than those of the pre-merged models $\boldsymbol{\theta}_a$ and $\boldsymbol{\theta}_b$, indicating that the LMC cannot be considered satisfied. Table 6(b) demonstrates that using WM decreases the $L^2$ distance between models $\boldsymbol{\theta}_a$ and $\boldsymbol{\theta}_b$. This decrease in $L^2$ distance is larger than that observed for VGG11 and ResNet20 in Table 1, suggesting that the singular vectors of models $\boldsymbol{\theta}_a$ and $\boldsymbol{\theta}_b$ are better aligned by permutations. Figure 19 presents the results of evaluating $R$ for each model pair. When $\gamma = 0.3$, Figure 19 shows that $R(\boldsymbol{\theta}_a, \boldsymbol{\theta}_b)$ rises to about 0.5 with WM, and the value of $R$ increases to approximately 0.9 between the pre- and post-merged models (i.e., $R(\boldsymbol{\theta}_a, \boldsymbol{\theta}_c)$ and $R(\boldsymbol{\theta}_b, \boldsymbol{\theta}_c)$). Thus, even with ResNet50, the singular vectors with large singular values are aligned between the pre- and post-merged models by using WM.

To investigate why LMC does not hold even though the singular vectors with large singular values are aligned by using WM, we examine the distributions of singular values at each layer and the inner products between the right singular vectors and the inputs. Figure 20 shows the distribution of singular values for each layer, and Figure 21 shows the distribution of the average absolute values of the inner products between the right singular vectors and the inputs for each layer. Each figure is plotted in a different color for the 10 trained models. Figure 20 demonstrates that the distributions of singular values are nearly identical across all models. Thus, the difference in singular values between models is not a reason for preventing LMC. In Figure 21, focusing on the distribution of the inner product between the right singular vectors and the inputs, we observe that the right singular vectors with large singular values do not necessarily have a large inner product with the input. As discussed in Section 4.4, WM can only align singular vectors with large singular values, so this discrepancy can be a reason preventing the establishment of LMC. As shown in Figures 11 and 12, the wider

Figure 19: Evaluation results of $R$ values between each pair of ResNet50 models trained on ImageNet dataset.

the model, the larger the inner product of the input and the right singular vectors with large singular values in the hidden layers. Therefore, it is considered necessary to increase the width of the model to establish LMC with ResNet50.

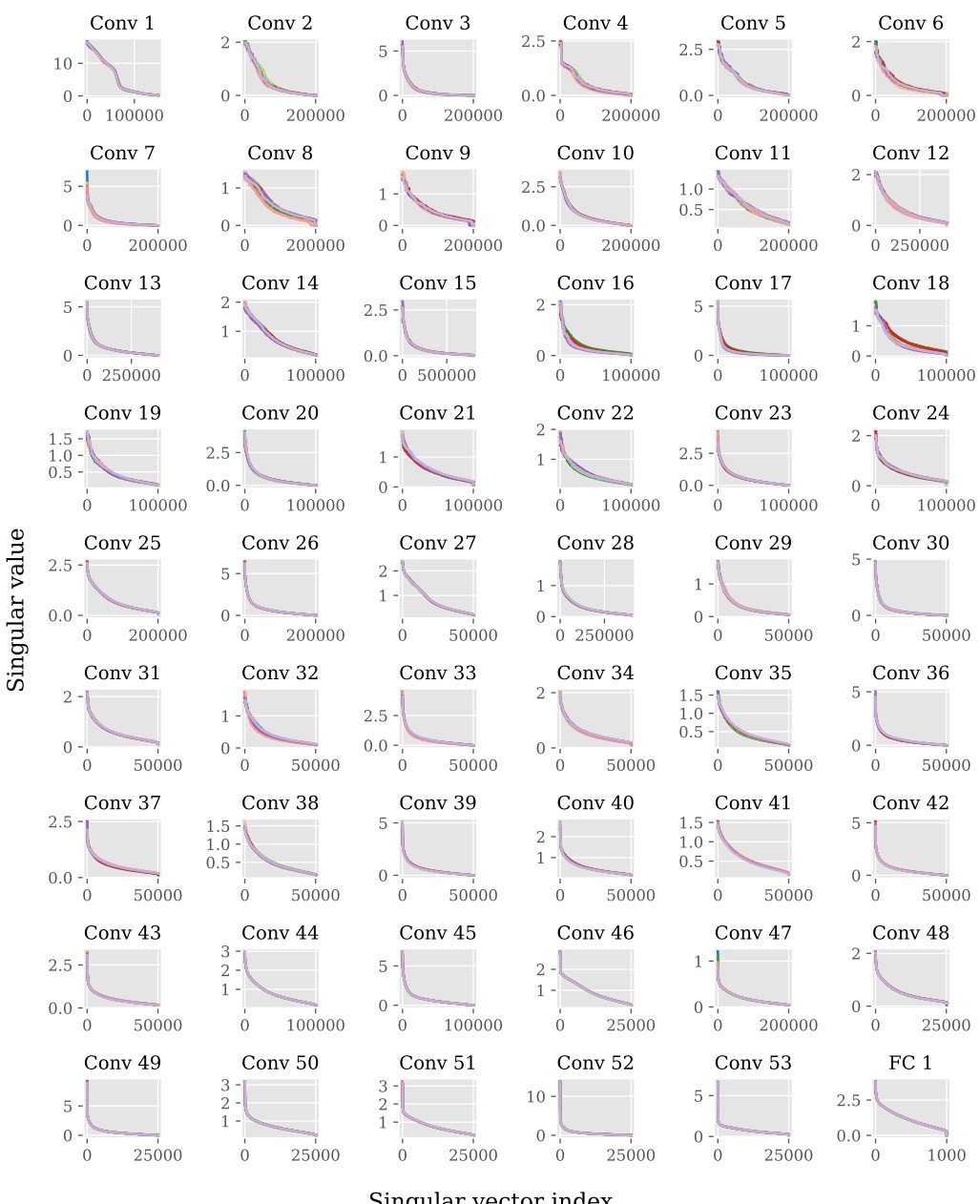

Figure 20: Distributions of the singular values of each layer of ResNet50 models trained on ImageNet dataset.

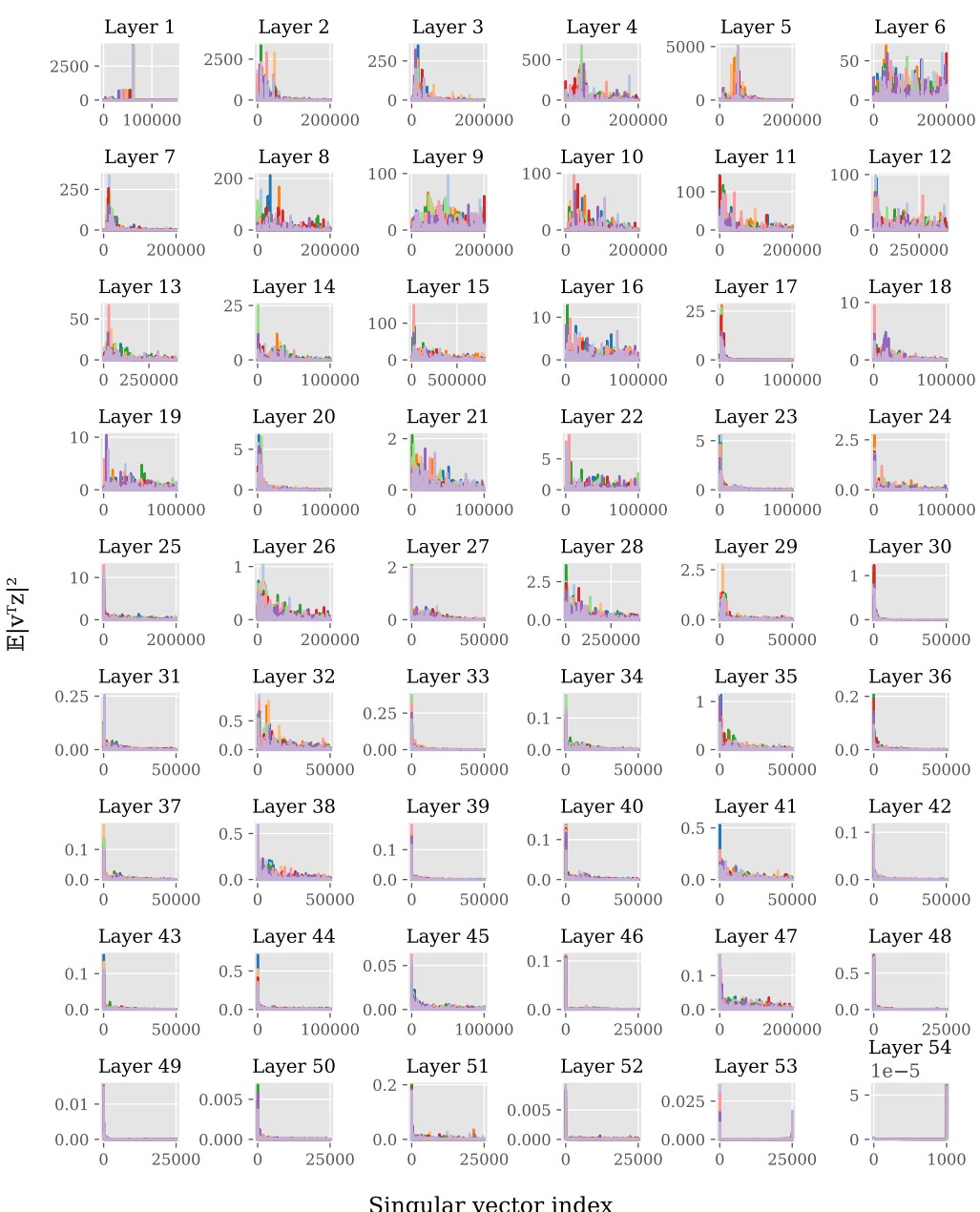

Figure 21: Average absolute values of inner products between the right singular vectors and the input of each layer in ResNet50 trained on ImageNet dataset ($\times 10^6$).

