# OpenReview forum: "Analysis of Linear Mode Connectivity via Permutation-Based Weight Matching: With Insights into Other Permutation Search Methods"
_ICLR.cc/2025/Conference — ICLR 2025 Poster_

### Official Review · Reviewer_oBtL · 2024-10-23

**Soundness:** 3
**Presentation:** 4
**Contribution:** 2
**Rating:** 8
**Confidence:** 4

**Summary:**

This paper analyzes the algorithms for enforcing linear mode connectivity (LMC) as discussed in Ainsworth et al. (2023). The central question addressed is the underlying mechanisms that lead to the removal of barriers when applying weight matching (WM), activation matching (AM), or the straight-through estimator (STE). The authors argue that the primary reason WM and AM achieve linear connectivity between models is the alignment of the eigenvectors corresponding to large eigenvalues of the weights. Theoretically, they demonstrate that similar eigenvectors lead to similar outputs at the layer level, and that minimizing the L_2 distance between weights is equivalent to aligning these eigenvectors. Furthermore, the paper suggests that STE operates through a different mechanism to achieve low-barrier perturbations.

**Strengths:**

The study offers a valuable perspective on analyzing LMC, a phenomenon that characterizes the properties of loss surfaces and solutions discovered by stochastic gradient descent (SGD) and its analogs. The results are presented clearly and are substantiated both empirically and, in part, theoretically.

**Weaknesses:**

A key weakness is the lack of significant novelty in the paper's primary insights. Similar observations have been made in previous works, such as [1] and [2]. These papers also explore Hessian-based barrier approximations and spectral analysis to explain LMC. Although this does not undermine the findings of the current paper, these existing works should be cited and discussed.

Another concern is the transition to per-layer analysis in the discussion of WM. The authors claim that eigenvector alignment results in aligned representations, but this can be challenged. Firstly, the use of the Lipschitz constant in the analysis limits the result to essentially one activation only. When multiple layers are involved, the Lipschitz constant becomes excessively large. Secondly, [3] show that there are often no barriers at the layer level (even when barriers exist at the full network level). Although this could still be explained by eigenvectors, it weakens the connection between similar activations on the level of one layer and the lack of barriers at the full-model level.

The analysis of STE seems rather straightforward, given that this algorithm is designed solely to minimize loss at a single point (the midpoint on the linear path). Unlike WM or AM, it does not aim to identify fully "matching" models. The numerical results in Table 4 are also not fully convincing, as the difference between the barriers found using WM and STE is minimal.

Additionally, the paper claims that models trained in the same manner exhibit higher barriers at the midpoint between them. However, in the experiments of [3], it is shown that when one model generalizes better than another, the higher barrier shifts. Therefore, the assumptions in Section 3.2 require further clarification.

Minor Comments:

1 - In Section 4.4, please specify that the results in Appendix H3 and H4 are empirical.

2 - I suggest replacing Figure 6 with Table 4, as low-dimensional visualizations provide an imprecise representation of loss surface properties.

[1] Singh et al., Landscaping Linear Mode Connectivity

[2] Yunis et al., Approaching Deep Learning through the Spectral Dynamics of Weights

[3] Adilova et al., Layerwise Linear Mode Connectivity

**Questions:**

1 - Both WM and AM are iterative algorithms. How does the L_2 distance evolve with an increasing number of iterations? It is surprising that the distance shrinkage is nearly identical between WM and AM, given that AM corresponds to layer-wise feature matching, which is closely linked to reduced distances between models.

2 - While the experimental results indicate that the Taylor approximation is imprecise for estimating the barrier, could it be that the models are still relatively close? Since L_2 distance is difficult to interpret in high-dimensional spaces, do you think that including the third term of the Taylor decomposition might help, or that a more precise analysis of the barrier (not just at the midpoint) could yield better insights?

3 - How does your analysis account for the necessity of post-matching fine-tuning, which is particularly critical for networks with batch normalization layers (e.g., Jordan et al., Repair: Renormalizing Permuted Activations for Interpolation Repair)?

---

> ### Author Response · Authors · 2024-11-21
>
> Thank you for your valuable feedback.
>
> - A key weakness is the lack of significant novelty in the paper's primary insights. Similar observations have been made in previous works, such as [1] and [2]. These papers also explore Hessian-based barrier approximations and spectral analysis to explain LMC. Although this does not undermine the findings of the current paper, these existing works should be cited and discussed.
>
>
> We apologize for the lack of discussion of these papers. We have added a discussion of these to the Appendix A in the manuscript.
>
> The existence of these works does not significantly diminish the novelty of our study. Specifically:
>
> 1. [1] demonstrated that the barrier value can be approximated using a second-order Taylor approximation, but this approach is only validated for the case of spawning and has not been validated for permutations. Our work highlights the distinctiveness of these scenarios. Unlike [1], we do not merely show that a Taylor approximation can be used to derive an equation to predict the barrier value. Instead, we reveal that the model distance under permutations cannot be reduced to a level where the barrier can be well-estimated by such an approximation. This distinction underscores the unique contribution of our findings.
>
> 2. [2] explored the reduction of rank during SGD-based training and the orientation of top singular vectors via SVD on the weights. While this is relevant to our paper, [2] does not examine the effects of permutations in details. Moreover, it does not discuss the interplay between hidden layer inputs and top singular vectors under permutations. Our study addresses these gaps, thus providing complementary insights that are absent in [2].
>
> We believe these points demonstrate that our contributions remain novel and valuable, even in light of these prior works.
>
> - Another concern is the transition to per-layer analysis in the discussion of WM. The authors claim that eigenvector alignment results in aligned representations, but this can be challenged. Firstly, the use of the Lipschitz constant in the analysis limits the result to essentially one activation only. When multiple layers are involved, the Lipschitz constant becomes excessively large.
>
> Indeed, the Lipschitz constant becomes very large when considering multiple layers, making it challenging to directly extend the discussion in this paper to such cases. However, as demonstrated in the paper, the analysis of the singular vectors and the singular values appear to be crucial factors in establishing LMC, as suggested by experimental results. For example, in Figure 14, we examine the relationship between the number of large singular values, the test loss of the original ResNet20 and VGG11 models, and the test loss of the merged model using WM. As shown in this figure, when the number of large singular values is relatively small compared to the model width and the loss of the original models is low, the loss of the merged model is also low. In this paper, we have demonstrated that singular vectors corresponding to large singular values play a crucial role based on single-layer analysis. Moreover, Figure 14 suggests that our single-layer analysis yields consistent insights for multi-layer models. In this regard, we believe that singular vectors play a significant role in the analysis of multiple layers. A more detailed and specific analysis remains a topic for future research.
>
> - Secondly, [3] show that there are often no barriers at the layer level (even when barriers exist at the full network level). Although this could still be explained by eigenvectors, it weakens the connection between similar activations on the level of one layer and the lack of barriers at the full-model level.
>
> Although further investigation is required to understand why LMC holds at the layer level as shown in [3] when permutations are not used, Figure 2 in [3] demonstrates that the barrier monotonically increases when multiple layers are averaged between the two models. We expect that closing the activations of each layer could reduce the barrier that arises during this averaging process. Otherwise, it would be counterintuitive for LMC to be established by optimizing permutations to minimize the objective functions used in activation matching and weight matching.
> In our study, we focused on single-layer analysis as a first step, given the difficulty of addressing multiple layers. However, we recognize the importance of extending the analysis to multiple layers in future work.

---

> > ### Author Response · Authors · 2024-11-21
> >
> > - The analysis of STE seems rather straightforward, given that this algorithm is designed solely to minimize loss at a single point (the midpoint on the linear path). Unlike WM or AM, it does not aim to identify fully "matching" models. The numerical results in Table 4 are also not fully convincing, as the difference between the barriers found using WM and STE is minimal.
> >
> > The aim of this section is to examine the difference between WM and STE from the perspective of singular vectors. By doing so, we also aim to provide additional support for the validity of our argument from an alternative perspective. While the analysis presented here is straightforward, we believe it is adequate for achieving the above objective within the scope of this paper. A more detailed investigation of STE will need to be addressed in future work.
> >
> > Finally, regarding the difference between WM and STE, Table 5 (correspondig to Table 4 in the first submitted paper) shows that the accuracy barrier between $\pi(\theta_b)$ and $\pi(\theta_c)$ differs by at least 20\% between WM and STE for ResNet20 and VGG11 on the CIFAR10 dataset. This result is based on the average of three independent trials (i.e., we repeated the following process three times: prepared three pre-trained models, and searched for permutations). This difference is statistically significant, and we believe that it reflects a fundamental difference between WM and STE.
> >
> > - Additionally, the paper claims that models trained in the same manner exhibit higher barriers at the midpoint between them. However, in the experiments of [3], it is shown that when one model generalizes better than another, the higher barrier shifts. Therefore, the assumptions in Section 3.2 require further clarification.
> >
> > At least in our experimental results, the mid-point showed the highest barrier value (see, for example, Figure 17). This is would be because the two models generalize to the same extent. We have added an explanation of this to Section 3.2 in the manuscript.
> >
> > - In Section 4.4, please specify that the results in Appendix H3 and H4 are empirical.
> >
> > Thank you for your suggestion. As you pointed out, we have modified the manuscript to clarify that these results are empirical.
> >
> > - I suggest replacing Figure 6 with Table 4, as low-dimensional visualizations provide an imprecise representation of loss surface properties.
> >
> > Thank you for your suggestion. It is indeed difficult to see the difference in the 2-dimensional landscape, so we moved Figure 6 to Appendix H.7 and changed to a new table showing the barrier between $\pi_b(\theta_b)$ and $\pi_c(\theta_c)$.
> >
> > - Both WM and AM are iterative algorithms. How does the L_2 distance evolve with an increasing number of iterations? It is surprising that the distance shrinkage is nearly identical between WM and AM, given that AM corresponds to layer-wise feature matching, which is closely linked to reduced distances between models.
> >
> > WM includes two main approaches: one involves solving the LAP iteratively for each layer as proposed by Ainsworth et al. (2023), and the other employs Sinkhorn's algorithm as suggested by Peña et al. (2023). Since the latter also requires iterative parameter updates, both approaches are inherently iterative methods. As noted in the main text, this study adopts the latter method using Sinkhorn's algorithm, as it enables simultaneous optimization of the permutations for the entire model.
> >
> > In contrast, AM is not an iterative algorithm. By calculating the similarity of activations between two models for each layer and using it as a cost matrix, the optimal permutation matrix can be obtained in a single step by solving the LAP. Therefore, we only examined the results of WM here. We have included experimental results in Appendix H.1 showing the decrease in $L_2$ distance during the training of WM.
> >
> > - While the experimental results indicate that the Taylor approximation is imprecise for estimating the barrier, could it be that the models are still relatively close? Since L_2 distance is difficult to interpret in high-dimensional spaces, do you think that including the third term of the Taylor decomposition might help, or that a more precise analysis of the barrier (not just at the midpoint) could yield better insights?
> >
> > For the second-order Taylor approximation to hold, one necessary condition is that the loss function must be at least class $C^3$ across all points between the two parameters. Intuitively, the inability of permutations to significantly reduce the distance between the two models suggests that the Taylor approximation might not be valid in the first place, primarily due to the use of the ReLU activation function. ReLU introduces non-differentiable points, making it likely that the path between the weights passes through discontinuities. It is probably difficult to avoid passing through discontinuous points even if we change the way the distance function is taken.

---

> > > ### Author Response · Authors · 2024-11-21
> > >
> > > - How does your analysis account for the necessity of post-matching fine-tuning, which is particularly critical for networks with batch normalization layers (e.g., Jordan et al., Repair: Renormalizing Permuted Activations for Interpolation Repair)?
> > >
> > > To begin with, WM is a permutation search method that reduces the distance between weights and does not account for batch normalization. As a result, the analysis in this paper does not include any explicit discussion of batch normalization layers. However, the trainable parameters of batch normalization, namely, the mean and variance correction terms (i.e., $\beta$ and $\gamma$), can be considered as part of an existing fully-connected or convolutional layers. Therefore, they can be addressed within the framework of the analysis presented in this paper.

---

> > > > ### Comment · Reviewer_oBtL · 2024-11-26
> > > >
> > > > Thank you for the replies.
> > > >
> > > > Can you please clarify these sentences in your comment: "We expect that closing the activations of each layer could reduce the barrier that arises during this averaging process. Otherwise, it would be counterintuitive for LMC to be established by optimizing permutations to minimize the objective functions used in activation matching and weight matching."?
> > > >
> > > > I believe that it should be emphasized that the theorem holds only for one layer, because otherwise it is misleading in a sense of possible interpretation that alignment of eigenvectors is the condition for full LMC.
> > > >
> > > > In my view results in Table4 are less significant not because of the difference of the barrier size between STE and WM, but rather because of the relative raise of the barrier - the initial barriers for STE in case of CIFAR10 with ResNET are much more significant than the barriers obtained with WM, so I guess the barrier between two permuted models is also to be expected to be high.
> > > >
> > > > >  Intuitively, the inability of permutations to significantly reduce the distance between the two models suggests that the Taylor approximation might not be valid in the first place, primarily due to the use of the ReLU activation function. ReLU introduces non-differentiable points, making it likely that the path between the weights passes through discontinuities. It is probably difficult to avoid passing through discontinuous points even if we change the way the distance function is taken.
> > > >
> > > > But we do assume that ReLU is differentiable when we compute gradients and Hessians, why do you think it is a problem?
> > > >
> > > > > To begin with, WM is a permutation search method that reduces the distance between weights and does not account for batch normalization.
> > > >
> > > > So you explicitly exclude batch normalization parameters from the weights in the implementation?

---

> ### Author Response · Authors · 2024-11-28
>
> Thank you very much for your response. Below, we have provided answers to your additional questions.
>
> - Can you please clarify these sentences in your comment: "We expect that closing the activations of each layer could reduce the barrier that arises during this averaging process. Otherwise, it would be counterintuitive for LMC to be established by optimizing permutations to minimize the objective functions used in activation matching and weight matching."?
>
> We apologize for any confusion or lack of clarity in these sentences. The objective functions of AM and WM are designed to find a permutation that makes the hidden layer outputs of the two models close. The fact that the permutations found by these methods reduce the barrier suggests that bringing the outputs of the hidden layers closer is effective in reducing the barrier. What we intended to convey in these sentences is that bringing the outputs of the hidden layers close would also reduce the barrier when merging some layers (but not all layers) between the two models.
>
> - I believe that it should be emphasized that the theorem holds only for one layer, because otherwise it is misleading in a sense of possible interpretation that alignment of eigenvectors is the condition for full LMC.
>
> You are correct. In this paper, we did not emphasize the fact that the results were based on an analysis focusing on only a single layer of the model. However, when discussing LMC in a rigorous manner, all layers of the model should be taken into account. One of the contributions of this paper was to empirically show that the results obtained from an analysis of just one layer can provide valuable insight into the merging of entire models, including all layers.
>
> We have revised Sections 1 and 7 to emphasize this point in the manuscript, with the changes highlighted in blue. In the Conclusion (Section 7), we added the following statement: "In this paper, we primarily analyzed WM from the perspective of individual layers (e.g., Theorem 4.2). However, since the actual network is multi-layered, it remains unclear why our analysis can explain the phenomenon so effectively. In the future, a more comprehensive analysis that accounts for the multi-layered structure of the network will be necessary."
>
> - In my view results in Table4 are less significant not because of the difference of the barrier size between STE and WM, but rather because of the relative raise of the barrier - the initial barriers for STE in case of CIFAR10 with ResNET are much more significant than the barriers obtained with WM, so I guess the barrier between two permuted models is also to be expected to be high.
>
> This is certainly true for ResNet20. However, as shown in Table 5, the results for VGG11 reveal a different trend. Specifically, the accuracy barrier between $\theta_a$ and $\pi_b(\theta_b)$ and the accuracy barrier between $\theta_a$ and $\pi_c(\theta_c)$ are smaller for STE compared to WM. In contrast, the barrier between $\pi_b(\theta_b)$ and $\pi_c(\theta_c)$ is larger for STE than for WM.
>
> These observations suggest that, at least for VGG11, the large barrier between $\pi_b(\theta_b)$ and $\pi_c(\theta_c)$ in STE is not merely due to the permutations found by STE being of lower quality than those found by WM (i.e., failing to effectively reduce the barriers). Instead, we believe that this behavior is driven by a more fundamental underlying cause.
>
> - But we do assume that ReLU is differentiable when we compute gradients and Hessians, why do you think it is a problem?
>
> The ReLU function is non-differentiable only at the point where the input is 0. In other words, it is differentiable at all other points. When calculating the test loss, we use the outputs corresponding to the test data inputs fed into the model. If the distribution of the test data is continuous, the probability of the model encountering non-differentiable points is 0, provided that the model's weights do not take special values. Thus, we can compute the gradient for the given weights and data.
>
> However, a point along the linear interpolation between the parameters $\theta_a$ and $\pi_b(\theta_b)$ may pass through a non-differentiable region. Specifically, if the distance between the two models is large enough to cross the non-differentiable region, the barrier estimation using Eq.(2) could fail. On the other hand, if the interpolation avoids the non-differentiable region, Eq.(2) remains valid. In such cases, the approximation accuracy depends on the order of the Taylor approximation.
>
> In this experiment, it is unclear whether the discrepancy between the barrier estimates based on the Taylor approximation and the actual barrier values is due to the interpolation passing through a non-differentiable region or the low order of the Taylor approximation. What can be stated with confidence, however, is that a large distance between the two models increases the likelihood of such issues arising.

---

> > ### Author Response · Authors · 2024-11-28
> >
> > - So you explicitly exclude batch normalization parameters from the weights in the implementation?
> >
> > We apologize, but there was an inaccuracy in the previous response. When performing WM, we used the implementation from the following GitHub repository.
> >
> > https://github.com/fagp/sinkhorn-rebasin/tree/main
> >
> > This implementation does not consider the mean and variance of batch normalization (i.e., `running_mean` and `running_var` in `torch.nn.BatchNorm2d`) in the permutation search for WM but does utilize the learnable parameters $\beta$ and $\gamma$. Accordingly, the results presented in this paper also reflect the use of these parameters. Therefore, the implementation of WM for permutation search partially incorporates batch normalization. Note that the effect of the trainable parameters of batch normalization layers on the permutation search based on WM is likely negligible because the proportion of batch normalization parameters to the total number of parameters is very small (approximately 0.015\% for VGG11 and 0.029\% for ResNet20).
> >
> > In our analysis, we primarily focused on the weight matrices and convolutional kernels, excluding the batch normalization parameters $\beta$ and $\gamma$. However, as mentioned in the previous response, these parameters can be included within the framework of this paper's analysis by treating them as part of the weights and biases of the existing fully connected and convolutional layers.

---

### Official Review · Reviewer_PdBc · 2024-10-31

**Soundness:** 3
**Presentation:** 3
**Contribution:** 2
**Rating:** 6
**Confidence:** 3

**Summary:**

This paper analyses the linear mode connectivity (LMC) using weight matching (WM), activation matching (AM), and straight-through estimator (STE), with a primary focus on WM. The authors propose that, even though WM does not significantly decrease the $L_2$ distance of the weights between two independently trained models, it still establishes LMC by aligning the singular vectors of each layer associated with the largest singular values. Empirically, the paper demonstrates that two models with a large $L_2$ weight distance after WM can nonetheless produce similar outputs, and empirical results support that singular vectors with largest singular values across the layers are aligned with WM process. Furthermore, it has been demonstrated that AM process exhibits a similar effect, while STE behaves differently. The theoretical analysis leverages SVD on layer weights, and numerical experiments using VGG, ResNet, and MLP on MNIST, CIFAR10, and ImageNet support the claims.

**Strengths:**

* The paper is clearly written and easy to follow.

* I have checked the proofs of the theorems presented in the main paper, and they appear correct and clearly articulated.

* The observation that $L_2$ distance reduction via WM is not the sole cause of LMC is interesting. This observation is supported with numerical experiments.

* The theoretical analysis is clearly explained when the underlying assumptions hold (though I have questions about the assumptions' validity; see the Weaknesses section).

* The appendix provides a good amount of detail for reproducing experiments, although the code itself is not shared.

**Weaknesses:**

* In line 70, the sentence "... this intuition is inaccurate ..." raises a concern. I think the intuition presented above this sentence is correct, i.e., if $\theta_a \approx \pi (\theta_b)$ then it is reasonable to think that the interpolated model will produce similar outputs. I believe the main narrative of the paper appears to be that even when $\theta_a \not\approx \pi (\theta_b)$, where $\pi$ is found by WM, the interpolated model can produce approximately equal outputs.

* In line 211, the phrase "... algorithm similar to the backpropagation method to efficiently compute $\mu^T H$ ..." would benefit from further clarification in the appendix. I couldn’t locate a section elaborating on this point. What specifically do the authors mean by "similar algorithm to backprop"?

* Regarding the sentence in line 261 "... we can find permutation matrices such that the left and right singular vectors of the two models match ...": it seems unlikely that all singular vectors can be matched purely through permutation in all scenarios.

* In line 286 "distributions of the singular values of the weights ... almost equal": This seems like a strong claim to me. The paper presents results with common datasets like MNIST and CIFAR10 using MLP, VGG, and ResNet. I think the conclusion requires further evidence or a more theoretical foundation. The appendix H.1 discusses this in a more detailed manner, and Figure 7 indicates that output layers' singular value distribution have variability. For this case, the paper suggests that the singular values are approximately a constant multiple across the two models. Even though the output layers' singular value distributions seem correlated, the "constant multiple" assumption seems like a strong one to me as there are contradicting examples in Figure 7 (e.g. Figure 7(a) light blue curve and red curve).

* The assumption that the input vector's direction aligns with the singular vectors associated with large singular values also seems like a strong claim. Analysis in Appendix H.8 with ResNet50 on ImageNet shows that this assumption does not always hold.

* The abbreviation "layer-wise model compression (LMC)" in line 890 conflicts with the abbreviation for linear mode connectivity, and I believe layer-wise model compression does not need an abbreviation.

**Questions:**

* For the sentence in line 139 "Entezari et al. (2022) conjectured that for SGD solutions ...": is this still a conjecture, or has a more solid theoretical foundation been established?

* In line 122, why is $W_{\ell + 1}' = W_{\ell + 1} P^{T}$ while $W_\ell' = P W_\ell$? I thought it should be $W_{\ell + 1}' = P W_{l+1} $

---

> ### Author Response · Authors · 2024-11-21
>
> Thank you very much for reading our paper so carefully. We especially appreciate that you read and commented on the details of the paper, such as the proofs.
>
> - In line 70, the sentence "... this intuition is inaccurate ..." raises a concern. I think the intuition presented above this sentence is correct, i.e., if $\theta_a \approx \pi(\theta_b)$ then it is reasonable to think that the interpolated model will produce similar outputs. I believe the main narrative of the paper appears to be that even when , where $\theta_a \not\approx \pi(\theta_b)$ is found by WM, the interpolated model can produce approximately equal outputs.
>
> You are absolutely right; the phrase "this intuition is inaccurate" was misleading. The intuition that $\theta_a \approx \pi(\theta_b)$ implies the establishment of LMC is correct. More precisely, what we intended to convey here is that this intuition is not the underlying reason why LMC holds. Even when $\theta_a \not\approx \pi(\theta_b)$, LMC can still hold by using WM. We have revised and updated Section 1 of our manuscript accordingly.
>
> - In line 211, the phrase "... algorithm similar to the backpropagation method to efficiently compute $\mu^\top H$ ..." would benefit from further clarification in the appendix. I couldn’t locate a section elaborating on this point. What specifically do the authors mean by "similar algorithm to backdrop"?
>
> We apologize for not including the details. We used the `vhp` function provided in PyTorch to compute $\boldsymbol{\mu}^\top \boldsymbol{H}$. The explanation for this computation is detailed in Chapter 5.4.6 of Bishop (2006). It is known that $\boldsymbol{\mu}^\top \boldsymbol{H}$ can be efficiently computed by applying the differential operator $\boldsymbol{\mu}^\top \nabla$ to the equations of forward and backward propagation. Consequently, the equations for evaluating $\boldsymbol{\mu}^\top \boldsymbol{H}$ closely resemble those of standard forward and backward propagation. PyTorch offers the `torch.autograd.functional.vhp` function for this purpose, which we used for our calculations. We have modified the sentence in our manuscript to clarify that we used this function to calculate $\boldsymbol{\mu}^\top \boldsymbol{H}$.
>
> https://pytorch.org/docs/stable/generated/torch.autograd.functional.vhp.html
>
> - Regarding the sentence in line 261 "... we can find permutation matrices such that the left and right singular vectors of the two models match ...": it seems unlikely that all singular vectors can be matched purely through permutation in all scenarios.
>
> As you pointed out, the explanation may be misleading, so we will rewrite "If the singular values of both models are almost equal (i.e., for all $\ell$ and $i$, $s_{\ell, i}^{(a)} = s_{\ell,i}^{(b)}$) and we can find permutation matrices such that the left and right singular vectors of the two models match, then two models $\theta_a$ and $\theta_b$ will be exactly the same and LMC clearly holds." as “The $L_2$ distance between the models is expressed by the difference in singular values and singular vectors between the two models, as indicated by Equation (3).”
>
>
> - In line 286 "distributions of the singular values of the weights ... almost equal": This seems like a strong claim to me. The paper presents results with common datasets like MNIST and CIFAR10 using MLP, VGG, and ResNet. I think the conclusion requires further evidence or a more theoretical foundation. The appendix H.1 discusses this in a more detailed manner, and Figure 7 indicates that output layers' singular value distribution have variability. For this case, the paper suggests that the singular values are approximately a constant multiple across the two models. Even though the output layers' singular value distributions seem correlated, the "constant multiple" assumption seems like a strong one to me as there are contradicting examples in Figure 7 (e.g. Figure 7(a) light blue curve and red curve).
>
> Indeed, this might seem like a strong argument at first glance. In our experiments, we observed that the distribution of singular values is nearly identical across independently trained models, at least within the hidden layers shown in Figure 7. However, it remains unclear whether this observation generalizes to other datasets or broader tasks. Providing theoretical support for this would require a detailed analysis of the weights trained by SGD, which is likely to be highly challenging. This remains an open question for future research.
>
> Finally, we have added an additional figure (Figure 8) to the paper, demonstrating that correcting the output layer by a constant factor can account for the differences in the distribution. While some discrepancies persist, the figure illustrates that these differences can largely be attributed to constant factors in all cases.

---

> > ### Author Response · Authors · 2024-11-21
> >
> > - The assumption that the input vector's direction aligns with the singular vectors associated with large singular values also seems like a strong claim. Analysis in Appendix H.8 with ResNet50 on ImageNet shows that this assumption does not always hold.
> >
> > We apologize for potentially misleading wording in the paper. This assumption holds only for wide models. We have added a note about this as the third footnote in the paper.
> >
> >
> > Table 6(a) shows that applying WM significantly improves test accuracy; however, it remains clearly lower than the original model's test accuracy of 75\%. The limited improvement in test accuracy with WM in ResNet50 is likely due to the fact that, as shown in Figure 21, the inner products between the right singular vectors with large singular values and the hidden layer inputs are not consistently large. In contrast, the ResNet20 results depicted in Figure 12 demonstrate that these inner products increase as the model width grows. Therefore, to further enhance the test accuracy of the merged model in ResNet50, increasing the model width would be necessary. In fact, Ainsworth et al. (2023) also experimentally showed that increasing the width of ResNet50 results in a smaller barrier.
> >
> > - The abbreviation "layer-wise model compression (LMC)" in line 890 conflicts with the abbreviation for linear mode connectivity, and I believe layer-wise model compression does not need an abbreviation.
> >
> > We apologize for the error. It was a misprint. The correct term is linear mode connectivity (LMC). The manuscript has been updated accordingly.
> >
> > - For the sentence in line 139 "Entezari et al. (2022) conjectured that for SGD solutions ...": is this still a conjecture, or has a more solid theoretical foundation been established?
> >
> > Through empirical analysis, we have partially clarified the principles by which the permutation symmetries of neural networks facilitate the establishment of LMC. However, this analysis does not constitute a theoretical proof. Thus, the conjecture of Entezari et al. (2022) remains unproven. Nonetheless, we believe that our experimental findings represent essential steps toward ultimately proving this conjecture. We hope that the results of this study will contribute to the theoretical foundation necessary for such proof in the future.
> >
> > - In line 122, why is $\boldsymbol{W}\_{\ell+1}' = \boldsymbol{W}\_{\ell+1} \boldsymbol{P}^\top$ while $\boldsymbol{W}\_{\ell}' = \boldsymbol{P} \boldsymbol{W}\_\ell$? I thought it should be $\boldsymbol{W}\_{\ell+1}' = \boldsymbol{P}\boldsymbol{W}\_{\ell+1}$.
> >
> > Line 122 describes how to modify the weights so that the permutation matrix is applied to the $\ell$-th layer without changing the output of the neural network. Considering the outputs of the $\ell+1$-st layer, we have
> > $$
> > \begin{align}
> > \boldsymbol{z}\_{\ell+1} &= \sigma(\boldsymbol{W}\_{\ell+1} \boldsymbol{z}\_{\ell} + \boldsymbol{b}\_{\ell+1}) \\\\
> > &= \sigma(\boldsymbol{W}\_{\ell+1} \sigma(\boldsymbol{W}\_{\ell} \boldsymbol{z}\_{\ell-1} + \boldsymbol{b}\_{\ell}) + \boldsymbol{b}\_{\ell+1}) \\\\
> > &= \sigma(\boldsymbol{W}\_{\ell+1} \boldsymbol{P}^\top \boldsymbol{P} \sigma(\boldsymbol{W}\_{\ell} \boldsymbol{z}\_{\ell-1} + \boldsymbol{b}\_{\ell}) + \boldsymbol{b}\_{\ell+1}) \\\\
> > &= \sigma(\boldsymbol{W}\_{\ell+1} \boldsymbol{P}^\top \sigma(\boldsymbol{P} \boldsymbol{W}\_{\ell} \boldsymbol{z}\_{\ell-1} + \boldsymbol{P}\boldsymbol{b}\_{\ell}) + \boldsymbol{b}\_{\ell+1}).
> > \end{align}
> > $$
> > Thus, to apply the permutation matrix $\boldsymbol{P}$ to the $\ell$-th layer without changing the output, the weights must be adjusted as follows: $\boldsymbol{W}'\_{\ell+1} = \boldsymbol{W}\_{\ell+1} \boldsymbol{P}^\top$,  $\boldsymbol{W}'\_{\ell} = \boldsymbol{P} \boldsymbol{W}\_\ell$, and $\boldsymbol{b}'\_\ell = \boldsymbol{P} \boldsymbol{b}\_\ell$.

---

> > > ### Comment · Reviewer_PdBc · 2024-11-25
> > >
> > > I thank the authors for their detailed responses and additional clarifications. I think the discussion points were crucial for improving clarity, and I appreciate the authors' effort in providing thorough explanations and revisions. I have increased my score from 5 to 6.

---

> > > > ### Author Response · Authors · 2024-11-26
> > > >
> > > > Thank you very much for your thoughtful review and for increasing your score. We truly appreciate your recognition of our efforts to improve clarity through the discussion.

---

### Official Review · Reviewer_hUsi · 2024-11-03

**Soundness:** 3
**Presentation:** 4
**Contribution:** 3
**Rating:** 8
**Confidence:** 3

**Summary:**

This work presents a novel analysis of weight matching (WM) for model merging, with comparisons to AM and STE. The authors demonstrate that permutations found by WM may not reduce the L2 distance between 2 models, albeit LMC is satisfied. Instead they show that the singular vectors are aligned. They also demonstrate that STE merging does not align singular vectors.

**Strengths:**

- This work covers a very important topic likely to have a broad impact in understanding learning and combining in deep neural networks.
- Paper is very well written; notation is clear and consistent, the problem is motivated, contributions are clear.
- The Taylor approximation to estimate the barrier for merging models provides a clear intuition for their approach, and demonstrate that WM permutations may not result in the best merge. Results in Table 1 provide good evidence for this stance (though perhaps too many unnecessary significant figures are given and appear arbitrary as a result).
- Results overall support the paper claims and demonstrate the different merging means for WM, STE, and AM.

**Weaknesses:**

- Table 1 (and others) could be improved through highlighting sections to support the conclusion, e.g., where some solutions violate the nearness assumption and the Taylor approx. is not good. Many values presented are overlapping with over values and it is not immediately clear which are significant different vs not.
- Figures are small and difficult to read. For the distribution plots (Fig. 1, 4), perhaps a logscale on the y axis would allow the reader to better evaluate the claims.
- "the barrier between πb(θb) and πc(θc) is smaller with WM than with STE. This means that there is a significant difference between the principles of permutations obtained by WM and STE." (line 519) is difficult to evaluate by eye in Figure 6 and not particularly convincing.
- The statement "suggesting that the reason AM achieves LMC is likely the same as for WM" (line 458) seems overly confident given the evidence presented. Are there other explanations for the similar results?

**Questions:**

- Connecting it back to the idea in the intro about SGD and the loss landscape, do the authors have any thoughts on how their findings relate to the efficacy of SGD?
- How well does this analysis scale beyond 2 or 3 models? "Therefore, WM is likely to be more advantageous, especially for merging three or more models" (line 522), is there additional proof of this?

---

> ### Author Response · Authors · 2024-11-21
>
> Thank you for your valuable comments.
>
> - Table 1 (and others) could be improved through highlighting sections to support the conclusion
> - Figures are small and difficult to read. For the distribution plots (Fig. 1, 4),
> - "the barrier between $\pi_b(\theta_b)$ and $\pi_c(\theta_c)$ is smaller with WM than with STE. This means that there is a significant difference between the principles of permutations obtained by WM and STE." (line 519) is difficult to evaluate by eye in Figure 6 and not particularly convincing.
>
> We are sorry about the difficulty in reading the tables and figures. For Table 1, a column labeled "Diff." was added to display the difference between the Taylor approximation and the actual barrier values, making the differences easier to read. Additionally, a t-test was performed, and values in the "Diff." column were bolded if the differences were statistically significant.
>
> Next, regarding the suggestion to use a logarithmic scale for the vertical axes of Figures 1 and 4 for better readability, we tested it but found no significant improvement in visibility. Therefore, we have kept the current format. However, we increased the font size in Figure 4 as it was previously too small.
>
> Finally, regarding the comment that the difference between STE and WM is difficult to see from the figures, we have moved the accuracy landscape figures to the appendix and introduced a new Table 3. This table represents the values of the loss and accuracy barriers between $\pi_b(\theta_b)$ and $\pi_c(\theta_c)$, making the differences clearer.
>
> - The statement "suggesting that the reason AM achieves LMC is likely the same as for WM" (line 458) seems overly confident given the evidence presented. Are there other explanations for the similar results?
>
> When using WM and AM, the $R$ values and distance reduction rates (Figures 2 and 15, Tables 1 and 4) between the two models are very similar. Therefore, we concluded that the principles of WM and AM are fundamentally the same. However, some degree of uncertainty remains, as they are not entirely identical. Meanwhile, providing additional evidence would be challenging. As a result, we have opted to moderate some of the more emphatic claims in the text.
>
> - Connecting it back to the idea in the intro about SGD and the loss landscape, do the authors have any thoughts on how their findings relate to the efficacy of SGD?
>
> This paper is inspired by Ainsworth et al. (2023). Section 1 in their paper states that if the solutions of SGD form a basin when considering permutation symmetries of NNs and the basin is furthermore a convex set, three key questions can be addressed:
>
> 1. Why does SGD work well?
> 2. What are all the local minima?
> 3. Why do independently trained models perform similarly?
>
> The paper demonstrated that LMC can be established by identifying an appropriate permutation between any two trained models. This partially supports the hypothesis that the solutions of SGD form a convex basin. Conversely, the existence of such permutations might also stem from a certain degree of similarity between independently trained models. In this study, we analyzed this similarity through singular vectors and singular values. We believe that these findings contribute to answering the three questions regarding the efficiency of SGD mentioned earlier.
>
> - How well does this analysis scale beyond 2 or 3 models? "Therefore, WM is likely to be more advantageous, especially for merging three or more models" (line 522), is there additional proof of this?
>
> As the number of models increases, the barriers are expected to grow; however, this analysis should scale reasonably well. The distance between models reduced by WM is small, but this reduction allows us to align a small number of singular vectors with large singular values. In other words, the rate of distance reduction is proportional to the degree of alignment of singular vectors with large singular values between the two models.
>
> For the three models $\theta_a$, $\theta_b$, and $\theta_c$, if $\theta_b$ and $\theta_c$ are brought closer to $\theta_a$, the distance between $\theta_b$ and $\theta_c$ also decreases. This process indirectly aligns the singular vectors with large singular values in $\theta_b$ and $\theta_c$, thereby reducing the barrier. In principle, the same trend should hold even as the number of models increases.
>
> In Section 6, we conducted experiments using three models to demonstrate the differences between WM and STE visually. A detailed investigation into how well this approach scales with four or more models, as well as exploring more appropriate merging methods for such cases, is left as a topic for future work.

---

> > ### Comment · Reviewer_hUsi · 2024-11-25
> >
> > Thank you for the detailed responses to my questions and those posed by the other reviewers. I will maintain my positive score.

---

> > > ### Author Response · Authors · 2024-11-26
> > >
> > > We sincerely appreciate your reply and your positive score. Your feedback has greatly contributed to making our paper more readable and improving its overall quality.

---

### Official Review · Reviewer_Q5i3 · 2024-11-03

**Soundness:** 3
**Presentation:** 3
**Contribution:** 2
**Rating:** 6
**Confidence:** 4

**Summary:**

This paper analyses the the weight matching algorithm through singular value decomposition.

Paper starts with the observation that algorithms like weight matching do not significantly reduce the  l2 distance. Followed by this observation, authors empirically verify that weight matching algorithm does not bring the models close enough such that the error barrier can be calculated by a second order Taylor approximation.

Authors then made the observation that minimizing the l2 distance wrt permutations is equivalent to maximizing the inner product wrt permutations. Using this relationship authors show that weight matching indeed matches the singular values of the layer-wise weight matrices.

Authors empirically validate that weight-matching and activation matching align the singular vectors however STE does not match the STE.

Authors empirically show that aligning via SVD can lead to better alignment for 3 models.

**Strengths:**

- Analysis of the weight matching via SVD of layerwise weight matrices is interesting.
- I find the observation that weight alignment leads to linear connectivity of more that 2 models also interesting

**Weaknesses:**

- I would suggest authors to keep the vectorized notation instead of summations to make the paper an easier read.
- Observation that minimizing the l2 distance is equivalent to  maximizing the inner product is pretty well know. [1] also leverages this observation to implement the weight matching algorithm. I found those contributions incremental.

[1] Ainsworth, S., Hayase, J., Srinivasa, S.: Git re-basin: merging models modulo permutation symmetries. In: The Eleventh International Conference on Learning Representations (2023)

**Questions:**

- What explains that permutations found by activation matching are much different from weight matching? Can we use SVD to analyze the qualitative / quantitative differences between the two methods that ostensibly minimize a similar objectives, i.e. matching the singular values.
- In [2], authors also analyzed linear connectivity of more than 3 models for checking strong linear connectivity where it is found that as width of the model increases, weight matching allows strong linear connectivity. Are there any insights we can gain from SVD when it studying strong linear connectivity.

[2] Sharma, E., Kwok D., Denton T., Roy DM, Rolnick D., and  Dziugaite G.K.: Simultaneous linear connectivity of neural networks modulo permutation. In Joint European Conference on Machine Learning and Knowledge Discovery in Databases (2024).

---

> ### Author Response · Authors · 2024-11-21
>
> Thank you for reading our paper carefully and for your constructive comments.
>
> - I would suggest authors to keep the vectorized notation instead of summations to make the paper an easier read.
>
> Thank you for your suggestion, and we apologize for the difficulty in reading the equations. While many of the equations involve summations, most are already expressed in vector form. Additionally, consolidating these expressions into matrix form would make it more challenging to highlight our core idea, namely, the inner products between left singular vectors or between right singular vectors, as shown in Equation (4). For this reason, we prefer to retain the current formulation.
>
> Nonetheless, your suggestion is highly valuable, and we will carefully consider whether other parts of the paper could be simplified or made more readable.
>
> - Observation that minimizing the l2 distance is equivalent to maximizing the inner product is pretty well know. [1] also leverages this observation to implement the weight matching algorithm. I found those contributions incremental.
>
> Indeed, [1] demonstrates that minimizing the $L_2$ distance is equivalent to maximizing the **Frobenius inner product (i.e., the inner product between the weight matrices of two models)**. However, [1] does not discuss how this relates to the inner product of singular vectors, as we have analyzed in our paper. Our findings reveal that WM maximizes the inner product of singular vectors with large singular values, which plays a key role in the formation of LMC. We believe this is an important contribution, as it has not been emphasized in existing research.

---

> ### Author Response · Authors · 2024-11-21
>
> - What explains that permutations found by activation matching are much different from weight matching? Can we use SVD to analyze the qualitative/quantitative differences between the two methods that ostensibly minimize a similar objectives, i.e. matching the singular values.
>
> At least in our paper, we did not conclude that the permutation matrices identified by WM and AM were entirely different. In our experimental results, applying the permutations found by WM and AM to the model produced very similar results, suggesting that these permutations may be closely related.
>
> We experimentally measured the distance between the permutation matrices found by these search methods when matching ResNet20 models. In ResNet20, there are 12 permutation matrices. Note that the number of permutation matrices is smaller than the total number of fully connected and convolutional layers due to the presence of skip connections. For the permutation matrices found by WM and AM at each layer, we calculated the number of matching rows and the ratio of matching rows to the matrix size. The results are summarized in the following table. For reference, the table also includes the experimental results for the permutation matrices found by STE.
>
> | WM vs AM [\%] | WM vs STE [\%] |
> | -- | -- |
> | $21.5625 \pm 4.2017$ | $0.8594 \pm 0.6988$ |
> | $28.5938 \pm 3.0931$ | $1.25 \pm 0.8469$ |
> | $25.625 \pm 4.3638$ | $0.7812 \pm 0.7308$ |
> | $23.125 \pm 4.4229$ | $0.7031 \pm 0.1747$ |
> | $22.0703 \pm 0.6623$ | $0.1172 \pm 0.1747$ |
> | $15.0391 \pm 2.3151$ | $0.1172 \pm 0.107$ |
> | $15.7031 \pm 2.0512$ | $0.3125 \pm 0.2227$ |
> | $8.0078 \pm 2.9068$ | $0.0391 \pm 0.0873$ |
> | $11.3672 \pm 1.4916$ | $0.0977 \pm 0.0691$ |
> | $8.8281 \pm 2.6291$ | $0.1758 \pm 0.107$ |
> | $11.875 \pm 3.8283$ | $0.0391 \pm 0.0535$ |
> | $6.6797 \pm 1.4884$ | $0.1367 \pm 0.131$ |
>
> The table presents the average and standard deviation of the results from five runs conducted with different seeds. The table shows that the permutation matrices found by WM and STE are entirely different, whereas those found by WM and AM are more similar compared to the results of STE. In fact, as shown in Tables 1 and 4, and Figures 2 and 15, both WM and AM reduce the distance between the two models and align singular vectors with large singular values.
>
> Finally, it is important to note that in models with smaller widths, the results may vary between AM and WM, although previous studies have focused on wider models to facilitate easier achievement of LMC. As described in the paper, WM aligns singular vectors with large singular values across the two models. In contrast, AM aims to make the outputs of the hidden layers close, prioritizing not only singular vectors with large singular values but also right singular vectors with large inner product values with the hidden layer inputs, as discussed in Section 5.
>
> In Subsection 4.4, we demonstrated that singular vectors with large singular values have large inner product values with hidden layer inputs, and therefore, the results of WM are very similar to those of AM. However, this observation applies primarily to models with large widths. When the width is small, right singular vectors with large singular values do not necessarily have large inner product values with the hidden layer inputs (see Figures 11 and 12). Consequently, for models with smaller widths, the results of WM and AM differ, and AM may yield superior performance.
>
> - In [2], authors also analyzed linear connectivity of more than 3 models for checking strong linear connectivity where it is found that as width of the model increases, weight matching allows strong linear connectivity. Are there any insights we can gain from SVD when it studying strong linear connectivity.
>
> The method used to evaluate strong linear connectivity in [2] is similar to the approach in our study. Specifically, three independently trained models, $\theta_a$, $\theta_b$, and $\theta_c$, are prepared, and permutations $\pi_a$ and $\pi_b$ are searched to make $\theta_a$ and $\theta_b$ close to $\theta_c$. The barrier between $\pi_a(\theta_a)$ and $\pi_b(\theta_b)$ is then calculated.
>
> By making $\theta_a$ and $\theta_b$ closer to $\theta_c$, the singular vectors between $\theta_a$ and $\theta_b$ will be aligned indirectly. At this time, increasing the model width facilitates aligning the singular vectors between models because the ratio of the number of larger singular values to the model width decreases with increasing model width (see Figure 10). In other words, widening the model width is expected to reduce the barrier between $\pi_a(\theta_a)$ and $\pi_b(\theta_b)$. Therefore, even in the case of strong linear connectivity, increasing the model width should further reduce this barrier.

---

> > ### Comment · Reviewer_Q5i3 · 2024-11-25
> > **Reponse to the rebuttal**
> >
> > I thank the authors for a detailed response and addressing my concerns. I have updated my score to reflect this.

---

> > > ### Author Response · Authors · 2024-11-26
> > >
> > > Thank you again for your thoughtful review, constructive feedback, and for updating your score. Your valuable comments were instrumental in enhancing the quality of our paper.

---

### Author Response · Authors · 2024-11-21

Dear reviewers,

Thank you for your meaningful and constructive comments. Based on your feedback, we have revised the manuscript, with the changes highlighted in red. Your comments have greatly contributed to improving the quality of the manuscript. Please note that some figure and table numbers may have shifted due to the addition of new figures and tables.

We sincerely appreciate your time and cooperation.

Best regards,

Authors

---

### Meta-Review · Area_Chair_DKid · 2024-12-19

**Metareview:**

The paper 'Analysis of Linear Mode Connectivity via Permutation-Based Weight Matching: With Insights into Other Permutation Search Methods' was reviewed by 4 reviewers who gave it an average score of 7.0 (final scores: 6+6+8+8). The reviewers found this work relevant for the conference and all of them recommend accepting this submission.

**Additional Comments On Reviewer Discussion:**

The authors posted a rebuttal and all reviewers were active during the discussion phase. The average score increased from 6.0 -> 7.0 during the discussion.

---

### Decision · Program_Chairs · 2025-01-22

Accept (Poster)